# TURBULENT FLOW SIMULATION USING AUTOREGRESSIVE CONDITIONAL DIFFUSION MODELS

## ABSTRACT

Simulating turbulent flows is crucial for a wide range of applications, and machine learning-based solvers are gaining increasing relevance. However, achieving stability when generalizing to longer rollout horizons remains a persistent challenge for learned PDE solvers. We address this challenge by demonstrating the performance of a fully data-driven fluid solver that utilizes an autoregressive rollout based on conditional diffusion models. We show that this approach offers advantages in terms of rollout stability compared to common learned baselines, and is on par with state-of-the-art stabilization techniques. Remarkably, this stability is achieved without compromising the quality of generated samples, and our model successfully generalizes to flow parameters beyond the training regime. Additionally, the probabilistic nature of the diffusion approach allows for inferring predictions that align with the statistics of the underlying physics. We quantitatively and qualitatively evaluate the performance of our method on a range of challenging scenarios, including incompressible and transonic flows, as well as isotropic turbulence.

## 1 INTRODUCTION

Simulations based on partial differential equations (PDEs), particularly those involving turbulent fluid flows, constitute a crucial research area with applications ranging from medicine (Olufsen et al., 2000) to climate research (Wyngaard, 1992), as well as numerous engineering fields (Moin & Mahesh, 1998; Verma et al., 2018). Historically, such flows have been simulated via iterative numerical solvers for the Navier-Stokes equations. Recently, there has been a growing interest in combining or replacing traditional solvers with deep learning methods. These approaches have shown considerable promise in terms of enhancing the accuracy and efficiency of fluid simulations (Wiewel et al., 2019; Han et al., 2021; Geneva & Zabaras, 2022; Stachenfeld et al., 2022).

However, despite the significant progress made in this field, a major remaining challenge is the ability to predict rollouts that maintain both stability and accuracy over longer temporal horizons (Um et al., 2020; Kochkov et al., 2021). Fluid simulations are inherently complex and dynamic, and therefore, it is highly challenging to accurately capture the intricate physical phenomena that occur over extended periods of time. Additionally, due to their chaotic nature, even small ambiguities of the spatially averaged states used for simulations can lead to fundamentally different solutions over time (Pope, 2000). However, most learned methods and traditional numerical solvers process simulation trajectories deterministically, and thus only provide a single answer.

We address these issues by exploring the usefulness of the recently emerging conditional diffusion models (Ho et al., 2020; Song et al., 2021b) for turbulent flows, which serve as representatives for more general PDE-based simulations. Specifically, we are interested in the probabilistic prediction of fluid flow trajectories from an initial condition. We aim for answering the question: *Does the increased inference cost of autoregressive diffusion models pay off in terms of posterior sampling, rollout stability, and accuracy for fluid simulations?* Our focus on fluid flows makes it possible to analyze the generated posterior samples with the statistical temporal metrics established by turbulence research (Dryden, 1943). Unlike application areas like imaging or speech, where the exact distribution of possible solutions is typically unknown, these turbulence metrics make it possible to reliably evaluate the quality of different samples generated by a probabilistic model. To summarize, the central contributions of our work are as follows: (*i*) We propose to use a conditional diffusion approach with an autoregressive rollout to produce a probabilistic surrogate simulator. This approach is robust, can

be flexibly conditioned on flow parameters, and generalizes to parameters outside the training domain. (*ii*) Most notably, we show that using a diffusion-based approach provides significant benefits over common learned autoregressive flow simulation methods, especially in terms of accuracy and stability over time. (*iii*) We additionally demonstrate that this simulator can generate physically plausible posterior samples, the statistics of which match those of the underlying ground truth physics.

## 2 RELATED WORK

**Fluid Solvers utilizing Machine Learning** A variety of works have used machine learning as means to improve numerical solvers. Several approaches focus on learning computational stencils (Bar-Sinai et al., 2019; Kochkov et al., 2021) or additive corrections (de Avila Belbute-Peres et al., 2020; Um et al., 2020; List et al., 2022) to increase simulation accuracy. In addition, differentiable solvers have been applied to solve inverse problems such as fluid control (Holl et al., 2020). An overview can be found, e.g., in Thuerey et al. (2021). When the solver is not integrated into the computational graph, typically a data-driven surrogate model is trained to replace the solver. Convolutional neural networks (CNNs) for such flow prediction problems are very popular, and often employ an encoder-processor-decoder architecture. For the latent space processor, multilayer perceptrons (Kim et al., 2019; Wu et al., 2022) as well as LSTMs (Wiewel et al., 2019) were proposed. As particularly successful latent architectures, transformers (Vaswani et al., 2017) have also been combined with CNN-based encoders as a reduced-order model (Hemmasian & Farimani, 2023), for example to simulate incompressible flows via Koopman-based latent dynamics (Geneva & Zabaras, 2022). Alternatives do not rely on an autoregressive latent model, e.g., by using spatio-temporal 3D convolutions (Deo et al., 2023), dilated convolutions (Stachenfeld et al., 2022), Bayesian neural networks for uncertainty quantification (Geneva & Zabaras, 2019), or problem-specific multi-scale architectures (Wang et al., 2020). Furthermore, various works utilize message passing architectures (Pfaff et al., 2021; Brandstetter et al., 2022), and adding noise to training inputs was likewise proposed to improve temporal prediction stability for graph networks (Sanchez-Gonzalez et al., 2020). Han et al. (2021) combine a transformer-based latent model with a graph network encoder and decoder.

**Diffusion Models** Diffusion models (Hyvärinen, 2005; Sohl-Dickstein et al., 2015) became popular after diffusion probabilistic models and denoising score matching were combined for high-quality unconditional image generation (Ho et al., 2020). This approach has since been improved in many aspects, e.g., with meaningful latent representations (Song et al., 2021a) or better sampling (Nichol & Dhariwal, 2021). In addition, generative hybrid approaches were proposed, for instance diffusion autoencoders (Preechakul et al., 2022) or score-based latent models (Vahdat et al., 2021). Diffusion models for image generation are typically conditioned on simple class labels (Dhariwal & Nichol, 2021) or textual inputs (Saharia et al., 2022). Pre-trained diffusion models are also employed for inverse image problems (Kawar et al., 2022), and different conditioning approaches were compared for score-based models on similar tasks (Batzolis et al., 2021). Song et al. (2022) combine an unconditional diffusion model with inverse problem solving for medical applications. For an in-depth review of diffusion approaches we refer to Yang et al. (2022).

**Diffusion Models for Fluids and Temporal Prediction** Selected works have applied diffusion models to temporal prediction tasks like unconditional or text-based video generation, as well as video prediction (e.g. Ho et al., 2022; Höppe et al., 2022; Harvey et al., 2022). These methods typically directly include time as a third dimension or re-use the batch dimension (Blattmann et al., 2023). As a result, autoregressive rollouts are only used to create longer output sequences compared to the training domain, with the drawback that predictions quickly accumulate errors or lose temporal coherence. Very few works exist that apply diffusion methods to transient physical processes. Holzschuh et al. (2023) utilize score matching to solve inverse problems, while Shu et al. (2023) employ a physics-informed diffusion model for a frame-by-frame super-resolution task. Lienen et al. (2023) take early steps towards turbulent flows in 3D, via a purely generative diffusion setup based on boundary geometry information. Instead of an autoregressive approach, Yang & Sommer (2023) use physical time as a conditioning for diffusion-based fluid field prediction, but report unphysical prediction results. Contemporarily, multi-step refinement (Lippe et al., 2023) and predictor-interpolator schemes (Cachay et al., 2023) inspired by diffusion models were proposed to improve the stability of PDE predictions, but both approaches provide little variance in posterior samples. In our experiments, the former is highly sensitive to hyperparameters and lacks accuracy, and we achieve better temporal coherence compared to the Bayesian interpolator samples from Cachay et al. (2023).

## 3 AUTOREGRESSIVE CONDITIONAL DIFFUSION

Our problem setting is the following: a temporal trajectory $s^1, s^2, \ldots, s^T$ of states should be predicted given an initial state $s^0$. Each $s^t$ consists of dense spatial fields, like velocity, and scalar parameters such as the Reynolds number. Numerical solvers $f$ iteratively predict $s^t = f(s^{t-1})$, and we similarly propose to use a diffusion model $f_\theta$ with parameters $\theta$ to autoregressively predict $s^t \sim f_\theta(s^{t-1})$. Below, we briefly summarize the basics of diffusion models, before providing details on how to condition and unroll them to obtain stable temporal predictions of physics systems. It is important to distinguish the *simulation rollout* via $f_\theta$ from the *diffusion rollout* $p_\theta$. The former corresponds to physical time and consists of *simulation-* or *time steps* denoted by $t \in 0, 1, \ldots, T$ superscripts. The latter refers to *diffusion steps* in the Markov chain, denoted by $r \in 0, 1, \ldots, R$ subscripts.

**Preliminaries: Diffusion Models**  A denoising diffusion probabilistic model (DDPM) is a generative model based on a parameterized Markov chain, and contains a fixed forward and a learned reverse process (Ho et al., 2020). The forward process

$$q(\boldsymbol{x}_r|\boldsymbol{x}_{r-1}) = \mathcal{N}(\boldsymbol{x}_r; \sqrt{1 - \beta_r}\boldsymbol{x}_{r-1}, \beta_r\mathbf{I}) \tag{1}$$

incrementally adds Gaussian noise to the original data $\boldsymbol{x}_0$ according to a variance schedule $\beta_1, \ldots, \beta_R$ resulting in the latent variable $\boldsymbol{x}_R$, that corresponds to pure Gaussian noise. The reverse process

$$p_\theta(\boldsymbol{x}_{r-1}|\boldsymbol{x}_r) = \mathcal{N}(\boldsymbol{x}_{r-1}; \boldsymbol{\mu}_\theta(\boldsymbol{x}_r, r), \boldsymbol{\Sigma}_\theta(\boldsymbol{x}_r, r)) \tag{2}$$

contains learned transitions, i.e. $\boldsymbol{\mu}_\theta$ and $\boldsymbol{\Sigma}_\theta$ are computed by a neural network parameterized by $\theta$ given $\boldsymbol{x}_r$ and $r$. The network is trained via the variational lower bound (ELBO) using reparameterization. During inference the initial latent variable $\boldsymbol{x}_R \sim \mathcal{N}(\mathbf{0}, \mathbf{I})$ as well as the intermediate diffusion steps are sampled, leading to a probabilistic generation of $\boldsymbol{x}_0$ with a distribution that is similar to the distribution of the training data. Note that the latent space of a DDPM by construction has the same dimensionality as the input space, in contrast to, e.g., variational autoencoders (VAEs) (Kingma & Welling, 2014). Thereby, it avoids problems with the generation of high frequency details due to compressed representations. Compared to generative adversarial networks (GANs), diffusion models typically do not suffer from mode collapse problems or convergence issues (Metz et al., 2017).

**Preliminaries: Conditioning**  Practical physics simulators need to be conditioned on information like the initial state and characteristic dimensionless quantities. To obtain a conditional DDPM, we employ a concatenation-based conditioning approach (Batzolis et al., 2021): Each element $\boldsymbol{x}_0 = (\boldsymbol{d}_0, \boldsymbol{c}_0)$ of the diffusion process now consists of a data component $\boldsymbol{d}_0$ that is only available during training and a conditioning component $\boldsymbol{c}_0$ that is always given. Correspondingly, the task at inference time is the conditional prediction $P(\boldsymbol{d}_0|\boldsymbol{c}_0)$. During training, the basic DDPM algorithm remains unchanged as $\boldsymbol{x}_r = (\boldsymbol{c}_r, \boldsymbol{d}_r)$, with $\boldsymbol{c}_r \sim q(\cdot|\boldsymbol{c}_{r-1})$ and $\boldsymbol{d}_r \sim q(\cdot|\boldsymbol{d}_{r-1})$, is still produced by the incremental addition of noise during the forward process. During inference $\boldsymbol{d}_R \sim \mathcal{N}(\mathbf{0}, \mathbf{I})$ is sampled and processed in the reverse process, while $\boldsymbol{c}_0$ is known and any $\boldsymbol{c}_r$ thus can be obtained from Eq. (1), i.e.,

$$\boldsymbol{x}_r = (\boldsymbol{c}_r, \boldsymbol{d}_r) \text{ where } \boldsymbol{c}_r \sim q(\cdot|\boldsymbol{c}_{r-1}) \text{ and } \boldsymbol{d}_r \sim p_\theta(\cdot|\boldsymbol{x}_{r+1}). \tag{3}$$

Here, $q(\boldsymbol{c}_r|\boldsymbol{c}_{r-1})$ denotes the forward process for $\boldsymbol{c}$, and $\boldsymbol{d}_r \sim p_\theta(\cdot|\boldsymbol{x}_{r+1})$ is realized by discarding the prediction of $\boldsymbol{c}_r$ when evaluating $p_\theta$. A visualization of this conditioning technique is shown in Fig. 1. We found the addition of noise to the conditioning, instead of simply using $\boldsymbol{c}_0$ over the entire diffusion process, to be crucial for the temporal simulation rollout stability as detailed in Sec. 5.

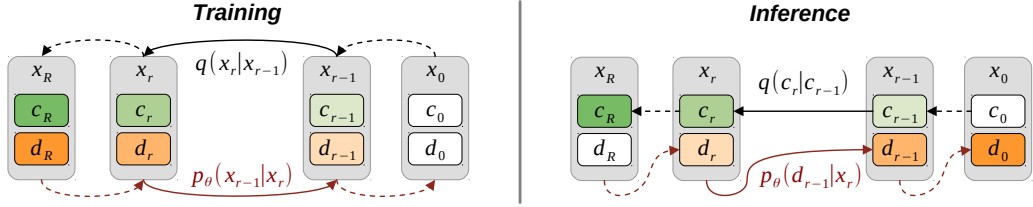

Figure 1: Diffusion conditioning approach with the forward (black) and reverse process (red) during training and inference. White backgrounds for $\boldsymbol{c}$ or $\boldsymbol{d}$ indicate given information, i.e., inputs in each phase, and $\boldsymbol{d}_0$ is the generated result during inference, i.e., the prediction of the next simulation step.

**Training and Autoregressive Rollout**  To extend the conditional setup to temporal tasks, we build on autoregressive single-step prediction as outlined above, with $k$ previous steps: $s^t \sim f_\theta(\,\cdot\,|s^{t-k}, \ldots, s^{t-1})$. The conditional DDPM for $f_\theta$ is trained in the following manner: Given a data set with different physical simulation trajectories, a random simulation state $s^t \in s^0, s^1, \ldots, s^T$ is selected from a sequence as the prediction target $d_0$. This state consists of dense simulation fields like velocity or pressure, and scalar parameters like the Mach number (see right of Fig. 2). The corresponding conditioning consists of $k$ previous simulation states $c_0 = (s^{t-k}, \ldots, s^{t-1})$. Next, a random diffusion time step $r$ is sampled, leading to $x_r$ via the forward process. The network learns to predict the added noise level via the ELBO as in the original DDPM (Ho et al., 2020).

This training objective allows for producing a single subsequent time step as the final output of the diffusion process $d_0$ during inference. We can then employ this single-step prediction for sampling simulation rollouts with arbitrary length by autoregressively reusing generated states as conditioning for the next iteration: For each simulation step, Eq. (3) is unrolled from $x_R^t$ to $x_0^t$ starting from $d_R^t \sim \mathcal{N}(\mathbf{0}, \mathbf{I})$ and $c_0^t = (s^{t-k}, \ldots, s^{t-1})$. Then, the predicted next time step is $s^t = d_0^t$. This process is visualized in Fig. 2, and we denote models trained with this approach as *autoregressive conditional diffusion models* (cDDPM) in the following.

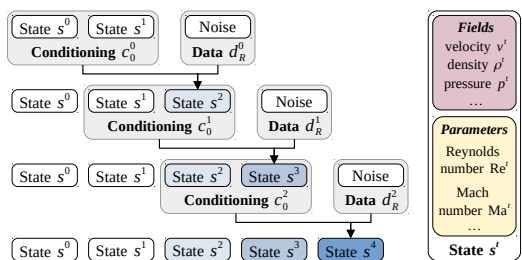

Figure 2: Autoregressive simulation rollout for $k = 2$ (left) and simulation state contents (right).

The motivation for this combination of conditioning and simulation rollout is the hypothesis that perturbations to the conditioning can be compensated during the diffusion rollout, leading to improved temporal stability. Especially so, when smaller inference errors inevitably accumulate over the course of long simulation rollouts. Furthermore, this autoregressive rollout approach ensures that the network produces a temporally coherent trajectory for every step along the inferred sequence. This stands in contrast to explicitly conditioning the DDPM on physical time $t$, i.e. treating it in the same way as the diffusion step $r$, as proposed by Yang & Sommer (2023). For a probabilistic model it is especially crucial to have access to previously generated outputs during the prediction, as otherwise temporal coherence can only potentially be achieved via tricks such as fixed temporal and spatial noise patterns.

**Implementation**  We employ a widely used U-Net (based on Ronneberger et al., 2015) with various established smaller architecture modernizations (Ho et al., 2020; Dhariwal & Nichol, 2021), to learn the reverse process with a linear variance schedule. We use $k = 2$ previous steps for the model input, and achieved high-quality samples with 20–100 diffusion steps $R$, depending on how strongly each setting is conditioned. This is in line with other recent works that achieve competitive results with as little as $R \approx 30$ in the image domain (Karras et al., 2022; Chung et al., 2022). Combining $d_0$ and $c_0$ to form $x_0$, as well as aggregating multiple states for $c_0$ is achieved via concatenation along the channel dimension. Likewise, scalar simulation parameters are concatenated as constant, spatially expanded channels.

**Baseline Models**  A crucial question is how much difference the diffusion training itself makes in comparison to a classic supervised training approach of the same backbone architecture. Hence, we use a network with identical architecture to the *cDDPM* model that is trained with such a supervised setup as a baseline. It is trained with an MSE loss to predict one future simulation state with a single model pass and denoted by *U-Net* in the following.

The success of transformer architectures (Vaswani et al., 2017) and their recent application to physics predictions (Han et al., 2021) raises the question how the *cDDPM* approach fares in comparison to state-of-the-art transformer architectures. Being tailored to sequential processing with a long-term observation horizon, these models operate on a latent space with a reduced size. In contrast, diffusion models by construction operate on the full spatial resolution. Specifically, we test the encoder-processor-decoder architecture from Han et al. (2021) adopted to regular grids via a CNN-encoding, denoted by $TF_{MGN}$ below. Furthermore, we provide an improved variant ($TF_{Enc}$) that allows to simulate flows with varying parameters over the simulation rollout, and varies key transformer parameters. Compared to $TF_{MGN}$ it relies on transformer encoder layers and uses full latent predictions instead of residual predictions. Lastly, we test the transformer-based prediction in conjunction with a

probabilistic VAE, denoted by *TF$_{VAE}$*. All transformer architectures have access to a large number of previous steps and use a rollout schedule in line with Han et al. (2021) during training, however teacher forcing is removed. By default we use a 30 step input window and a rollout length of 60.

Furthermore, we investigate dilated ResNets (based on Stachenfeld et al., 2022) and Fourier Neural Operators (FNOs) (Li et al., 2021) as two other popular approaches. For the former, the proposed dilated ResNet (*ResNet$_{dil.}$*) as well as the same architecture without dilations (*ResNet*) are included. For the latter, we show models using $(16, 8)$ Fourier modes in x- and y-direction (*FNO$_{16}$*) and $(32, 16)$ modes (*FNO$_{32}$*). To ensure a fair comparison, all models were parameterized with a similar parameter count, and suitable key hyperparameters were determined with a broad search for each architecture. Additional *cDDPM* implementation information, and details for each baseline can be found in App. B.

## 4 EXPERIMENTS

We quantitatively and qualitatively evaluate our DDPM-based simulator and the baseline models on three scenarios with increasing difficulty: (*i*) an incompressible wake flow, (*ii*) a transonic cylinder flow with shock waves, and (*iii*) an isotropic turbulence flow. Test cases for each scenario contain out-of-distribution data via simulation parameters outside of the training data range. Example visualizations are shown in Fig. 3, and experimental details are provided in App. A. We use $R = 20$ diffusion steps for each experiment unless denoted otherwise (see App. C.4 for an ablation on $R$).

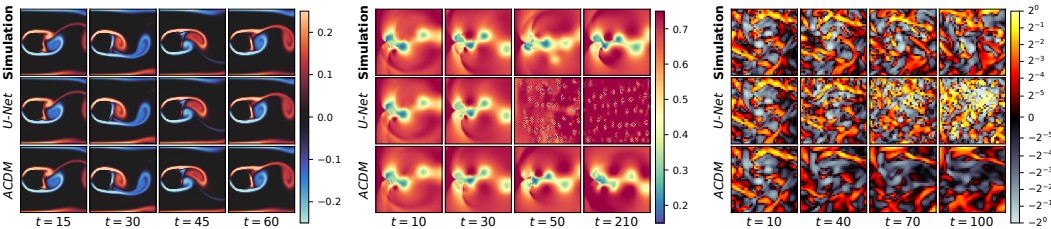

Figure 3: Zoomed examples of Inc$_{\texttt{high}}$ with $Re = 1000$ (left, vorticity), Tra$_{\texttt{long}}$ with $Ma = 0.64$ (middle, pressure), and Iso with $z = 280$ (right, vorticity). Shown are trajectories from the numerical solver, and predictions by *U-Net* and *cDDPM* (see App. D for more extensive visualizations).

**Incompressible Wake Flow**  Our first case targets incompressible wake flows. These flows already encompass the full complexity of the Navier-Stokes equations with boundary interactions, but due to their direct unsteady periodic nature represent the simplest of our three scenarios. We simulate a fully developed incompressible Karman vortex street behind a cylindrical obstacle with PhiFlow (Holl et al., 2020) for a varying Reynolds number $Re \leq 1000$. The corresponding flows capture the transition from laminar to the onset of turbulence. Models are trained on data from simulation sequences with $Re \in [200, 900]$. We evaluate generalization on the extrapolation test sets Inc$_{\texttt{low}}$ with $Re \in [100, 180]$ for $T = 60$, and Inc$_{\texttt{high}}$ with $Re \in [920, 1000]$ for $T = 60$. While all training is done with constant $Re$, we add a case with varying $Re$ as a particularly challenging test set: Inc$_{\texttt{var}}$ features a sequence of $T = 250$ steps with of a smoothly varying $Re$ from 200 to 900 over the course of the simulation time.

**Transonic Cylinder Flow**  As a significantly more complex scenario we target transonic flows. These flows require the simulation of a varying density, and exhibit the formation of shock waves that interact with the flow, especially at higher Mach numbers $Ma$. These properties make the problem highly chaotic and longer prediction rollouts especially challenging. We simulate a fully developed compressible Karman vortex street using SU2 (Economon et al., 2015) with $Re = 10000$, while varying $Ma$ in a transonic regime where shock waves start to occur. Models are trained on sequences with $Ma \in [0.53, 0.63] \cup [0.69, 0.90]$. We evaluate extrapolation on Tra$_{\texttt{ext}}$ with $Ma \in [0.50, 0.52]$ for $T = 60$, interpolation via Tra$_{\texttt{int}}$ with $Ma \in [0.66, 0.68]$ for $T = 60$, and longer rollouts of about 8 vortex shedding periods using Tra$_{\texttt{long}}$ with $Ma \in [0.64, 0.65]$ for $T = 240$.

**Isotropic Turbulence**  As a third scenario we evaluate the inference of planes from 3D isotropic turbulence. This case is inherently difficult, due to its severely underdetermined nature, as the information provided in a 2D plane allows for a large space of possible solutions, depending on the 3D motion outside of the plane. Thus, it is expected that deviations from the reference trajectories occur across methods, and we use $R = 100$ steps in *cDDPM* as a consequence. For this setup,

we observed a tradeoff between accuracy and sampling speed, where additional diffusion steps continued to improve prediction quality. As training data, we utilize 2D z-slices of 3D data from the Johns Hopkins Turbulence Database (Perlman et al., 2007). Models are trained on sequences with $z \in [1, 199] \cup [351, 1000]$, and we test on Iso with sequences from $z \in [200, 350]$ for $T = 100$.

## 5 RESULTS

In the following, we analyze the *cDDPM* posterior sampling and evaluate the different methods in terms of their general accuracy and temporal stability. Appendix C contains additional results and evaluations for various aspects discussed in the following. Unless denoted otherwise, mean and standard deviation over all sequences from each data set, multiple training runs, and multiple random model evaluations are reported. We evaluate two training runs with different random seeds for Iso, and three for Inc and Tra. For the probabilistic methods $TF_{VAE}$ and *cDDPM*, five random model evaluations are taken into account per trained model.

**Accuracy** As the basis for assessing the quality of flow predictions, we first measure direct errors towards the ground truth sequence. We use a mean-squared-error (MSE) and LSiM, a similarity metric for numerical simulations (Kohl et al., 2020). For both metrics, lower values indicate better reconstruction accuracy. Reported errors are rollout errors, i.e., computed per time step and field, and averaged over the full temporal rollout.

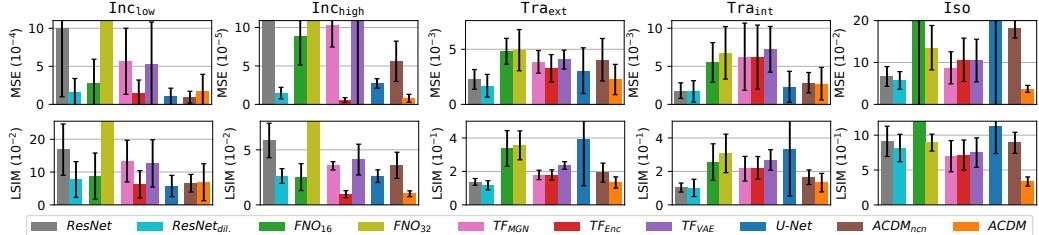

Figure 4: Quantitative comparison across our test sets for different network architectures in terms of MSE (top) and LSiM (bottom). The full, numerical accuracy results are also included in App. C.3.

As shown in Fig. 4, *cDDPM* produces accurate results with low standard deviation across all experiments and test sets. For the easiest test case Inc, all model classes can do well, as shown by the comparable performance of $ResNet_{dil.}$, $FNO_{16}$, $TF_{Enc}$, U-Net and *cDDPM* with low error. On the more complex Tra case, all transformer-based and *FNO* architectures are already substantially less accurate, and mainly the *ResNet* variants remain competitive with *cDDPM* in terms of error. However, for the longer rollouts in $Tra_{long}$ they additionally face temporal stability issues, as discussed below. On Iso, all models are struggling due to the highly underdetermined nature of this experiment. The transformer-based methods lack accuracy, as the compressed latent representations are unable to capture the high frequency details of this data set. The remaining baselines diverge at some point over the rollout as shown in App. D, while *cDDPM* remains stable and achieves an improvement of more than 35% compared to the best baseline. Below, we will only further investigate the most successful architecture in each baseline model class, i.e., $ResNet_{dil.}$, $FNO_{16}$ ($FNO_{32}$ for Iso), $TF_{Enc}$, and U-Net.

The improved performance of $ResNet_{dil.}$ compared to *ResNet*, and the generally weak results of *FNO* on our more complex tasks confirm the findings from Stachenfeld et al. (2022). The regular U-Net, despite its network structure being identical to *cDDPM*, frequently performs worse on Tra and Iso. Thus, we include an ablation on *cDDPM* in Fig. 4, that behaves similarly to U-Net in terms of error propagation: For the $cDDPM_{ncn}$ model no conditioning noise is applied, i.e., $c_0$ is used over the entire diffusion process. It performs substantially worse than *cDDPM* across cases, as it does not prevent the buildup of errors similar to the U-Net or *ResNet* baselines, due to the tight coupling between conditioning and prediction. This highlights the benefits of the *cDDPM* approach, where the next step is always created from scratch, leading to less error propagation and increased temporal stability.

**Posterior Sampling** Another attractive aspect of a DDPM-based simulator is posterior sampling, i.e., the ability to create different samples from the solution manifold for one initial condition. Below, we qualitatively and quantitatively evaluate the *cDDPM* posterior samples. First, it becomes apparent that *cDDPM* can produce samples with substantial differences. This is illustrated in Fig. 5, where

zoomed areas from three random *cDDPM* predictions on Tra_long and Iso at different time steps are displayed along with a sample from the regular simulation. The diffusion approach produces realistic, diverse features, such as the strongly varying formation of shock waves near the immersed cylinder for Tra_long. Figure 5 also visualizes the resulting spatially varying standard deviation across five samples. Due to the chaotic nature of both test cases, it increases over time, as expected. The locations of high variance match areas that are more difficult to predict, such as vortices and shock wave regions for Tra_long, and regions with high vorticity for Iso. In contrast, the posterior sampling of $TF_{VAE}$ does not manage to produce such results, as samples from one model exhibit only minor differences and high-frequency structures are either missing or unphysical, e.g., in areas where shock waves should normally occur. A posterior sample analysis for $TF_{VAE}$ is contained in App. C.2.

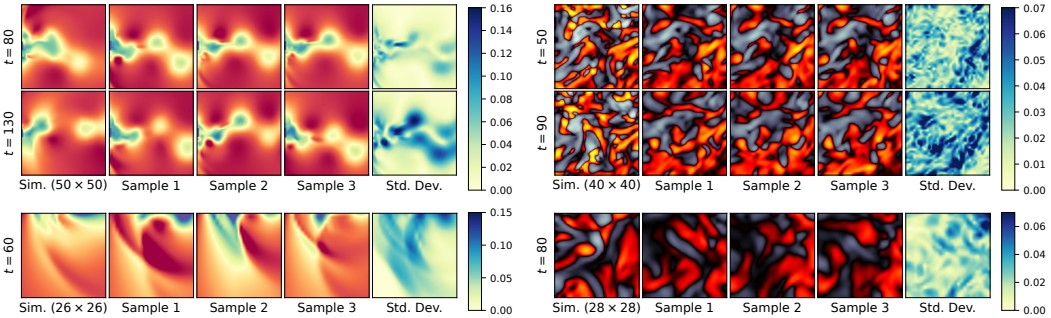

Figure 5: *cDDPM* posterior samples with corresponding standard deviation on Tra_long with $Ma = 0.64$ (left, pressure) and Iso with $z = 300$ (right, vorticity) at different zoom levels and time steps $t$.

To analyze the quality of a distribution of predicted simulation trajectories from a probabilistic algorithm, it is naturally not sufficient to directly compare to a single target sequence, as even highly accurate numerical simulations would eventually decorrelate from a target simulation over time (Hu & Liao, 2020). Instead, our experimental setups allow for using temporal and spatial evaluations to measure whether different samples *statistically* match the reference simulation, as established by turbulence research (Dryden, 1943). While a broader evaluation is provided in App. C.1, two metrics for a sequence from Tra_long are discussed here: We evaluate the wavenumber of the horizontal motion across a vertical line in the flow (averaged over time), and the temporal frequency of the vertical motion at a point probe. As shown in Fig. 6, the samples produced by *cDDPM* accurately match the statistics of the reference simulation. Even the high frequency content on the right side of the spectrum is on average correctly reproduced by the *cDDPM* outputs. Thus, the posterior samples exhibit a high correspondence to the statistical physical behavior of the reference simulations. Note that a high-quality deterministic baseline can achieve a comparable spectral mean in this evaluation.

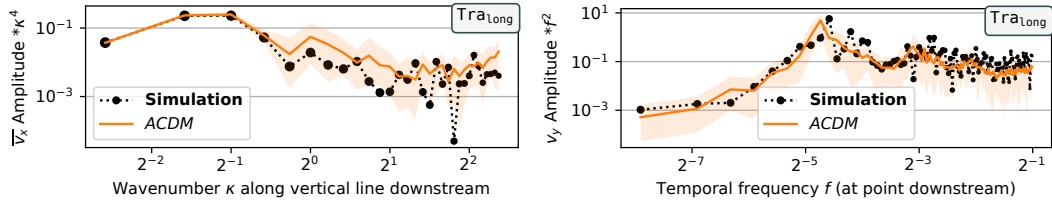

Figure 6: Spatial (left) and temporal (right) frequency analysis across posterior samples for a full sequence from Tra_long with $Ma = 0.65$. The shaded area shows the 5th to 95th percentile across all trained models and posterior samples.

**Comparing Temporal Statistics**   Temporal statistics also highlight the differences of the model architectures under consideration. The left of Fig. 7 shows an evaluation in terms of the frequency of the x-velocity for Iso averaged across every spatial point. $TF_{Enc}$ is lacking across the frequency band, while models with direct error propagation such as *ResNet_dil.*, *FNO_16*, *U-Net*, and *cDDPM_ncn* clearly overshoot. High frequencies are modeled well by *cDDPM*, but it also slightly deviates in terms of lower temporal frequencies. This is most likely caused by the strongly under-determined nature of Iso, causing *cDDPM* to unnecessarily dissipate spatial high-frequency motions, which impacts low temporal frequencies over longer rollouts. Nonetheless, it still outperforms other approaches.

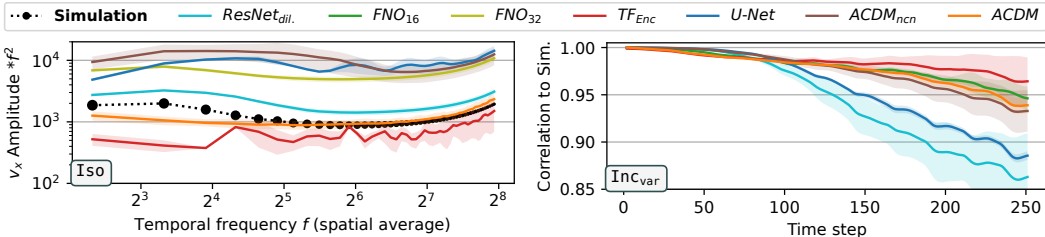

Figure 7: Temporal frequency analysis on a sequence from `Iso` with $z = 300$ (left). Correlation of predictions from different methods with the reference simulation over the rollout on `Inc_var` (right).

**Temporal Stability**  A central motivation for employing diffusion models in the context of transient simulations is the hypothesis that the stochastic training procedure leads to a more robust temporal behavior at inference time. This is especially crucial for practical applications of fluid simulations, where rollouts with thousands of steps are not uncommon. We now evaluate this aspect in more detail, first, by measuring the Pearson correlation coefficient (Pearson, 1920) between prediction and reference over time. We evaluate this for the `Inc_var` test, which contains $T = 250$ steps with a previously unseen change of the Reynolds number over the rollout (see Fig. 7 on the right). Initially, *U-Net* and *ResNet* are most accurate, but both exhibit a fast decorrelation over time. This already indicates a lack of tolerance to rollout errors observed on our more complex cases. *FNO₁₆* is on par with *cDDPM*, as `Inc_var` mainly contains low frequencies. However, the *FNO* variants have difficulties with high-frequency information, as shown below and reported, e.g., by Stachenfeld et al. (2022). *TF_Enc* keeps a high level of correlation that even slightly outperforms *cDDPM*. However, all transformer-based methods have the advantage of receiving a larger range of previous simulation states during training and inference (here $k = 30$) to predict the next state.

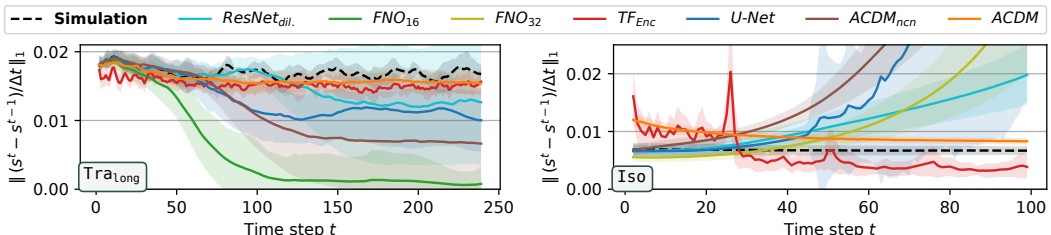

Figure 8: Stability analysis via error to previous time step on `Tra_long` (left) and `Iso` (right).

To assess the dynamics of more complex cases, we measure the magnitude of the rate of change of $s$, computed as $\|(s^t - s^{t-1})/\Delta t\|_1$ for every normalized time step. Compared to the correlation, this metric stays meaningful even for long rollout times, and indicates whether a simulator preserves the expected evolution of states as given by the reference simulation. Figure 8 shows this evaluation for `Tra_long` and `Iso`. For `Tra_long`, the reference simulation features steady oscillations as given by the main vortex shedding frequency. *TF_Enc* generally remains stable due to the long training rollout indicated by a mostly constant rate of change. However, it exhibits minor temporal inconsistencies and slightly undershoots compared to the reference, most likely due to temporal updates being performed suboptimally in the latent space. The other baselines diverge at different points during the rollout and mostly settle into a stable but clearly wrong state of a mean flow prediction without vortices. Like these baselines, the *cDDPM* simulator closely follows the reference until about $t = 60$, however it transitions to a relatively constant rate of change without diverging. Note that the vortex shedding oscillations are averaged out over posterior samples and training runs in this evaluation.

The right side of Fig. 8 repeats this evaluation for the `Iso` experiment. Its isotropic nature in combination with forcing leads to an almost constant rate of change over time for the reference simulation. All methods struggle to replicate this accurately due the underdetermined learning task. Apart from issues with the reconstruction quality, *TF_Enc* exhibits undesirable spikes corresponding to their temporal prediction window of $k = 25$ previous steps, and undershoots after one rollout window. Similar as observed on `Tra_long`, *ResNet_dil.*, *FNO*, and *U-Net* initially predict a quite accurate rate of change, but diverge at different points over the rollout. Here, the common failure mode is an incorrect

addition of energy to the system that causes a quick and significant divergence from the reference. Despite the initially slightly larger rate of change, and the decay corresponding to an overly dissipative prediction, *cDDPM* fares best among the investigated methods and remains fully stable over the simulation rollout. These results confirm the initial hypothesis, and show that the DDPM-based training not only leads to accurate instantaneous predictions, but also yields an excellent temporal stability. This mainly caused by the increased error tolerance of *cDDPM* compared to $cDDPM_{ncn}$, as the latter performs very similar to *U-Net* for both evaluated experiments in Fig. 8.

**Discussion and Limitations**   So far, we have evaluated *cDDPM* and *U-Net* with training method-ologies that were kept as similar as possible for fairness. However, several works have reported improvements from unrolling predictions during training (Lusch et al., 2018; Geneva & Zabaras, 2020). We investigate *U-Net* architectures with such unrolling over $m$ steps during training, where the gradient is fully backpropagated (see App. C.5 for details). This additional complexity at training time improves the results, for example, $U\text{-}Net_{m8}$ is fully temporally stable on `Iso`, and is on par with the *cDDPM* accuracy of 0.037 with an MSE of 0.045. Too large $m$ can deteriorate performance, but this can be mitigated via pre-training with smaller $m$. Furthermore, the usage of training noise was proposed to reduce problems from error accumulation during inference (Sanchez-Gonzalez et al., 2020). We investigate adding normally distributed noise to the *U-Net* input with varying standard deviation $n$. With a well-tuned value of $n = 10^{-2}$, *U-Net* results in an MSE of 0.0014 compared to *cDDPM* with an MSE of 0.0023 on `Tra`$_{ext}$, and does not show the issues with temporal stability exhibited by the standard *U-Net* on `Tra`$_{long}$ or `Iso` (see App. C.6 for details).

Both changes to the *U-Net* training are faster at inference time than *cDDPM*, with a factor roughly equal to the number of backbone model evaluations, i.e., diffusion steps $R$. While improvements in terms of sampling procedures are to be expected in future work, the advantages of diffusion-based approaches most likely stem from their iterative nature over the diffusion rollout, and hence we anticipate that a constant factor over deterministic, single-pass inference will remain. However, both stabilization methods naturally do not provide the *U-Net* with capabilities for posterior sampling. Unrolling also introduces additional computational overheads during training (see App. B.6). Fur-thermore, training noise necessitates hyperparameter tuning for $n$, as suboptimal values can even deteriorate performance (see details in App. C.6), while *cDDPM* works well out-of-the-box, with $R$ only serving as a balancing factor between accuracy and inference costs.

Concurrently, two methods with a similar focus as *cDDPM* were proposed: PDE-Refiner (Lippe et al., 2023), and DYffusion (Cachay et al., 2023). PDE-Refiner relies on a refinement of direct one-step predictions, by adding noise and denoising the result via diffusion within a single model. Compared to our method, it can achieve similar temporal stability in less inference time, as only about $R = 4$ model evaluations are required. However PDE-Refiner, (*i*) consistently achieves worse accuracy than *cDDPM*, (*ii*) exhibits unstable behavior across its multiple hyperparameters, (*iii*) results in worse posterior coverage compared to *cDDPM*, due to its probabilistic refinement of deterministic predictions. We provide a detailed comparison against PDE-Refiner in App. C.9. DYffusion combines a predictor, that equates the diffusion time step with the physical time step of the simulation, with a probabilistic interpolator model. Compared to *cDDPM*, the method can use large time steps, generalizes to arbitrary prediction time intervals, and is significantly faster, with little overhead compared to a standard *U-Net* inference. However, *cDDPM* achieves better temporal coherence and posterior coverage, due to the fully diffusion-based approach compared to the Bayesian interpolation.

# 6   CONCLUSION AND FUTURE WORK

We demonstrated the attractiveness of autoregressive conditional diffusion models for the simulation of complex flow phenomena. Our findings show that using a diffusion-based approach has clear advantages in terms of accuracy for complex, underdetermined flow prediction problems, while at the same time enabling probabilistic inference, that faithfully reproduces the physical statistics of the reference solutions. Our results show improved temporal stability of the diffusion-based training, which surpasses the established methodologies for classical supervised training, and transformer-based sequence modeling methods. We believe that recent advances in sampling procedures such as distillation (Salimans & Ho, 2022) are a promising avenue for improved inference performance for *cDDPM* in future work. Naturally, considering other PDEs, or larger, three-dimensional flows is likewise a highly interesting direction. For the latter, single-step *cDDPM* training is particularly attractive, as it avoids the substantial costs of temporal training rollouts (Sirignano et al., 2020).

ETHICS STATEMENT

Our method primarily focuses on an early exploration of integrating diffusion models as a beneficial tool in a larger fluid simulation toolbox. As such, we anticipate that our research will not directly result in negative societal or ethical consequences. However, it is important to acknowledge that there may be a potential military relevance, similar to existing flow simulation technologies like numerical fluid solvers. Furthermore, the broader environmental implications of deep learning as a whole are an aspect worth considering.

REPRODUCIBILITY STATEMENT

To ensure reproducibility, we provide a detailed appendix with additional information for data sets, model variants, training, and additional evaluations below. Furthermore, we provide our source code for dataset generation, training, model sampling, and evaluations alongside this submission. The source code and data sets will be publicly available upon acceptance as well.

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

## A    DATA DETAILS

In the following, we provide details for each simulation setup: the incompressible wake flow `Inc` in App. A.1, the transonic cylinder flow `Tra` in App. A.2, and the isotropic turbulence `Iso` in App. A.3. Further details can be found in the source code accompanying this submission.

### A.1    INCOMPRESSIBLE FLOW SIMULATION

To create the incompressible cylinder flow we employ the fluid solver PhiFlow[1] (Holl et al., 2020). Velocity data is stored on a staggered grid, we employ an advection scheme based on the MacCormack method, and use the adaptive conjugate gradient method as a pressure solver. We enforce a given Reynolds number in $[100, 1000]$ via an explicit diffusion step.

Our domain setup is illustrated in Fig. 9. We use Neumann boundary conditions in vertical x-direction of the domain and around the cylinder, and a Dirichlet boundary condition for the outflow on the right of the domain. For the inflow on the left of the domain we prescribe a fixed freestream velocity of $\binom{0}{0.5}$ during the simulation. To get oscillations started, the y-component of this velocity is replaced with $0.5 \cdot (\cos(\pi \cdot x) + 1)$, where $x$ denotes normalized vertical domain coordinates in $[0, 1]$, during a warmup of 20 time steps. We run and export the simulation for 1300 iterations at time step 0.05, using data after a suitable warmup period $t > 300$. The spatial domain discretization is

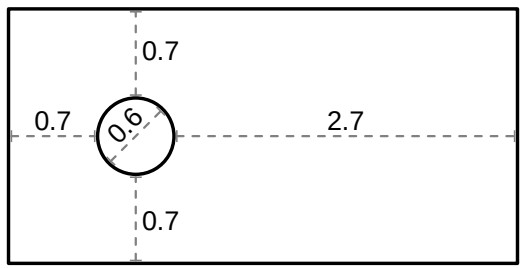

Figure 9: Simulation domain for incompressible flow simulation.

$256 \times 128$, but we train and evaluate models on a reduced resolution via downsampling the data to $128 \times 64$. Velocities are resampled to a regular grid before exporting, and pressure values are exported directly. In addition, we normalize all fields and scalar components to a standard normal distribution. The velocity is normalized in terms of magnitude. During inference we do not evaluate the cylinder area; i.e., all values inside the cylinder are set to zero via a multiplicative binary mask before every evaluation or loss computation.

We generated a data set of 91 sequences with Reynolds number $Re \in \{100, 110, \dots, 990, 1000\}$. Running and exporting the simulations on a machine with an NVIDIA GeForce GTX 1080 Ti GPU and an Intel Core i7-6850k CPU with 6 cores at 3.6 GHz took about 5 days. Models are trained using the data of 81 sequences with $Re \in \{200, 210, \dots, 890, 900\}$ for $t \in [800, 1300]$. Training and test sequences employ a temporal stride of 2. As test sets we use:

- `Inc`$_{\text{low}}$: five sequences with $Re \in \{100, 120, 140, 160, 180\}$ for $t \in [1000, 1120)$ with $T = 60$.

- `Inc`$_{\text{high}}$: five sequences with $Re \in \{920, 940, 960, 980, 1000\}$ for $t \in [1000, 1120)$ with $T = 60$.

- `Inc`$_{\text{var}}$: one sequence for $t \in [300, 800)$ with $T = 250$, and a smoothly varying $Re$ from 200 to 900 during the simulation. This is achieved via linearly interpolating the diffusivity to the corresponding value at each time step.

For the `Inc`$_{\text{var}}$ test set, we replace the model predictions of $Re$ that are learned to be constant for *cDDPM*, *U-Net*, *ResNet*, and *FNO* with the linearly varying Reynolds numbers over the simulation rollout during inference. The transformer-based methods *TF*$_{Enc}$ and *TF*$_{VAE}$ receive all scalar simulation parameters as an additional input to the latent space for each iteration of the latent processor. Note that the architectural design of *TF*$_{MGN}$ does not allow for varying simulation parameters over the rollout, as only one fixed parameter embedding is provided as a first input step for the latent processor, i.e. the model is expected to diverge quickly in this case.

---

[1]https://github.com/tum-pbs/PhiFlow

## A.2 Transonic Flow Simulation

To create the transonic cylinder flow we use the simulation framework SU2[2] (Economon et al., 2015). We employ the delayed detached eddy simulation model (SA-DDES) for turbulence closure, which is derived from the one-equation Spalart-Allmaras model (Spalart et al., 2006). By modifying the length scale, the model behaves like RANS for the attached flow in the near wall region and resolves the detached flows in the other regions. No-slip and adiabatic conditions are applied on the cylinder surface. The farfield boundary conditions are treated by local, one-dimensional Riemann-invariants. The governing equations are numerically solved by the finite-volume method. Spatial gradients are computed with weighted least squares, and the biconjugate gradient stabilized method (BiCGSTAB) is used as the implicit linear solver. For the freestream velocity we enforce a given Mach number in $[0.5, 0.9]$ while keeping the Reynolds number at a constant value of $10^4$.

To prevent issues with shockwaves from the initial flow phase, we first compute a steady RANS solution for each case for 1000 solver iterations and use that as the initialization for the unsteady simulation. We run the unsteady simulation for $150\,000$ iterations overall, and use every $50^{\text{th}}$ step once the vortex street is fully developed after the first $100\,000$ iterations. This leads to $T = 1000$ exported steps with velocity, density, and pressure fields. The non-dimensional time step for each simulation is $0.002 * \tilde{D}/\tilde{U}_\infty$, where $\tilde{D}$ is the dimensional cylinder diameter, and $\tilde{U}_\infty$ the free-stream velocity magnitude.

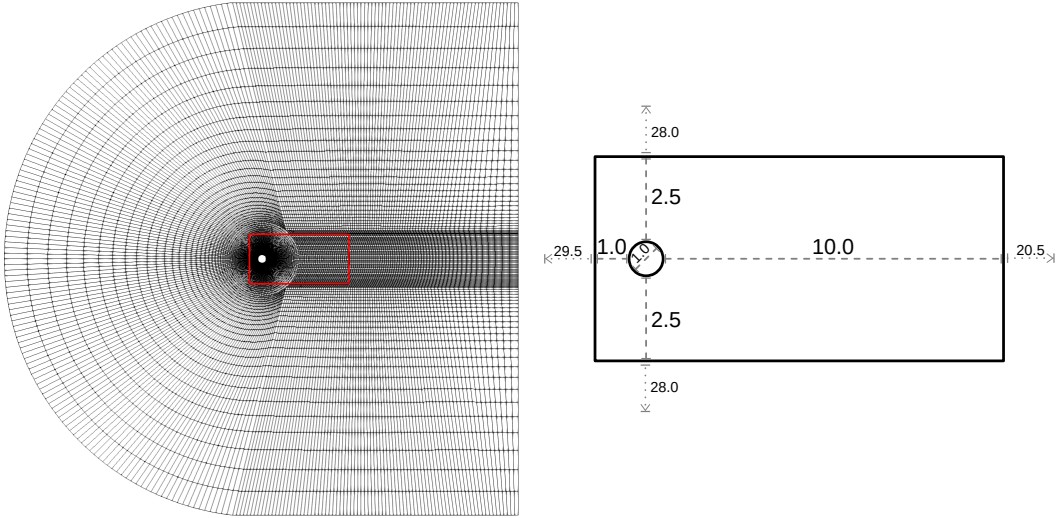

Figure 10: Full simulation mesh with highlighted resampling area (left) and resampling domain setup (right) for the transonic flow simulation.

The computational mesh is illustrated on the left in Fig. 10. Inference is focused on the near field region around the obstacle (marked in red on the left, and shown in detail on the right). To interpolate from the original mesh to the resampled training and testing domain, which is a regular, Cartesian grid with resolution $128 \times 64$, we use an interpolation based on radial basis functions. It employs a linear basis function across the 5 nearest data points of the original mesh. In terms of field normalization and masking the cylinder area during inference, we treat this case in the same way as described in App. A.1.

We created a data set of 41 sequences with Mach number $Ma \in \{0.5, 0.51, \ldots, 0.89, 0.90\}$ at Reynolds number $10^4$ with the $T = 1000$ exported steps each. We sequentially ran the simulations on one CPU cluster node that contains 28 Intel Xeon E5-2690 v3 CPU cores at 2.6 GHz in about 5 days. Each simulation was computed in parallel with 56 threads, and one separate thread simultaneously resampled and processed the simulation outputs online during the simulation. All models are trained on the data of 33 sequences with $Ma \in \{0.53, 0.54, \ldots, 0.62, 0.63\} \cup \{0.69, 0.70, \ldots, 0.89, 0.90\}$. Training and test sequences use a temporal stride of 2. The used test cases for this compressible, transonic flow setup are:

---

[2] https://su2code.github.io

- $\mathtt{Tra_{ext}}$: six sequences from $Ma \in \{0.50, 0.51, 0.52\}$, for $t \in [500, 620)$ and for $t \in [620, 740)$ with $T = 60$.
- $\mathtt{Tra_{int}}$: six sequences from $Ma \in \{0.66, 0.67, 0.68\}$, for $t \in [500, 620)$ and for $t \in [620, 740)$ with $T = 60$.
- $\mathtt{Tra_{long}}$: four sequences from $Ma \in \{0.64, 0.65\}$, for $t \in [0, 480)$ and for $t \in [480, 960)$ with $T = 240$.

### A.3 ISOTROPIC TURBULENCE

For the isotropic turbulence experiment, we make use of the 3D *isotropic1024coarse* simulation from the Johns Hopkins Turbulence Database[3] (Perlman et al., 2007). It contains simulations of forced turbulence with a direct numerical simulation (DNS) using a pseudo-spectral method on $1024^3$ nodes for 5028 time steps. The database allows for direct download queries of parameterized simulation cutouts; filtering and interpolation are already provided. We utilize sequences of individual 2D slices with a spatio-temporal starting point of $(s_x, s_y, s_z, s_t) = (1, 1, z, 1)$ and end point of $(e_x, e_y, e_z, e_t) = (256, 128, z + 1, 1000)$ for different values of $z$. A spatial striding of 2 leads to the training and evaluation resolution of $128 \times 64$. We use the pressure, as well as the velocity field including the velocity z-component. We normalize all fields to a standard normal distribution before training and inference. In this case, the velocity components are normalized individually, which is statistically comparable to a normalization in terms of magnitude for isotropic turbulence.

We utilize 1000 sequences with $z \in \{1, 2, \ldots, 999, 1000\}$ and $T = 1000$. Models are trained on 849 sequences with $z \in \{1, 2, \ldots, 198, 199\} \cup \{351, 352, \ldots, 999, 1000\}$. The test set in this case is $\mathtt{Iso}$ using 16 sequences from $z \in \{200, 210, \ldots, 340, 350\}$ for $t \in [500, 600)$, meaning $T = 100$.

## B IMPLEMENTATION AND MODEL DETAILS

Using the data generated with the techniques described above, the deep learning aspects of this work are implemented in PyTorch (Paszke et al., 2019). For every model we optimize network weights using the Adam optimizer (Kingma & Ba, 2015) with a learning rate of $10^{-4}$ (using $\beta_1 = 0.9$ and $\beta_2 = 0.999$), where the batch size is chosen as 64 by default. If models would exceed the available GPU memory, the batch size is reduced accordingly. For each epoch, the long training sequences are split into shorter parts according to the required training sequence length for each model and the temporal strides described in App. A. To prevent issues with a bias towards certain initial states, the start (and corresponding end) of each training sequence is randomly shifted forwards or backwards in time by half the sequence length every time the sequence is loaded. This is especially crucial for the oscillating cylinder flows when training models with longer rollouts. For instance, training a model with a training rollout length of 60 steps on a data set that contains vortex shedding oscillations with a period of 30 steps would lead to a correlation between certain vortex arrangements and the temporal position in the rollout during training (and inference). This could potentially lead to generalization problems when the model is confronted with a different vortex arrangement than expected at a certain time point in the rollout. The sequences for each test set are used directly without further modifications.

In the following, we provide architectural and training details for the different model architectures discussed in the main paper: *cDDPM* in App. B.1, *U-Net* in App. B.2, *ResNet* in App. B.3, *FNO* in App. B.4, and the transformer-based models in App. B.5. In addition, App. B.6 contains an overview on training cost and inference performance across architectures.

### B.1 ACDM IMPLEMENTATION

For the *cDDPM* models, we employ a "modern" U-Net architecture commonly used for diffusion models: The setup at its core follows the traditional U-Net architecture (Ronneberger et al., 2015) with an initial convolution layer, several downsampling blocks, one bottleneck block, and several upsampling blocks followed by a final convolution layer. The downsampling and upsampling block at one resolution are connected via skip connections in addition to the connections through lower layers. The modernizations mainly affect the number and composition of the blocks: We use three feature

---

[3]https://turbulence.pha.jhu.edu/

map resolutions ($128 \times 64$, $64 \times 32$, and $32 \times 16$), i.e. three down- and three upsamling blocks, with a constant number of channels of 128 at each resolution level. The down- and upsampling block at each level consists of two ConvNeXt blocks (Liu et al., 2022) and a linear attention layer (Shen et al., 2021). The bottleneck block uses a regular multi-head self-attention layer (Vaswani et al., 2017) instead. As proposed by Ho et al. (2020), we:

- use group normalization (Wu & He, 2018) throughout the blocks,
- use a diffusion time embedding for the diffusion step $r$ via a Transformer sinusoidal position embedding layer (Vaswani et al., 2017) combined with an MLP consisting of two fully connected layers, that is added to the input of every ConvNeXt block,
- train the model via reparameterization,
- and employ a linear variance schedule.

Since the variance hyperparameters provided by Ho et al. (2020) only work for a large number of diffusion steps $R$, we adjust them accordingly to fewer diffusion steps: $\beta_0 = 10^{-4} * (500/R)$ and $\beta_R = 0.02 * (500/R)$. We generally found $R = 20$ to be sufficient on the strongly conditioned data set Inc and Tra, but on the highly complex Iso data, *cDDPM* showed improvements up to about $R = 100$. The same value of $R$ is used during training and inference. In early exploration runs, we found $k = 2$ input steps to show slightly better performance compared to $k = 1$ used by *U-Net* below, and kept this choice for consistency across diffusion evaluations. However, the differences for changing the number of input steps from $k \in \{1, 2, 3, 4\}$ are minor compared to the performance difference between architectures. The resulting models are trained for 3100 epochs on Inc and Tra, and 100 epochs on Iso. All setups use a batch size of 64 during training, and employ a Huber loss, which worked better than an MSE loss. However, the performance difference between loss are marginal, compared to the difference between architectures.

For the *cDDPM$_{ncn}$* variants, we leave all these architecture and training parameters untouched, and only change the conditioning integration: Instead of adding noise to $c_0$ in the forward and reverse diffusion process at training and inference time, $c_0$ is used without alterations over the entire diffusion rollout.

## B.2 IMPLEMENTATION OF U-NET (AND U-NET VARIANTS)

For the implementation of *U-Net* we use an identical U-Net architecture as described above in App. B.1. The only difference being that the diffusion time embeddings are not necessary. The resulting model is trained with an MSE loss on the subsequent time step. In early exploration runs, we found $k = 1$ input steps to perform best for this direct next-step prediction setup with *U-Net* (and similarly for *ResNet* and *FNO* below), when investigating $k \in \{1, 2, 3, 4\}$. However, compared to the difference between architectures, these changes are minor.

The additional *U-Net* variants with time unrolling during training share the same architecture. They are likewise trained with an MSE loss applied equally to every step of the predicted rollout with length $m$ against the ground truth. A *U-Net* trained with, e.g., $m = 8$ is denoted by *U-Net$_{m8}$* below. To keep a consistent memory level during training, the batch size is reduced correspondingly when $m$ is increased. Thus, the training time of *U-Net* significantly depends on $m$. While $m = 2$ allows for a batch size of 64, $m = 4$ reduces that to 32, $m = 4$ leads to 16, and finally, for $m = 16$ the batch size is only 8. All *U-Net* variants were trained for 1000 epochs on Inc and Tra, and 100 epochs on Iso.

## B.3 IMPLEMENTATION OF DILATED RESNETS

For the implementation of *ResNet$_{dil.}$* and *ResNet*, we follow the setup proposed by Stachenfeld et al. (2022) that relies on a relatively simple architecture: both models consist of 4 blocks connected with skip connections as originally proposed by He et al. (2016). Each block consists of 8 convolution layers with kernel size 3 and stride 1, followed by ReLU activations. For the *ResNet$_{dil.}$* model, the convolution layers in each block employ the following dilation and padding values: $(1, 2, 4, 8, 4, 2, 1)$. For *ResNet*, all dilation and padding values are set to 1. Both models use a batch size of 64, receive $k = 1$ input steps, predict a single next step, and are trained via an mean-squared-error (MSE) on the prediction against the simulation trajectory as described in App. B.2.

### B.4 Implementation of FNOs

For the implementation of the *FNO* variants, we follow the official PyTorch FNO implementation.[4] The lifting and projection block setups are directly replicated from Li et al. (2021), and all models use 4 FNO layers. We vary the number of modes that are kept in in x- and y-direction in each layer as follows: $FNO_{16}$ uses $(16, 8)$ modes and $FNO_{32}$ uses $(32, 16)$ modes. To ensure a fair comparison, the hidden size of all models are parameterized to reach a number of trainable parameters similar to *cDDPM*, i.e. 112 for $FNO_{16}$ and 56 for $FNO_{32}$. Both models use a batch size of 64, receive $k = 1$ input steps, predict a single next step, and are trained via an mean-squared-error (MSE) on the prediction against the simulation trajectory as described in App. B.2.

### B.5 Transformer Implementation

To adapt the approach from Han et al. (2021) to regular grids instead of graphs, we rely on CNN-based networks to replace their Graph Mesh Reducer (GMR) network for encoding and their Graph Mesh Up-Sampling (GMUS) network for decoding. Our encoder model consists of convolution+ReLU blocks with MaxPools and skip connections. In the following, convolution parameters are given as input channels → output channels, kernel size, stride, and padding. Pooling parameters are given as kernel size, stride, and Upsampling parameters are give as scale factor in x, scale factor in y, interpolation mode. The number of channels of the original flow state are denoted by $in$, the encoder width is $w_e$, the decoder width is $w_d$, and $L$ is the size of the latent space. The encoder layers are:

1. $Conv(in \to w_e, 11, 4, 5)$ + ReLU + MaxPool$(2, 2)$
2. $Conv(w_e + in_1 \to 3 * w_e, 5, 1, 2)$ + ReLU + MaxPool$(2, 2)$
3. $Conv(3 * w_e + in_2 \to 6 * w_e, 3, 1, 1)$ + ReLU
4. $Conv(6 * w_e + in_2 \to 4 * w_e, 3, 1, 1)$ + ReLU
5. $Conv(4 * w_e + in_2 \to w_e, 3, 1, 1)$ + ReLU
6. $Conv(w_e + in_2 \to L, 1, 1, 0)$ + ReLU + MaxPool$(2, 2)$

Here, $in_1$ and $in_2$ are skip connections to spatially reduced inputs that are computed directly on the original encoder input with an AvgPool$(8, 8)$ and AvgPool$(16, 16)$ layer, respectively. Finally, the output from the last convolution layer is spatially reduced to a size of 1 via an adaptive average pooling operation. This results in a latent space with $L$ elements. This latent space is then decoded with the following decoder model based on convolution+ReLU blocks with Upsampling layers:

1. $Conv(L \to w_d, 1, 1, 0)$ + ReLU + Upsample$(4, 2, nearest)$
2. $Conv(w_d + L \to w_d, 3, 1, 1)$ + ReLU + Upsample$(2, 2, nearest)$
3. $Conv(w_d + L \to w_d, 3, 1, 1)$ + ReLU + Upsample$(2, 2, nearest)$
4. $Conv(w_d + L \to w_d, 3, 1, 1)$ + ReLU + Upsample$(2, 2, nearest)$
5. $Conv(w_d + L \to w_d, 3, 1, 1)$ + ReLU + Upsample$(2, 2, nearest)$
6. $Conv(w_d + L \to w_d, 3, 1, 1)$ + ReLU + Upsample$(2, 2, bilinear)$
7. $Conv(w_d + L \to w_d, 5, 1, 2)$ + ReLU
8. $Conv(w_d + L \to w_d, 3, 1, 1)$ + ReLU
9. $Conv(w_d \to in, 3, 1, 1)$

Here, the latent space is concatenated along the channel dimension and spatially expanded to match the corresponding spatial input size of each layer for the skip connections. In our implementation, an encoder width of $w_e = 32$, a decoder width of $w_d = 96$ with a latent space dimensionality of $L = 32$ worked best across experiments. For the model $TF_{Enc}$ on the experiments `Inc` and `Tra`, we employ $L = 31$ and concatenate the scalar simulation parameter that is used for conditioning, i.e., Reynolds number for `Inc` and Mach number for `Tra`, to every instance of the latent space. For $TF_{VAE}$ we proceed identically, but here every latent space element consists of two network weights for mean and variance via reparameterization as detailed by Kingma & Welling (2014). For $TF_{MGN}$, we use an additional first latent space of size $L$ that contains a simulation parameter encoding via an MLP as proposed by Han et al. (2021). Compared to our improved approach, this means $TF_{MGN}$ is not capable to change this quantity over the course of the simulation.

---

[4]https://github.com/NeuralOperator/neuraloperator

For the latent processor in $TF_{MGN}$ we directly follow the original transformer specifications of Han et al. (2021) via a single transformer decoder layer with four attention heads and a layer width of 1024. Latent predictions are learned as a residual from the previous step. For our adaptations $TF_{Enc}$ and $TF_{VAE}$, we instead use a single transformer encoder layer and learn a full new latent state instead of a residual prediction.

To train the different transformer variants end-to-end, we always use a batch size of 8. We train each model with a training rollout of $m = 60$ steps ($m = 50$ for `Iso`) using a transformer input window of $k = 30$ steps ($k = 25$ for `Iso`). We first only optimize the encoder and decoder to obtain a reasonably stable latent space, and then the training rollout is linearly increased step by step as proposed by Han et al. (2021). We start increasing the rollout at epoch 300 (40 for `Iso`) until the full sequence length is reached at epoch 1200 (160 for `Iso`). Each model is trained with an MSE loss over the full sequence (adjusted to the current rollout length). On `Inc` and `Tra` these transformer-based models were trained for 5000 epochs, and on `Iso` for 200 epochs.

We do not train the decoder to recover values inside the cylinder area for `Inc` and `Tra`, by applying a binary masking (also see App. A.1 for details) before the training loss computation. Note that this masking is not suitable for autoregressive approaches in the input space, as the masking can cause a distribution shift via unexpected values in the masked area during inference, leading to instabilities. The pure reconstruction, i.e. the first step of the sequence that is not processed by the latent processor, receives a relative weight of $1.0$, and all steps of the rollout *jointly* receive a weight of $1.0$ as well, to ensure that the model balances reconstruction and prediction quality. For $TF_{VAE}$, an additional regularization via a Kullback–Leibler divergence on the latent space with a relative weight of $0.1$ is used. As detailed by Kingma & Welling (2014), for a given mean $l_m^i$ and log variance $l_v^i$ of each latent variable $l^i$ with $i \in 0, 1, \ldots, L$, the regularization $\mathcal{L}_{KL}$ is computed as

$$\mathcal{L}_{KL} = -0.5 * \frac{1}{L} * \sum_{i=0}^{L} 1 + l_v^i - {l_m^i}^2 - e^{l_v^i}.$$

## B.6 TRAINING AND INFERENCE PERFORMANCE

All model architectures were trained, evaluated, and benchmarked on a server with an NVIDIA RTX A5000 GPU with 24GB of video memory and an Intel Xeon Gold 6242R CPU with 20 cores at 3.1 GHz. A performance overview across models can be found in Tab. 1. The training speed in the central columns indicates how many hours are approximately required to fully train a single model according to the training epochs and batch size given further left. For each architecture, we train 3 models (2 for `Iso`) based on randomly seeded runs for the evaluations in the main paper.

Table 1: Overview of training and inference performance for different model architectures.

| Architecture | Training Epochs `Inc`/`Tra`/`Iso` | Batch Size | Training Speed `Inc` [h] | Training Speed `Tra` [h] | Training Speed `Iso` [h] | Inference Speed without I/O [s] | Inference Speed with I/O [s] |
|---|---|---|---|---|---|---|---|
| $cDDPM_{R20}$ | 3100 / 3100 / 100 | 64 | 65-66 | 40-43 | 61-62 | 193.9 | 195.7 |
| $cDDPM_{R100}$ | | | | | | 973.2 | 975.0 |
| U-Net | | 64 | 24-28 | 18-20 | 89-91 | | |
| $U\text{-}Net_{m4}$ | 1000 / 1000 / 100 | 32 | 33-34 | 30-31 | 154-157 | 9.4 | 11.1 |
| $U\text{-}Net_{m8}$ | | 16 | 44-45 | 42-44 | 216-218 | | |
| $U\text{-}Net_{m16}$ | | 8 | 53-54 | 50-52 | 260-263 | | |
| $TF_{MGN}$ | | | 42-43 | 41-43 | 68-70 | 0.8 | 2.8 |
| $TF_{Enc}$ | 5000 / 5000 / 200 | 8 | 36-37 | 36-39 | 66-67 | 0.6 | 2.8 |
| $TF_{VAE}$ | | | 37-38 | 36-37 | 66-69 | 0.7 | 2.7 |
| $ResNet_{dil.}$ | 1000 / 1000 / 100 | 64 | 51-52 | 48-49 | 261-263 | 4.2 | 6.0 |
| $ResNet$ | | | 51-52 | 48-49 | 262-264 | | |
| $FNO_{16}$ | 2000 / 2000 / 200 | 64 | 12-13 | 11-12 | 54-55 | 2.3 | 4.1 |
| $FNO_{32}$ | | | 7-8 | 7-8 | 32-33 | 2.5 | 4.2 |

All model architectures were trained on each data set until their training loss curves were visually fully converged. This means, architectures with more complex learning objectives require more

epochs compared to simpler methods. As such, the transformer variants are highly demanding, as they first need to learn a good latent embedding via the encoder and decoder, and afterwards need to learn the transformer unrolling schedule that is faded in during training time. *cDDPM* which needs to learn a full denoising schedule via random sampling can also require more training iterations compared to direct next-step predictors such as *U-Net*, *ResNet*, or *FNO*. Furthermore, we found the performance of next-step predictors to degrade when trained substantially past the point of visual convergence in early exploration runs. As mentioned above, the default training batch size of $64$ is reduced for architectures that exceed available GPU memory, so the training time comparison is performed at roughly equal memory. Thus, training unrolled *U-Net* models is highly expensive, especially on `Iso`, both via higher memory requirements that result in a lower batch size, but also in the number of computations required for the training rollout.

The right side of Tab. 1 features the inference speed of each method. It is measured on a single example sequence consisting of $T = 1000$ time steps. We report the overall time in seconds that each architecture required during inference for this sequence. Shown in the table is the pure model inference time, as well as the performance including I/O operations and data transfers from CPU to GPU. Note that compared to the performance of *U-Net*, the inference speed slowdown factor of *cDDPM* is closely related to the number of diffusion steps $R$, which corresponds to the number of backbone model evaluations.

## C  ADDITIONAL RESULTS, EVALUATIONS, AND ABLATIONS

In the following, we provide additional evaluations and results. This includes further frequency evaluations in App. C.1, as well as an analysis of the posterior sampling of $TF_{VAE}$ in App. C.2. We also include full numerical results for the accuracy analysis in App. C.3. Furthermore, we perform ablations on the number of diffusion steps in App. C.4, as well as ablations on the stabilization techniques of longer training rollouts in App. C.5 and training noise in App. C.6. Finally, we investigate different loss formulations in App. C.7, analyze the impact of the recently proposed architecture modernizations for U-Nets in App. C.8, and compare to the contemporarily proposed PDE-Refiner method (Lippe et al., 2023) in App. C.9.

### C.1  ADDITIONAL FREQUENCY EVALUATIONS

In addition to the statistical frequency evaluations in the main paper, below we provide further evaluations on different data sets across the models under consideration. For `Inc`$_{low}$, we evaluate the wavenumber of the horizontal motion across a vertical line in the flow (averaged over time), shown on the left of Fig. 11. All models and posterior samples for the given sequence are used in the analysis, and the shaded area corresponds to the 5$^{th}$ to 95$^{th}$ percentile across them. For this relatively simple case, all models, accurately reconstruct low and medium frequencies. Only $FNO_{16}$ exhibits a large variance in the higher frequencies, indicating stability problems within some trained model runs. The difference between the other methods are the high spatial frequencies, where all models overshoot to different degrees, however these differences are not apparent to the human eye in prediction visualizations. $cDDPM_{ncn}$ performs worse compared to *cDDPM* due the lack of error mitigation mechanisms, as there is no noise on the conditioning component during the diffusion processes. The *cDDPM* model with conditioning noise retains the highest spectral accuracy, and stays in line with the simulation reference as shown in the zoomed inset area.

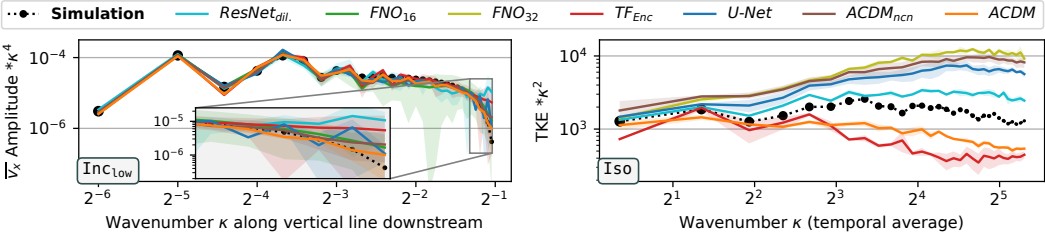

Figure 11: Spatial frequency on a sequence from `Inc`$_{low}$ with $Re = 100$ (left), and spatial frequency via the turbulent kinetic energy (TKE) on a sequence from `Iso` with $z = 300$ (right).

On the right in Fig. 11, we compute a spatial frequency analysis on `Iso` in terms of the turbulent kinetic energy (TKE) averaged across all time steps. *TF_Enc* can only reproduce medium spatial frequencies and lacks in terms of low and high frequencies. *ResNet_dil.*, *FNO_32*, and *U-Net* clearly overshoot in medium and high frequencies, leading to a lacking temporal stability as additional energy is introduced in the prediction. Medium and high spatial frequencies are modeled best by *cDDPM*, but there is still a gap to the reference simulation, meaning *cDDPM* is more dissipative than necessary in the spatial high-frequency regime. This is most likely caused by the strongly under-determined setting of the isotropic turbulence case, where even a numerical solver in 2D would struggle to provide accurate predictions. Note how *cDDPM_ncn* diverges for medium and high frequencies, very similar to *U-Net*, as errors propagate easily over the simulation rollout compared to a conditioning with noise during training.

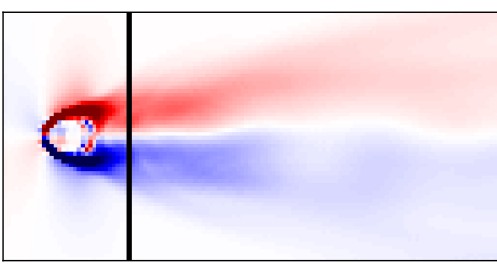 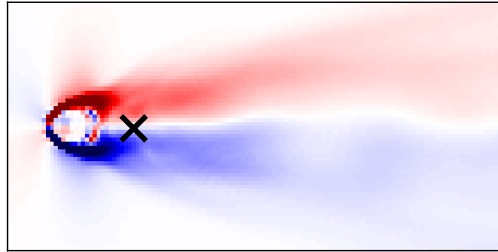

Figure 12: Evaluation line for spatial frequency analysis (top) and evaluation point for temporal frequency analysis (bottom) on `Tra` mean flow (vorticity).

For the frequency evaluations on `Tra_long`, we follow the setup from the main paper. Spatial frequencies are evaluated via the horizontal motion across a vertical line in the flow (averaged over time), and temporal frequencies of the vertical motion are computed at a point probe. Figure 12 illustrates the evaluation locations on top of the mean flow of a sequence from `Tra_long`, where both the point and line probes are positioned one cylinder diameter downstream. Figure 13 contains spatial and temporal frequency analyses on a sequence with $Ma = 0.65$ from `Tra_long` for each model architecture. All trained models and posterior samples are used in this analysis, and the shaded area corresponds to the 5th to 95th percentile across them. Models such as *U-Net*, *FNO_16*, or *cDDPM_ncn* that clearly diverge can be easily identified due to large temporal or spatial spectral errors. *cDDPM* reconstructs both spatial and temporal frequencies most accurately among the compared approaches, closely followed by *ResNet_dil*. Note that the variance for deterministic methods is only calculated over three training runs in this evaluation.

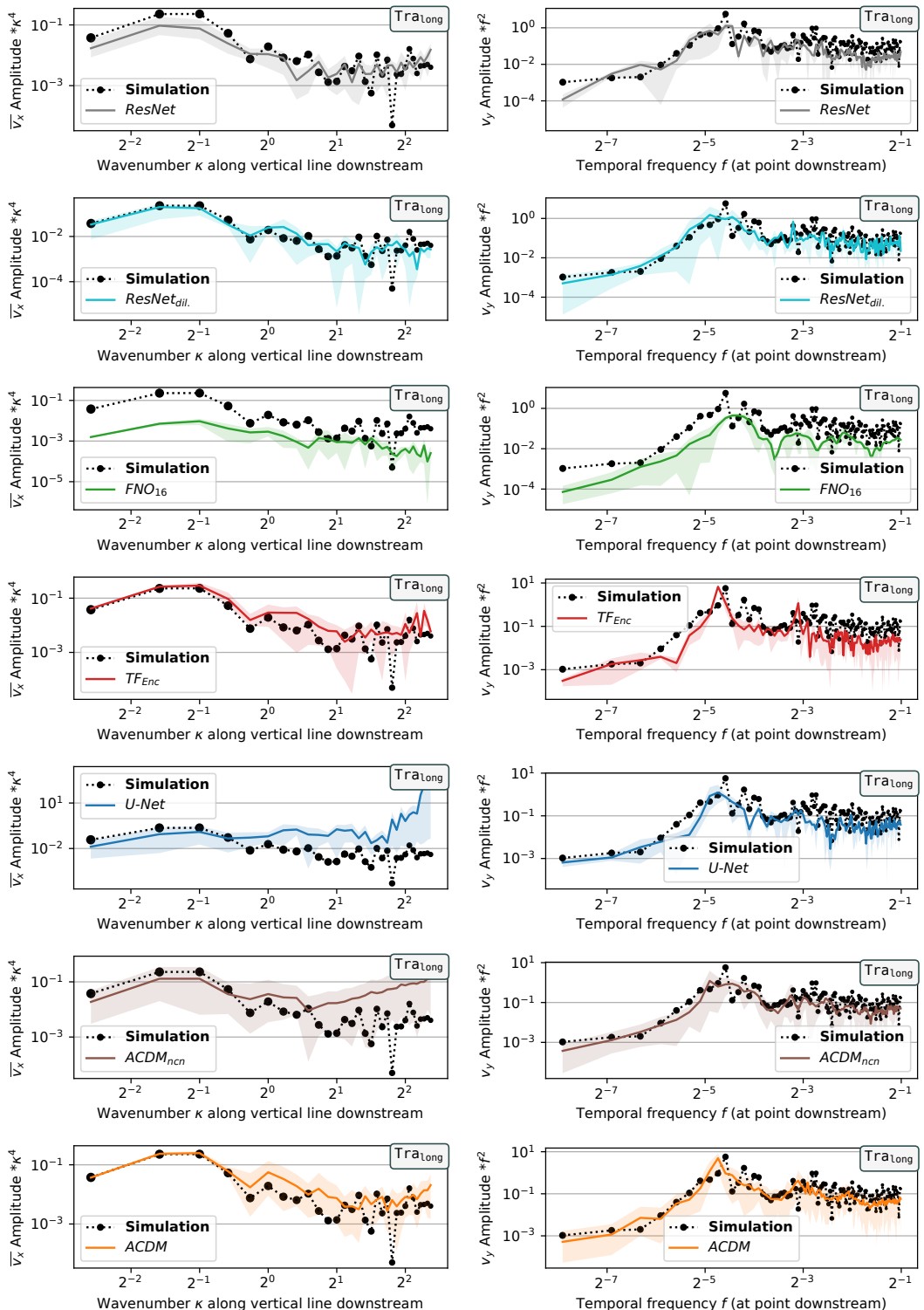

Figure 13: Spatial (left) and temporal (right) frequency analysis for a full sequence from $\texttt{Tra}_{\texttt{long}}$ with $Ma = 0.65$. The shaded area shows the 5$^{\text{th}}$ to 95$^{\text{th}}$ percentile across all trained models and posterior samples for probabilistic models.

## C.2 ADDITIONAL VARIATIONAL AUTOENCODER EVALUATIONS

**Temporal Coherence** Here, we analyze the temporal coherence between individual time steps of the $TF_{VAE}$ model. As an important difference to $cDDPM$, the decoder of $TF_{VAE}$ does not have access to previously generated time steps, as its input is only a sample from the latent space at every step. This leads to temporal artifacts where large differences between consecutive time steps can occur. In Fig. 14 on the left, we display the first three simulation steps of a sequence from Iso, along with the corresponding predictions of $TF_{Enc}$ and $TF_{VAE}$. In addition, the change between the first two predicted steps and $s^0$ is shown on the right. While both methods struggle to reproduce the original vorticity field at $t = 0$, there is a clear difference between both trajectories: the distance between the predictions at $t = 0$ and $t = 1$ is relatively small for $TF_{Enc}$, but big for $TF_{VAE}$. This results in an even visually noticeable jump in the predictions of $TF_{VAE}$.

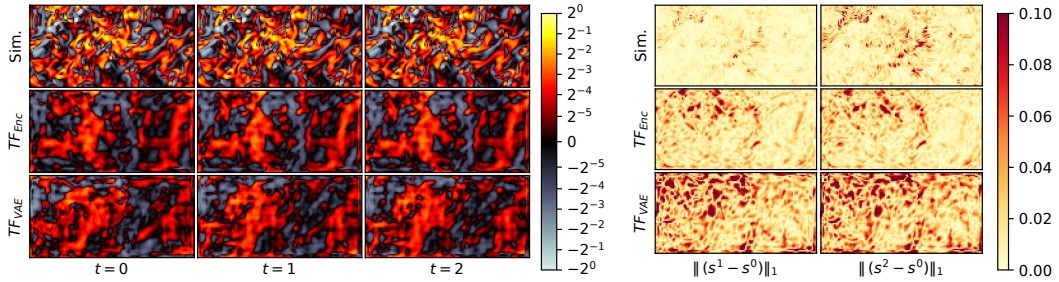

Figure 14: Temporal coherence of $TF_{Enc}$ and $TF_{VAE}$ (left) and difference between reconstruction and first two prediction steps for both models (right) on an example from Iso with $z = 300$.

**Posterior Sampling** Similar to the posterior sampling evaluation for $cDDPM$, we also analyze the posterior samples create by the $TF_{VAE}$ model. We use the same sequence, time step, and zoomed sample area as shown in the main paper. Figure 15 contains three random $TF_{VAE}$ example samples and a spatial standard deviation for all five samples across different time steps on $Tra_{long}$ and Iso. Note that the zoomed samples are displayed via Catmull-Rom spline interpolation for visual clarity in this visualization. Compared to $cDDPM$, all samples are generally highly similar and exhibit very little variance across random model evaluations. While small scale details are varying, the overall structure, e.g., vortex positions for $Tra_{long}$ or areas of high vorticity for Iso, is identical for each sample. In addition, the variance does not substantially increase over time as it would be expected. Due to the inherent data compression, $TF_{VAE}$ introduces some noise artifacts which are especially noticeable in the vorticity prediction for Iso on the bottom right in Fig. 15. Furthermore, it also struggles to create fine details like the shock waves in $Tra_{long}$ as shown at the bottom left.

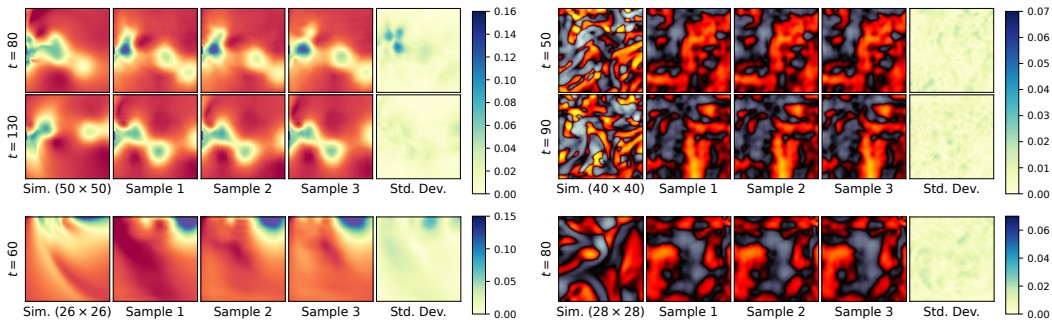

Figure 15: Zoomed $TF_{VAE}$ posterior samples with corresponding standard deviation from $Tra_{long}$ with $Ma = 0.64$ (left, pressure) and Iso with $z = 300$ (right, vorticity) at different time steps $t$.

### C.3 FULL ACCURACY RESULTS AND EVALUATION DETAILS

Table 2: Quantitative comparison across our test sets for different network architectures (best and second best results are highlighted for each data set).

| Method | Inc$_{low}$ MSE $(10^{-4})$ | Inc$_{low}$ LSiM $(10^{-2})$ | Inc$_{high}$ MSE $(10^{-5})$ | Inc$_{high}$ LSiM $(10^{-2})$ | Tra$_{ext}$ MSE $(10^{-3})$ | Tra$_{ext}$ LSiM $(10^{-1})$ | Tra$_{int}$ MSE $(10^{-3})$ | Tra$_{int}$ LSiM $(10^{-1})$ | Iso MSE $(10^{-2})$ | Iso LSiM $(10^{-1})$ |
|---|---|---|---|---|---|---|---|---|---|---|
| *ResNet* | 10±9.1 | 17±7.8 | 16±3.0 | 5.9±1.6 | 2.3±0.9 | 1.4±0.2 | 1.8±1.0 | 1.0±0.3 | 6.7±2.4 | 9.1±2.2 |
| *ResNet$_{dil.}$* | 1.6±1.8 | 7.7±5.5 | 1.5±0.8 | 2.6±0.7 | 1.7±1.0 | 1.2±0.3 | 1.7±1.4 | 1.0±0.5 | 5.7±2.1 | 8.2±2.0 |
| *FNO$_{16}$* | 2.8±3.1 | 8.8±7.1 | 8.9±3.8 | 2.5±1.2 | 4.8±1.2 | 3.4±1.1 | 5.5±2.6 | 2.6±1.1 | $2m$±$6m$ | 15±1.5 |
| *FNO$_{32}$* | 160±50 | 80±5.4 | $1k$±140 | 57±4.9 | 4.9±1.9 | 3.6±0.9 | 6.8±3.4 | 3.1±1.1 | 14±5.3 | 8.9±1.2 |
| *TF$_{MGN}$* | 5.7±4.3 | 13±6.4 | 10±2.9 | 3.5±0.4 | 3.9±1.0 | 1.8±0.3 | 6.3±4.4 | 2.2±0.7 | 8.7±3.8 | 7.0±2.2 |
| *TF$_{Enc}$* | 1.5±1.7 | 6.3±4.2 | 0.6±0.3 | 1.0±0.3 | 3.3±1.2 | 1.8±0.3 | 6.2±4.2 | 2.2±0.7 | 11±5.2 | 7.2±2.1 |
| *TF$_{VAE}$* | 5.4±5.5 | 13±7.2 | 14±19 | 4.1±1.4 | 4.1±0.9 | 2.4±0.2 | 7.2±3.0 | 2.7±0.6 | 11±5.1 | 7.5±2.1 |
| *U-Net* | 1.0±1.1 | 5.8±3.2 | 2.7±0.6 | 2.6±0.6 | 3.1±2.1 | 3.9±2.8 | 2.3±2.0 | 3.3±2.8 | 26±35 | 11±3.9 |
| *cDDPM$_{ncn}$* | 0.9±0.8 | 6.6±2.7 | 5.6±2.6 | 3.6±1.2 | 4.1±1.9 | 1.9±0.6 | 2.8±1.3 | 1.7±0.4 | 18.3±2.5 | 8.9±1.5 |
| *cDDPM* | 1.7±2.2 | 6.9±5.7 | 0.8±0.5 | 1.0±0.3 | 2.3±1.4 | 1.3±0.3 | 2.7±2.1 | 1.3±0.6 | 3.7±0.8 | 3.3±0.7 |

Table 2 contains the full, numerical accuracy values corresponding to Fig. 4. Shown are the mean-squared-error (MSE) and LSiM, a similarity metric for numerical simulation data (Kohl et al., 2020), which is described in more detail below. For both metrics, lower values indicate better reconstruction accuracy, and rollout errors reported, i.e., computed per time step and field, and averaged over the full temporal rollout. Shown are mean and standard deviation over all sequences from each data set, multiple training runs, and multiple random model evaluations: We evaluate two training runs with different random seeds for Iso, and three for Inc and Tra. For the probabilistic methods *TF$_{VAE}$* and *cDDPM*, five random model evaluations are taken into account per trained model. Errors of models that diverge during inference, e.g. *FNO$_{16}$* on Iso or *FNO$_{32}$* on Inc$_{high}$, are displayed with factors of $10^3$ ($k$) or $10^6$ ($m$) in addition to the error scaling indicated in the second table row.

**LSiM Overview** The LSiM metric (Kohl et al., 2020) is a deep learning-based similarity measure for data from numerical simulations. It is designed to more accurately capture the similarity behavior of larger patterns or connected structures that are neglected by the element-wise nature of point-based metrics like MSE. As a simple example, consider a vortex inside a fluid flow that is structurally correctly predicted, but spatially misplaced compared to a reference simulation. While MSE would result in a large distance value, LSiM results in a relatively low distance, especially compared to another vortex that is spatially correctly positioned, but structurally different. LSiM works by embedding both inputs that should be compared in a latent space of a feature extractor network, computing an element-wise difference, and aggregating this difference to a scalar distance value via different operations. The metric is trained on a range of data sets consisting of different transport-based PDE simulations like advection-diffusion equations, Burgers' equation, or the full Navier-Stokes equations. It has been shown to generalize well to flow simulation data outside its training domain like isotropic turbulence.

### C.4 ABLATION ON DIFFUSION STEPS

In the following, we will investigate the *cDDPM* approach with respect to the effect of the number of diffusion steps $R$ in each autoregressive prediction step. For this purpose, we use the adjusted linear variance schedule as discussed in App. B.1, according to the investigated diffusion step $R$. At training and inference time, models always use $R$ diffusion steps. Prediction examples for this evaluation can be found in Figs. 34 and 35 in App. E.

**Accuracy** Tab. 3 contains the accuracy, which is computed as described in App. C.3, of *cDDPM* models with a different number of diffusion steps $R$. While too few diffusion steps on Tra are detrimental, as visible for *cDDPM$_{R10}$*, adding more steps after around $R = 20$ does not improve accuracy. However, on Iso the accuracy of *cDDPM* does continue to improve slightly with increased values of $R$ up to our evaluation limit of $R = 500$. We believe this results from the highly underdetermined

Table 3: Accuracy ablation for different diffusion steps $R$.

| Method | $R$ | $\mathtt{Tra_{ext}}$ MSE $(10^{-3})$ | LSiM $(10^{-1})$ | $\mathtt{Tra_{int}}$ MSE $(10^{-3})$ | LSiM $(10^{-1})$ | $\mathtt{Iso}$ MSE $(10^{-2})$ | LSiM $(10^{-1})$ |
|--------|-----|------|------|------|------|------|------|
| *cDDPM* | 10 | 3.8±1.4 | 1.8±0.3 | 6.2±2.5 | 2.1±0.6 | 15.1±7.4 | 6.6±1.4 |
| *cDDPM* | 15 | 2.5±1.5 | 1.4±0.3 | 2.7±2.0 | 1.4±0.5 | 4.8±1.6 | 4.3±1.0 |
| *cDDPM* | 20 | 2.3±1.4 | 1.3±0.3 | 2.7±2.1 | 1.3±0.6 | 4.5±1.3 | 4.1±0.8 |
| *cDDPM* | 30 | 2.5±1.9 | 1.4±0.4 | 2.7±2.3 | 1.3±0.6 | 4.8±1.9 | 4.1±0.9 |
| *cDDPM* | 50 | 2.3±1.4 | 1.3±0.3 | 2.4±2.1 | 1.3±0.6 | 3.4±0.9 | 3.4±0.7 |
| *cDDPM* | 100 | 2.3±1.3 | 1.3±0.3 | 3.1±2.7 | 1.4±0.6 | 3.7±0.8 | 3.3±0.7 |
| *cDDPM* | 500 | 2.5±1.5 | 1.4±0.4 | 3.1±2.5 | 1.4±0.6 | 3.5±0.9 | 3.2±0.7 |

setting of the $\mathtt{Iso}$ experiment. Note that there is a relatively sharp boundary between too few and a sufficient number of steps; in our experiments $15 - 20$ steps on $\mathtt{Tra}$ and $50 - 100$ steps on $\mathtt{Iso}$.

**Temporal Stability** In Fig. 16, we evaluate the temporal stability via the magnitude of the rate of change of $s$, as detailed in the main paper. Here, different behavior for the ablation models with respect to the number of diffusion steps emerges on $\mathtt{Tra_{long}}$ and $\mathtt{Iso}$. For the former, too little steps, i.e., for *cDDPM_{R10}*, result in unwanted, high-frequency temporal spikes that are also visible as slightly noisy predictions. For $15 - 20$ diffusion steps, these issues vanished, and adding further iterations does not substantially improve temporal stability. Only a slightly higher rate of change can be observed for *cDDPM_{R50}* and *cDDPM_{R500}*.

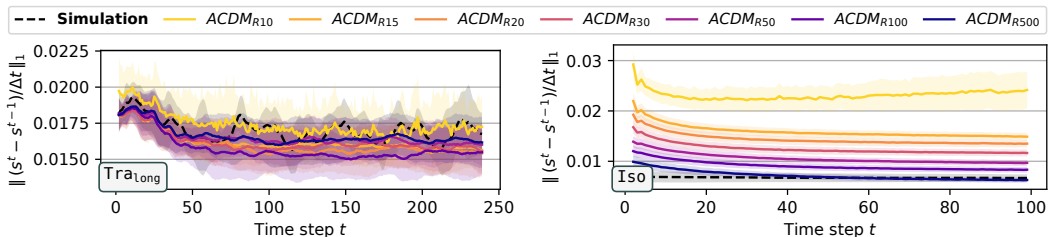

Figure 16: Temporal stability evaluation via error to previous time step for different diffusion steps $R$ on $\mathtt{Tra_{long}}$ (left) and $\mathtt{Iso}$ (right).

On $\mathtt{Iso}$, a tradeoff between prediction accuracy and sampling speed occurs. Even though there are some minor temporal inconsistencies in the first few time steps for very low $R$, all variants result in a stable prediction. However, the magnitude of the rate of change consistently matches the reference trajectory more closely when increasing $R$. This also corresponds to a slight reduction in the overly diffusive prediction behavior for large $R$, both visually and in a spatial spectral analysis via the TKE, as shown in Fig. 17. We believe this tradeoff is caused by the highly underdetermined nature of the $\mathtt{Iso}$ experiment, that leads to a weaker conditioned learning setting, that naturally requires more diffusion steps for high-quality results.

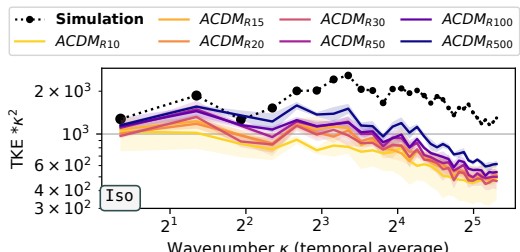

Figure 17: Spatial frequency analysis via the turbulent kinetic energy (TKE) on a sequence from $\mathtt{Iso}$ with $z = 300$ for the diffusion step ablation.

Furthermore, the predictions of *cDDPM_{R100}* exhibit minor visually visible temporal coherence issues on $\mathtt{Iso}$, where small-scale details can flicker quickly. This is caused by highly underdetermined nature of $\mathtt{Iso}$, and can be mitigate by more diffusion steps as well, as *cDDPM_{R500}* reduces this behavior.

**Summary** *cDDPM* works well out-of-the-box with a large number of diffusion steps, but $R$ can be used to balance accuracy and inference performance. Finding the number of diffusion steps for

the best tradeoff is dependent on the data set and learning problem formulation. Generally, setups with stronger conditioning work with few diffusion steps, while less restrictive learning problems can benefit from more diffusion samples. In our experiments, the ideal thresholds emerged relatively clearly.

## C.5 ABLATION ON TRAINING ROLLOUT

Here, we investigate the impact of unrolling the *U-Net* model at training time, via varying the training rollout length $m$. For these models, we use the U-Net architecture as described in App. B.2 with $k = 1$ input steps. However, gradients are propagated through multiple state predictions during training, and corresponding MSE loss over all predicted steps is applied. Prediction examples can be found in Figs. 36 and 37 in App. E.

Table 4: Accuracy ablation for different training rollout lengths $m$ and pre-training (Pre.).

| | | | $\texttt{Tra}_{\texttt{ext}}$ | | $\texttt{Tra}_{\texttt{int}}$ | | $\texttt{Iso}$ | |
|---|---|---|---|---|---|---|---|---|
| **Method** | $m$ | Pre. | MSE $(10^{-3})$ | LSiM $(10^{-1})$ | MSE $(10^{-3})$ | LSiM $(10^{-1})$ | MSE $(10^{-2})$ | LSiM $(10^{-1})$ |
| *U-Net* | 2 | no | 3.1±2.1 | 3.9±2.8 | 2.3±2.0 | 3.3±2.8 | 25.8±35 | 11.3±3.9 |
| *U-Net* | 4 | no | 1.6±1.0 | 1.4±0.8 | 1.1±1.0 | 0.9±0.4 | 3.7±0.8 | 2.8±0.5 |
| *U-Net* | 8 | no | 1.6±0.7 | 1.1±0.2 | 1.5±1.5 | 1.0±0.5 | 4.5±2.8 | 2.4±0.5 |
| *U-Net* | 16 | no | 2.2±1.1 | 1.3±0.3 | 2.4±1.3 | 1.3±0.5 | 13.0±11 | 3.8±1.5 |
| *U-Net* | 4 | yes | — | — | — | — | 5.7±2.6 | 3.6±0.8 |
| *U-Net* | 8 | yes | — | — | — | — | 2.6±0.6 | 2.3±0.5 |
| *U-Net* | 16 | yes | — | — | — | — | 2.9±1.4 | 2.3±0.5 |

**Accuracy**   Table 4 shows models trained with different rollout lengths, and also includes the performance of *U-Net* with $m = 2$ for reference. For the transonic flow, $m = 4$ is already sufficient to substantially improve the accuracy compared to *U-Net* for the relatively short rollout of $T = 60$ steps during inference for $\texttt{Tra}_{\texttt{ext}}$ and $\texttt{Tra}_{\texttt{int}}$. Increasing the training rollout further does not lead to additional improvements and only slightly changes the accuracy. However, note that there is still a substantial difference between the temporal stability of *U-Net$_{m4}$* compared to *U-Net$_{m8}$* or *U-Net$_{m16}$* for cases with a longer inference rollout as analyzed below.

On $\texttt{Iso}$, the behavior of *U-Net* models with longer training rollout is clearly different as models with $m > 4$ substantially degrade compared to $m = 4$. The main reason for this behavior is that gradients from longer rollouts can be less useful for complex data when predictions strongly diverge from the ground truth in early training stages. Thus, we also considered variants, with $m > 2$ that are finetuned from an initialization of a trained basic *U-Net*, denoted by e.g., *U-Net$_{m4,Pre}$*. With this pre-training the previous behavior emerges, and *U-Net$_{m8,Pre}$* even clearly improves upon *U-Net$_{m4}$*.

**Temporal Stability**   In Fig. 18, we evaluate the temporal stability via the magnitude of the rate of change of $s$, as detailed in the main paper. On $\texttt{Tra}_{\texttt{long}}$ all models perform similar until about $t = 50$ where *U-Net* deteriorates. *U-Net$_{m4}$* also exhibits similar signs of deterioration around $t = 130$ during the rollout. Only *U-Net$_{m8}$* and *U-Net$_{m16}$* are fully stable across the entire rollout of $T = 240$ steps. On $\texttt{Iso}$, *U-Net$_{m8}$* achieves comparable stability to *cDDPM*, with an almost constant rate of change for the entire rollout. Models with shorter rollouts, i.e., *U-Net* and *U-Net$_{m4}$* deteriorate after an initial phase, and longer rollouts prevent effective training for *U-Net$_{m16}$* as explained above. The variants with additional pre-training are also included: *U-Net$_{m4,Pre}$* does not substantially improve upon *U-Net$_{m4}$*, and *U-Net$_{m8,Pre}$* performs very well, similar to *U-Net$_{m8}$*. Only for the longer rollouts in the *U-Net$_{m16,Pre}$* model pre-training clearly helps, such that *U-Net$_{m16,Pre}$* is also fully stable.

**Summary**   Compared to *cDDPM*, the variants of *U-Net* with longer training rollouts can achieve similar or slightly higher accuracy and a equivalent temporal stability at a faster inference speed. However, this method requires additional computational resources during training, both in terms of memory over the rollout as well as training time. For example, *U-Net$_{m16,Pre}$* on $\texttt{Iso}$ increases the required training time ($90h$ of pre-training + $260h$ of refinement) by a factor of more than $5.6\times$ at

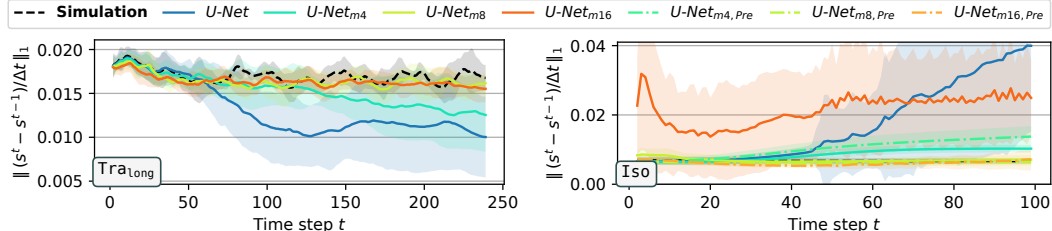

Figure 18: Temporal stability evaluation via error to previous time step for different training rollout lengths $m$ on $\texttt{Tra}_{\texttt{long}}$ (left) and $\texttt{Iso}$ (right).

equal epochs and memory compared to *cDDPM_{R100}* as shown in Tab. 1. Naturally, longer training rollouts do not provide *U-Net* with the ability for posterior sampling.

## C.6    ABLATION ON TRAINING NOISE

We investigate the usage of training noise (Sanchez-Gonzalez et al., 2020) to stabilize predictions, as it features interesting connections to our method. Instead of generating predictions from noise to achieve temporal stability, this method relies on the addition of noise to the training inputs, to simulate error accumulation during training. In this way, the model adapts to disturbances during training, such that the data shift is reduced once errors inevitably accumulate during the inference rollout, leading to increased temporal stability. We test this approach on *U-Net* and on *cDDPM_{ncn}*. The latter evaluation serves as an example to understand if the lost tolerance for error accumulation in *cDDPM_{ncn}*, the setup without conditioning noise, can be replaced with training noise. This *cDDPM_{ncn}* version is not intended as a practical architecture as it inherits the drawbacks of both methods, the inference cost from diffusion models, and the overhead and additional hyperparameters from added training noise. For these ablations we use the same *U-Net* and *cDDPM_{ncn}* models as described in Appendices B.1 and B.2. We only add normally distributed noise with standard deviation $n$ to every model input during training, while leaving the prediction target untouched. At inference time, the models operate identically to their counterparts without training noise. In the following, *U-Net* or *cDDPM_{ncn}* models trained with training noise of e.g., $n = 10^{-1}$, are denoted by *U-Net_{n1e-1}* or *cDDPM_{ncn,n1e-1}* respectively. Prediction examples can be found in Figs. 38 and 39 in App. E.

Table 5: Accuracy ablation for different training noise standard deviations $n$.

| Method | $n$ | $\texttt{Tra}_{\texttt{ext}}$ MSE $(10^{-3})$ | $\texttt{Tra}_{\texttt{ext}}$ LSiM $(10^{-1})$ | $\texttt{Tra}_{\texttt{int}}$ MSE $(10^{-3})$ | $\texttt{Tra}_{\texttt{int}}$ LSiM $(10^{-1})$ | $\texttt{Iso}$ MSE $(10^{-2})$ | $\texttt{Iso}$ LSiM $(10^{-1})$ |
|---|---|---|---|---|---|---|---|
| *U-Net* | — | 3.1±2.1 | 3.9±2.8 | 2.3±2.0 | 3.3±2.8 | 25.8±35 | 11.3±3.9 |
| *U-Net* | 1e−4 | 2.7±1.8 | 3.9±2.1 | 1.9±0.8 | 2.4±2.1 | 16.0±22 | 9.6±3.0 |
| *U-Net* | 1e−3 | 5.6±2.2 | 3.3±2.5 | 3.5±1.6 | 3.0±2.2 | 36.4±39 | 12.9±2.2 |
| *U-Net* | 1e−2 | 1.4±0.8 | 1.1±0.3 | 1.8±1.1 | 1.0±0.4 | 3.1±0.9 | 4.5±2.5 |
| *U-Net* | 1e−1 | 1.8±0.8 | 1.2±0.2 | 2.2±2.0 | 1.2±0.6 | 3.2±0.5 | 2.9±0.6 |
| *U-Net* | 1e0 | 4.0±1.5 | 1.8±0.3 | 11.4±6.3 | 2.9±1.3 | 16.2±7.8 | 7.5±2.7 |
| *cDDPM_{ncn}* | — | 4.1±1.9 | 1.9±0.6 | 2.8±1.3 | 1.7±0.4 | 18.3±2.5 | 8.9±1.5 |
| *cDDPM_{ncn}* | 1e−4 | 3.8±1.5 | 2.0±0.3 | 4.3±2.3 | 1.7±0.4 | 14.2±1.7 | 8.1±1.2 |
| *cDDPM_{ncn}* | 1e−3 | 3.6±1.4 | 2.2±0.3 | 3.9±2.3 | 1.8±0.4 | 11.1±3.8 | 8.5±1.6 |
| *cDDPM_{ncn}* | 1e−2 | 3.6±1.6 | 1.7±0.4 | 2.6±2.3 | 1.3±0.5 | 26.7±25 | 12.2±2.8 |
| *cDDPM_{ncn}* | 1e−1 | 3.6±1.9 | 1.5±0.4 | 2.5±2.2 | 1.2±0.6 | 2.8±0.6 | 4.0±2.2 |
| *cDDPM_{ncn}* | 1e0 | 4.2±1.7 | 1.8±0.4 | 6.2±2.8 | 2.0±0.6 | 11.1±1.4 | 6.2±0.9 |

**Accuracy**    The accuracy of *U-Net* and *cDDPM_{ncn}* setups with training noise using different standard deviations $n$ is analyzed in Tab. 5. On $\texttt{Tra}$, the accuracy trend is not fully consistent. Small values of $n$ such as $10^{-4}$ and $10^{-3}$ occasionally even reduce the final performance, but training noise with a well-tuned standard deviation between $10^{-2}$ and $10^{-1}$ does increase accuracy. Choosing very large standard deviations corrupts the training data too much, and reduces accuracy again as expected.

The results on the isotropic turbulence experiment show a similar behavior for *U-Net* as well as *cDDPM$_{ncn}$*.

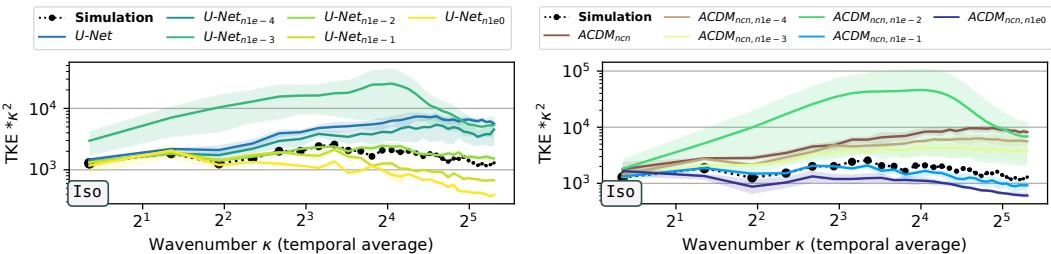

Figure 19: Temporal stability evaluation for different training noise standard deviations $n$ of *U-Net* (top) and *cDDPM$_{ncn}$* (bottom) on `Tra`$_{\text{long}}$ (left) and `Iso` (right).

**Temporal Stability**  In Fig. 19, we evaluate the temporal stability of models with training noise via the magnitude of the rate of change of $s$, as detailed in the main paper. On `Tra`$_{\text{long}}$ both architectures *U-Net* and *cDDPM$_{ncn}$* behave similarly: while training noise with a standard deviation $n$ that is too low does not improve the stability and occasionally even deteriorates it, finding a suitable magnitude is key for stable inference rollouts. In both cases, values of $n$ between $10^{-2}$ and $10^{-1}$ produce the best results. Increasing the noise further has detrimental effects, as for example slight overshooting and high-frequency fluctuations occur for *cDDPM$_{ncn,n1e0}$* or predictions can diverge early from the simulation for *U-Net$_{n1e0}$*. On `Iso`, a similar stabilizing effect from training noise can be observed, given the noise magnitude is tuned sufficiently: While lower standard deviations barely alter the time point $t = 40$, where predictions diverge from the reference simulation, too much training noise already causes major problems at the very beginning of the prediction.

This behavior can also be observed on a spatial spectral analysis via the TKE in Fig. 20, where the training noise can balance predictions between under- and overshooting. For both *U-Net* and *cDDPM$_{ncn}$*, training noise with a suitable magnitude can result in a comparable temporal stability to *cDDPM*, that includes noise on the conditioning.

Figure 20: Spatial frequency analysis via the turbulent kinetic energy (TKE) on a sequence from `Iso` with $z = 300$ for the training noise ablations on *U-Net* (left) and *cDDPM$_{ncn}$* (right).

**Summary**  Training *U-Net* with training noise can achieve similar or slightly higher accuracy and a competitive temporal stability compared to *cDDPM*. While this method exhibits faster inference speeds, it does rely on the additional noise variance hyperparameter, that can even reduce performance if not tuned well. Furthermore, training noise does not provide deterministic models with the ability

for posterior sampling. Interestingly, the lost error tolerance of the *cDDPM_ncn* architecture without conditioning noise, can be mostly restored with training noise of suitable magnitude.

## C.7 ABLATION ON TRAINING WITH AN LSIM LOSS

In this section, we investigate usage of the LSiM metric (Kohl et al., 2020) as an additional loss term, similar to perceptual losses in the computer vision domain (Dosovitskiy & Brox, 2016; Johnson et al., 2016). This means, in addition to training *U-Net* with an MSE loss as above, the differentiable learned LSiM metric model is also used during back-propagation. Given a predicted state $s^t$ and the corresponding ground truth state $\hat{s}^t$, we evaluate the training loss as

$$\mathcal{L}_{MSE+LSiM} = \left(s^t - \hat{s}^t\right)^2 + \lambda * \text{LSiM}(s^t, \hat{s}^t)$$

while leaving the inference of the models untouched. To use LSiM, each field from both states is individually normalized to $[0, 255]$. The resulting loss values are aggregated with an average operation across fields. Fields containing the scalar simulation parameters are not evaluated with this metric. In the following, the impact of $\lambda$, the weight that controls the influence of the LSiM loss, is investigated. *U-Net* models trained with e.g., $\lambda = 10^{-1}$, are denoted by *U-Net*$_{\lambda 1e-1}$.

Table 6: Accuracy ablation for training with LSiM losses of different strengths $\lambda$.

| Method | $\lambda$ | Tra$_{ext}$ MSE $(10^{-3})$ | Tra$_{ext}$ LSiM $(10^{-1})$ | Tra$_{int}$ MSE $(10^{-3})$ | Tra$_{int}$ LSiM $(10^{-1})$ | Iso MSE $(10^{-2})$ | Iso LSiM $(10^{-1})$ |
|---|---|---|---|---|---|---|---|
| *U-Net* | — | 3.1±2.1 | 3.9±2.8 | 2.3±2.0 | 3.3±2.8 | 25.8±35 | 11.3±3.9 |
| *U-Net* | 1e–5 | 4.2±2.9 | 4.5±3.0 | 2.6±2.2 | 2.1±2.0 | 67.4±75.7 | 12.4±3.8 |
| *U-Net* | 1e–4 | 2.3±1.2 | 3.7±2.6 | 1.6±1.4 | 2.0±1.8 | 12.3±9.3 | 11.8±2.5 |
| *U-Net* | 1e–3 | 2.9±1.9 | 1.7±0.8 | 2.2±2.3 | 1.5±0.9 | 6.3±3.1 | 9.4±2.8 |
| *U-Net* | 1e–2 | 4.5±1.3 | 3.5±1.1 | 3.0±2.3 | 1.8±0.9 | 0.1b±0.2b | 15.3±1.2 |
| *U-Net* | 1e–1 | 5.8±1.8 | 3.0±0.8 | 5.2±1.9 | 2.3±0.6 | 12b±29b | 15.0±1.0 |
| *U-Net* | 1e0 | 6.8±1.5 | 4.8±1.1 | 6.6±3.0 | 2.4±0.7 | 17b±552b | 14.9±1.0 |

**Accuracy** In terms of accuracy, adding very small amounts of the LSiM term with $\lambda = 10^{-5}$ to the MSE loss does decrease performance, most likely due to suboptimal gradient signals through the additional steps during back-propagation, as shown in Tab. 6. Similarly, adding too much, such that it predominantly influences the overall loss causes problems. Especially on Iso, this causes models to aggressively diverge after $30 - 40$ prediction steps, leading to errors in the range of $10^9$ ($b$) in Tab. 6. As expected, choosing a suitable loss magnitude around $\lambda = 10^{-3}$ substantially reduces errors in terms of LSiM across test sets. However, the added loss term does also improve performance in terms of MSE, as similarly observed in the image domain (Dosovitskiy & Brox, 2016; Johnson et al., 2016).

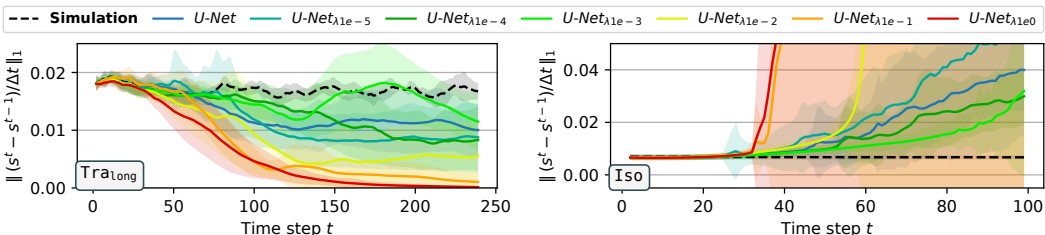

Figure 21: Temporal stability evaluation of *U-Net* for training with LSiM losses of different strengths $\lambda$ on Tra$_{long}$ (left) and Iso (right).

**Temporal Stability** In line with the accuracy results, *U-Net*$_{\lambda 1e-3}$ exhibits improved temporal stability compared to *U-Net* as displayed in Fig. 21. Choosing unsuitable $\lambda$ causes models to diverge earlier from the reference trajectory when evaluating the difference between predictions steps, for both Tra$_{long}$ and Iso. When analyzing the frequency behavior of the models trained with LSiM in Fig. 22 the results are similar: Improved performance across the frequency band can be observed for

*U-Net*$_{\lambda 1e\text{-}3}$, while smaller values of $\lambda$ are less potent and can be detrimental or only slightly beneficial compared to *U-Net*.

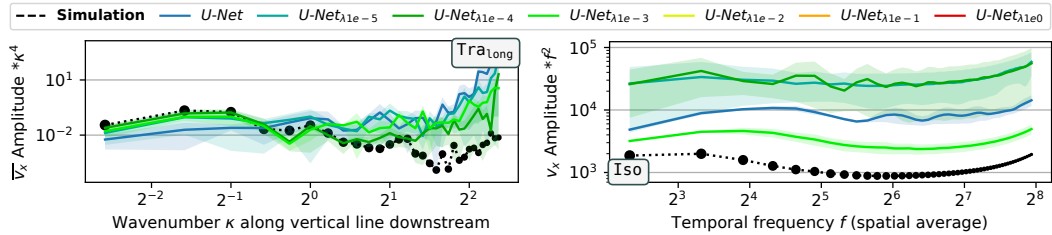

Figure 22: Spatial frequency along a vertical line downstream on $\texttt{Tra}_{long}$ (left) and temporal frequency analysis on a sequence from $\texttt{Iso}$ with $z = 300$ (right) for the LSiM loss ablation models.

**Summary**  Training *U-Net* with LSiM as an additional loss term, can increase accuracy, temporal stability, and frequency behavior across evaluations. However, the resulting models are neither competitive compared to other stabilization techniques discussed above, such as training rollouts or training noise, nor to the proposed diffusion architecture.

## C.8  ABLATION ON U-NET MODERNIZATIONS

As described in App. B.2, our U-Net implementation follows established diffusion model architectures, that contain a range of modernizations compared to the original approach proposed by Ronneberger et al. (2015). Here, we compare to a more traditional U-Net architecture which is known to work well for fluid problems. We adapted the DFP model implementation[5] of Thuerey et al. (2020) for our settings. The architecture features:

- batch normalization instead group normalization, and no attention layers in the blocks,
- six downsampling blocks consisting of strided convolutions and leaky ReLU layers,
- six upsampling blocks consisting of convolution, bilinear upsamling, and ReLU layers,
- six feature map levels with spatial sizes of $64 \times 32$, $32 \times 16$, $16 \times 8$, $8 \times 4$, $4 \times 2$, and $2 \times 1$,
- an increasing number of channels for deeper features, i.e., 72, 72, 144, 288, 288, and 288.

It is trained as a direct one-step predictor (*DFP*) in the same way as described in App. B.2, as well as employing it in the diffusion setup as a backbone architecture (*DFP$_{ACDM}$*). In both cases, we keep all other hyperparameters identical with the corresponding baseline architecture.

Table 7: Accuracy of the "modern" U-Net architecture compared to *DFP*.

| | $\texttt{Tra}_{ext}$ | | $\texttt{Tra}_{int}$ | | $\texttt{Iso}$ | |
|---|---|---|---|---|---|---|
| **Method** | MSE $(10^{-3})$ | LSiM $(10^{-1})$ | MSE $(10^{-3})$ | LSiM $(10^{-1})$ | MSE $(10^{-2})$ | LSiM $(10^{-1})$ |
| *U-Net* | 3.1±2.1 | 3.9±2.8 | 2.3±2.0 | 3.3±2.8 | 25.8±35 | 11.3±3.9 |
| *DFP* | 4.5±1.3 | 3.9±0.7 | 4.8±2.1 | 3.6±1.7 | 5.1±1.3 | 5.1±2.0 |
| *cDDPM* | 2.3±1.4 | 1.3±0.3 | 2.7±2.1 | 1.3±0.6 | 3.7±0.8 | 3.3±0.7 |
| *DFP$_{ACDM}$* | *NaN* | *NaN* | *NaN* | *NaN* | *NaN* | *NaN* |

**Accuracy**  Table 7 shows a comparison of both architectures compared to *U-Net* and *cDDPM* on our more challenging data sets $\texttt{Tra}$ and $\texttt{Iso}$. Training *DFP$_{ACDM}$* as a diffusion backbone (with additional time embeddings for the diffusion step $r$ as discussed in App. B.1) failed to generalize beyond the first few prediction timesteps across test sets in our experiments. This highlights the general usefulness of the recently introduced modernizations to the U-Net architecture. There is a noticeable drop in accuracy on $\texttt{Tra}$ for *DFP* compared to *U-Net*, but it performs clearly better than *U-Net* on $\texttt{Iso}$, however still lacking compared to *cDDPM*. This unexpected trend in accuracy is mainly caused by the different rollout behavior of these architectures discussed in the following.

---

[5] https://github.com/thunil/Deep-Flow-Prediction

**Temporal Stability**    Figure 23 shows temporal stability evaluations of the variants on $\texttt{Tra}_\texttt{long}$ and $\texttt{Iso}$, that illustrate the different rollout behavior of *DFP* compared to *U-Net* depending on the data set. On $\texttt{Tra}_\texttt{long}$ on the left, *DFP* diverges earlier and more substantially compared to *U-Net*, when measured via the difference to the previously predicted time step. However, the rollout behavior is different on $\texttt{Iso}$, as illustrated via the Pearson correlation coefficient to the ground truth trajectory on the right in Fig. 23. The simpler *DFP* model decorrelates more quickly for the first 50 steps, while keeping a relatively constant decorrelation rate. *U-Net* is initially more in line with the reference, however it sharply decreases after about 50 steps, meaning errors accumulate more quickly after an initial phase of higher stability.

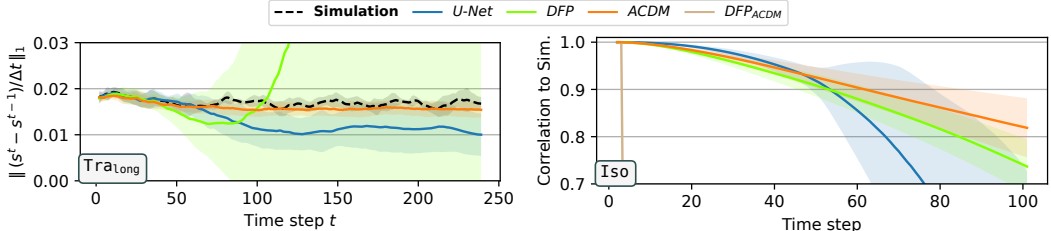

Figure 23: Temporal stability evaluation via error to previous time step on $\texttt{Tra}_\texttt{long}$ (left) and the correlation to the ground truth on $\texttt{Iso}$ (right) for the U-Net modernization ablation.

**Summary**    We found the recently proposed architecture modernizations to U-Nets to be an important factor, when employing them as backbones in a diffusion-based setup. For some direct prediction cases, the modernizations can delay diverging behavior due to unrolling during inference to some degree. On other data, using no modernizations can be beneficial for longer rollouts in direct prediction setting, but this comes at the costs of less initial accuracy, and lacking capacities as a diffusion backbone.

### C.9    COMPARISON TO PDE-REFINER

Lippe et al. (2023) recently proposed a multi-step refinement process to improve the stability of learned PDE predictions called PDE-Refiner. Their approach relies on starting from the predictions of a trained one-step model, and iteratively refining them by adding noise of decreasing variances and denoising the result. The resulting model is then autoregressively unrolled to form a prediction trajectory, similar to our simulation rollout. This method implies, that only probabilistic refinements are applied to a deterministic initial prediction. To train a model that can predict and refine at the same time, a random step $r \in [0, R]$ in the refinement process is sampled, and the model is trained with a next-step MSE objective if $r = R$ and with a standard denoising objective otherwise[6]. We re-implement this method, closely following the provided pseudocode in their paper, only changing the backbone network to our *U-Net* implementation (see App. B.2) for a fair comparison against our architectures. Lippe et al. (2023) report that models with around $R = 4$ refinement steps perform best, when paired with a custom, exponential noise schedule, parameterized with a minimum noise variance[7] around $\sigma = 10^{-6}$. Nevertheless, we sweep over combinations of $R \in \{2, 4, 8\}$ and $\sigma \in \{10^{-7}, 10^{-6}, 10^{-5}, 10^{-4}, 10^{-3}\}$ here, to ensure ideal values for these hyperparameters in our setting. Due to computational constraints for this large sweep, only one model per combination is trained. Five samples from each model are considered, as above. We denote models trained with e.g., $R = 2$ and $\sigma = 10^{-3}$ by *Refiner$_{R2,\sigma 1e\text{-}3}$*. Prediction examples can be found in Figs. 40 and 41 in App. E.

**Accuracy**    Table 8 evaluates the accuracy of these PDE-Refiner variants compared to *cDDPM* and *U-Net* on our data sets $\texttt{Tra}$ and $\texttt{Iso}$. Overall, the performance of *Refiner* across data sets, number of refinement steps $R$, and noise variances $\sigma$ is highly unpredictable. There is neither a clear accuracy trend over few or many refinement steps, nor high or low noise variance. Furthermore, a high accuracy

---

[6]Compared to Lippe et al. (2023), we switch the notation to $R$ being the first step in the reverse process here, in line with our notation above, which also matches the notation in the original DDPM (Ho et al., 2020).

[7]For brevity, we use $\sigma$ for the minimum noise variance here, Lippe et al. (2023) refer to it as $\sigma_{min}^2$.

Table 8: Accuracy compare to PDE-Refiner for different refinement steps $R$ and noise variances $\sigma$.

| | | | Tra$_{ext}$ | | Tra$_{int}$ | | Iso | |
|---|---|---|---|---|---|---|---|---|
| **Method** | $R$ | $\sigma$ | MSE $(10^{-3})$ | LSiM $(10^{-1})$ | MSE $(10^{-3})$ | LSiM $(10^{-1})$ | MSE $(10^{-2})$ | LSiM $(10^{-1})$ |
| *cDDPM* | — | — | 2.3±1.4 | 1.3±0.3 | 2.7±2.1 | 1.3±0.6 | 3.7±0.8 | 3.3±0.7 |
| *U-Net* | — | — | 3.1±2.1 | 3.9±2.8 | 2.3±2.0 | 3.3±2.8 | 25.8±35 | 11.3±3.9 |
| *Refiner* | 2 | 1e–3 | 3.3±1.3 | 1.4±0.3 | 3.9±1.6 | 1.4±0.3 | 6.1±1.9 | 7.2±1.6 |
| *Refiner* | 2 | 1e–4 | 12.7±2.9 | 4.2±0.5 | 10.1±1.5 | 2.4±0.3 | 0.1m±0.3m | 12.5±5.2 |
| *Refiner* | 2 | 1e–5 | 4.8±1.4 | 2.6±0.3 | 4.0±3.1 | 2.1±0.5 | 3.3e30 | 15.2±0.9 |
| *Refiner* | 2 | 1e–6 | 5.0±1.9 | 2.0±0.3 | 3.6±2.6 | 1.9±0.4 | 0.1m±0.2m | 16.1±1.0 |
| *Refiner* | 2 | 1e–7 | 13.6±9.9 | 6.1±4.0 | 54.6±68.7 | 6.7±5.0 | 22k±13k | 14.9±0.9 |
| *Refiner* | 4 | 1e–3 | 5.3±0.8 | 3.2±0.4 | 6.0±1.2 | 2.6±0.4 | 5.1±1.8 | 4.7±0.8 |
| *Refiner* | 4 | 1e–4 | 3.4±2.0 | 1.9±0.3 | 5.7±2.4 | 1.9±0.5 | 7.0±3.1 | 5.0±1.0 |
| *Refiner* | 4 | 1e–5 | 7.0±1.7 | 2.7±0.4 | 3.1±0.8 | 1.7±0.2 | 4.9±2.0 | 7.6±2.1 |
| *Refiner* | 4 | 1e–6 | 3.5±1.1 | 2.1±0.5 | 8.8±0.9 | 4.3±2.1 | 66.1±38.4 | 11.7±0.7 |
| *Refiner* | 4 | 1e–7 | 5.4±1.0 | 3.1±0.2 | 8.3±2.2 | 2.7±0.2 | 1.9e18 | 14.8±1.0 |
| *Refiner* | 8 | 1e–3 | 7.1±1.5 | 3.5±0.4 | 4.4±1.8 | 2.7±0.4 | 5.5±1.3 | 6.9±1.0 |
| *Refiner* | 8 | 1e–4 | 13.8±2.3 | 5.0±0.5 | 8.6±4.2 | 2.4±0.7 | 5.1±1.3 | 5.9±1.1 |
| *Refiner* | 8 | 1e–5 | 6.3±1.1 | 3.5±0.4 | 6.0±1.8 | 2.4±0.6 | 4.7±0.7 | 5.4±1.2 |
| *Refiner* | 8 | 1e–6 | 3.1±1.3 | 2.2±0.2 | 6.4±2.1 | 2.0±0.4 | 0.1k±0.3k | 6.1±4.3 |
| *Refiner* | 8 | 1e–7 | 4.3±1.4 | 2.1±0.3 | 3.3±1.2 | 1.6±0.3 | 88±70 | 6.2±1.9 |

on Tra is not directly correlated with a high accuracy on Iso either. As *Refiner* essentially improves upon one-step predictions of *U-Net* via additional refinement steps, the results of a direct comparison are interesting: On Tra, while *Refiner* consistently outperforms *U-Net* in terms of LSiM, it just as consistently remains worse in terms of the MSE across hyperparameter combinations. We hypothesize that these results are linked to the fundamentally different spectral behavior of *Refiner* described by Lippe et al. (2023), but further research is required in this direction. On Iso, *Refiner* either improves upon *U-Net* or substantially diverges (marked in grey in Tab. 8), especially for small $\sigma$. Overall, PDE-Refiner is less effective than the stabilization techniques discussed in Appendices C.5 and C.6 in terms of accuracy improvements, and thus consistently falls short with respect to *cDDPM* across the test sets and hyperparameter combinations considered here.

**Temporal Stability** To investigate the temporal stability of *Refiner*, we analyze the difference to the previous time step in Fig. 24. First, it is shown that there are combinations of $R$ and $\sigma$ that substantially improve the rollout stability of *Refiner* compared to *U-Net*, confirming the results from Lippe et al. (2023). However, as observed in terms of accuracy above, there is no consistent trend across hyperparameters and data sets. Especially, finding a suitable minimum noise variance $\sigma$ depends on both, data set and number of refinement steps $R$: While $\sigma = 10^{-6}$ works best on Tra$_{long}$ for $R = 2$, $\sigma = 10^{-7}$ is ideal for $R = 8$. On Iso, $R = 2$ only works with $\sigma = 10^{-3}$, $R = 4$ requires $\sigma = 10^{-4}$, and $R = 8$ is most stable with $\sigma = 10^{-5}$. This unpredictable behavior with respect to important hyperparameters makes PDE-Refiner resource-intensive and difficult to employ in practice. The best *Refiner* variants on Iso, while more stable compared to *U-Net*, are nevertheless showing signs of instabilities around $t = 70$. This means the refinement increases stability, but still falls short compared to the other stabilization techniques discussed in Appendices C.5 and C.6.

**Posterior Sampling** As PDE-Refiner relies on deterministic predictions combined with probabilistic refinements, achieving a broad and diverse posterior distribution is difficult. In Fig. 25, we visualize posterior samples for Tra$_{long}$ from *cDDPM* and *Refiner* for $R \in \{2, 4, 8\}$ and with the ideal $\sigma = 10^{-6}$ reported by Lippe et al. (2023). While *cDDPM* creates a broad range of samples as discussed in more detail in the main paper above, *Refiner*$_{R2,\sigma 1e\text{-}6}$ does not create any visual variance. While additional refinement steps slightly improve the spread across samples, even *Refiner*$_{R8,\sigma 1e\text{-}6}$ can only create minor differences with very similar spatial structures. Note that the *Refiner* models are unable to create the detailed shockwaves below the cylinder that are found in the simulation and the *cDDPM* samples. Similar, unphysical predictions after longer rollouts can be observed across refinement steps and noise variances in Fig. 40 as well. Using the largest $\sigma = 10^{-3}$ we evaluated should theoretically allow *Refiner* to focus on a larger range of frequencies, but also does not substantially improve quality or diversity of posterior samples over Fig. 25.

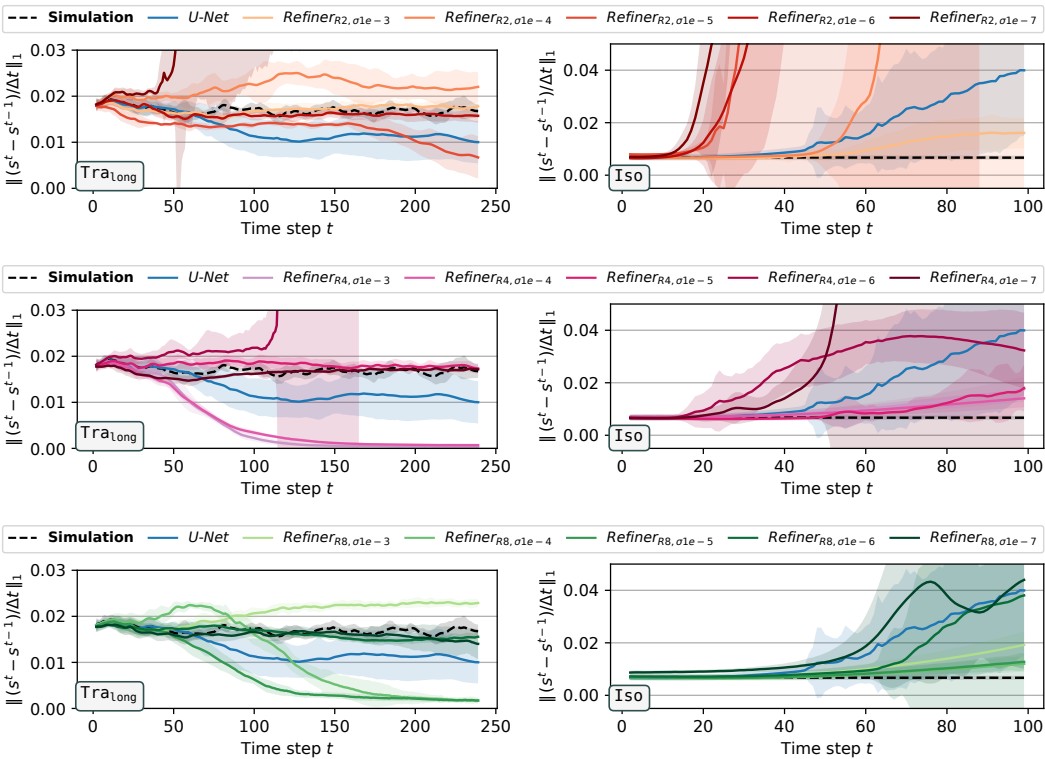

Figure 24: Temporal stability evaluation via error to previous time step on $\mathtt{Tra_{long}}$ (left) and on $\mathtt{Iso}$ (right) for PDE-Refiner with different hyperparameter combinations of refinement steps $R$ and noise variances $\sigma$. The temporally most stable *Refiner* configuration is highly inconsistent, and for a given $R$ depends on the data set and noise variance.

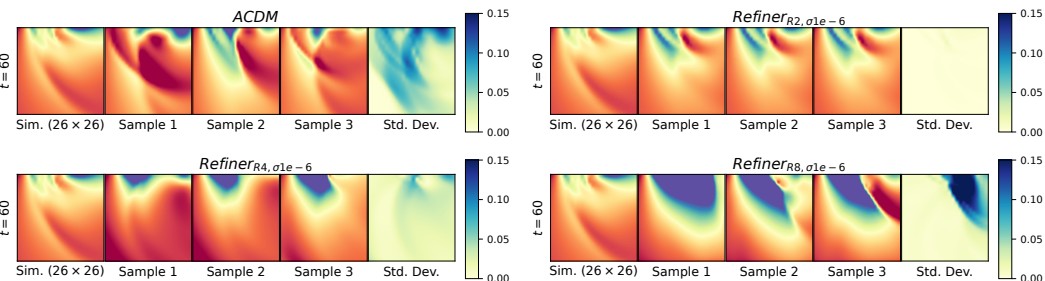

Figure 25: Posterior samples on $\mathtt{Tra_{long}}$ from *cDDPM* (top left) compared to PDE-Refiner (bottom and right). *Refiner* lacks sample diversity and quality compared to *cDDPM* across refinement steps $R$.

**Summary** While the stability benefits of a well-tuned setup with PDE-Refiner compared to a simple one-step prediction with *U-Net* are highly desirable and can be achieved with less inference overhead compared to *cDDPM*, the method has several disadvantages: We found the setup to be very sensitive regarding changes to refinement steps, data set, or noise variance. This means, a large amount of computational resources are required for parameter tuning, which is crucial to obtain good results. Suboptimal combinations of refinement steps and noise variance show substantially degraded performance compared to *U-Net* in our experiments. Furthermore, *Refiner* achieves lower overall accuracy across data sets and has substantial limits in terms of the posterior sampling compared to a more direct application of diffusion models in *cDDPM*.

# D    PREDICTION EXAMPLES

Over the following pages, prediction examples from all analyzed methods in the main paper are displayed. Shown are the different fields contained in an exemplary test sequence from each experiment. Figures 26 and 27 feature the $\texttt{Inc}_{\texttt{var}}$ case, Figs. 28 to 30 contain an example from $\texttt{Tra}_{\texttt{long}}$ with $Ma = 0.64$, and Figs. 31 to 33 display a sequence from $\texttt{Iso}$ with $z = 280$.

Videos of predictions from some example sequences are also provided alongside this submission, as they can visualize several aspects like temporal stability, temporal coherence, and visual quality better than still images. We also include videos of different *cDDPM* posterior samples.

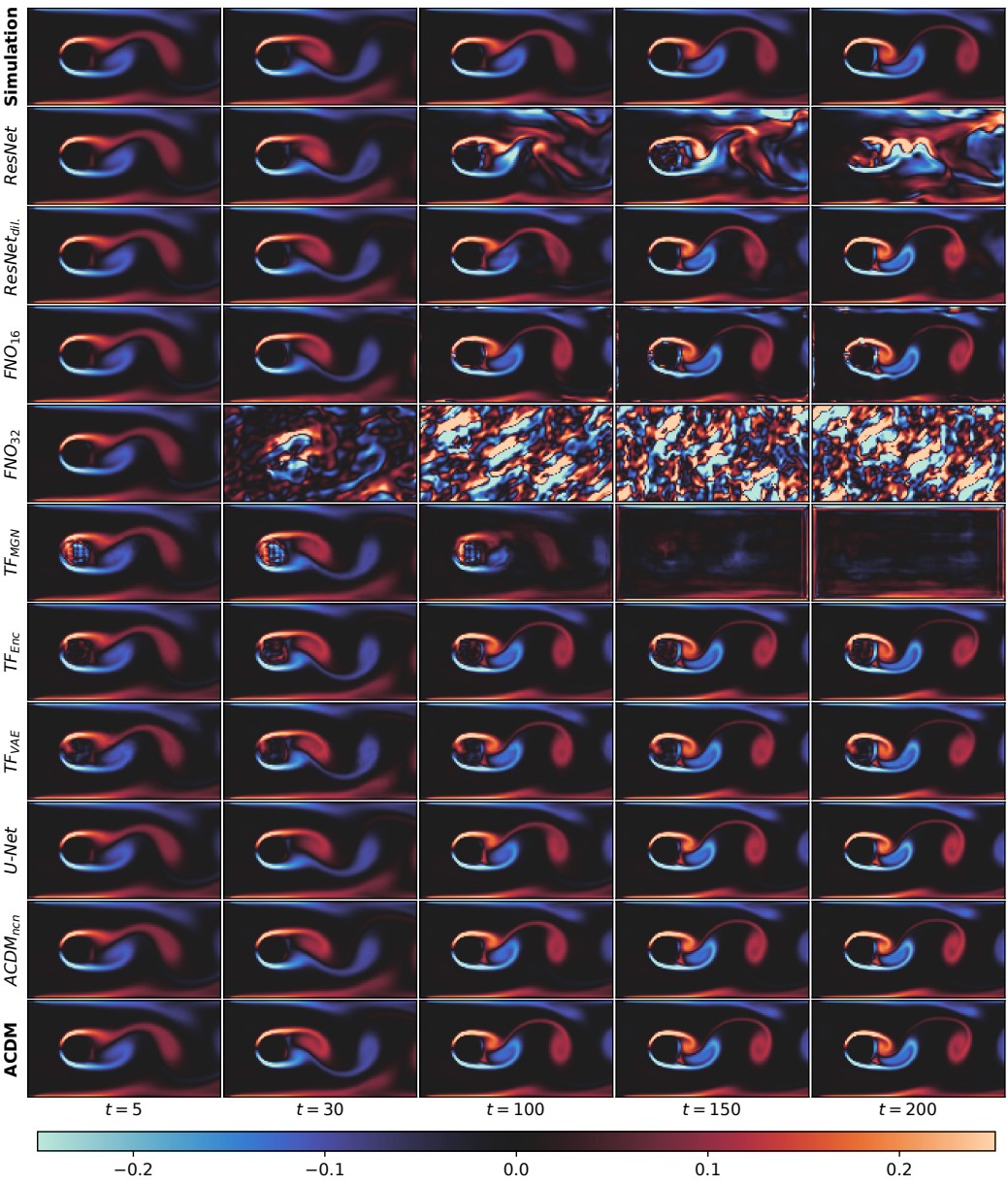

Figure 26: Vorticity predictions for the $\texttt{Inc}_{\texttt{var}}$ sequence.

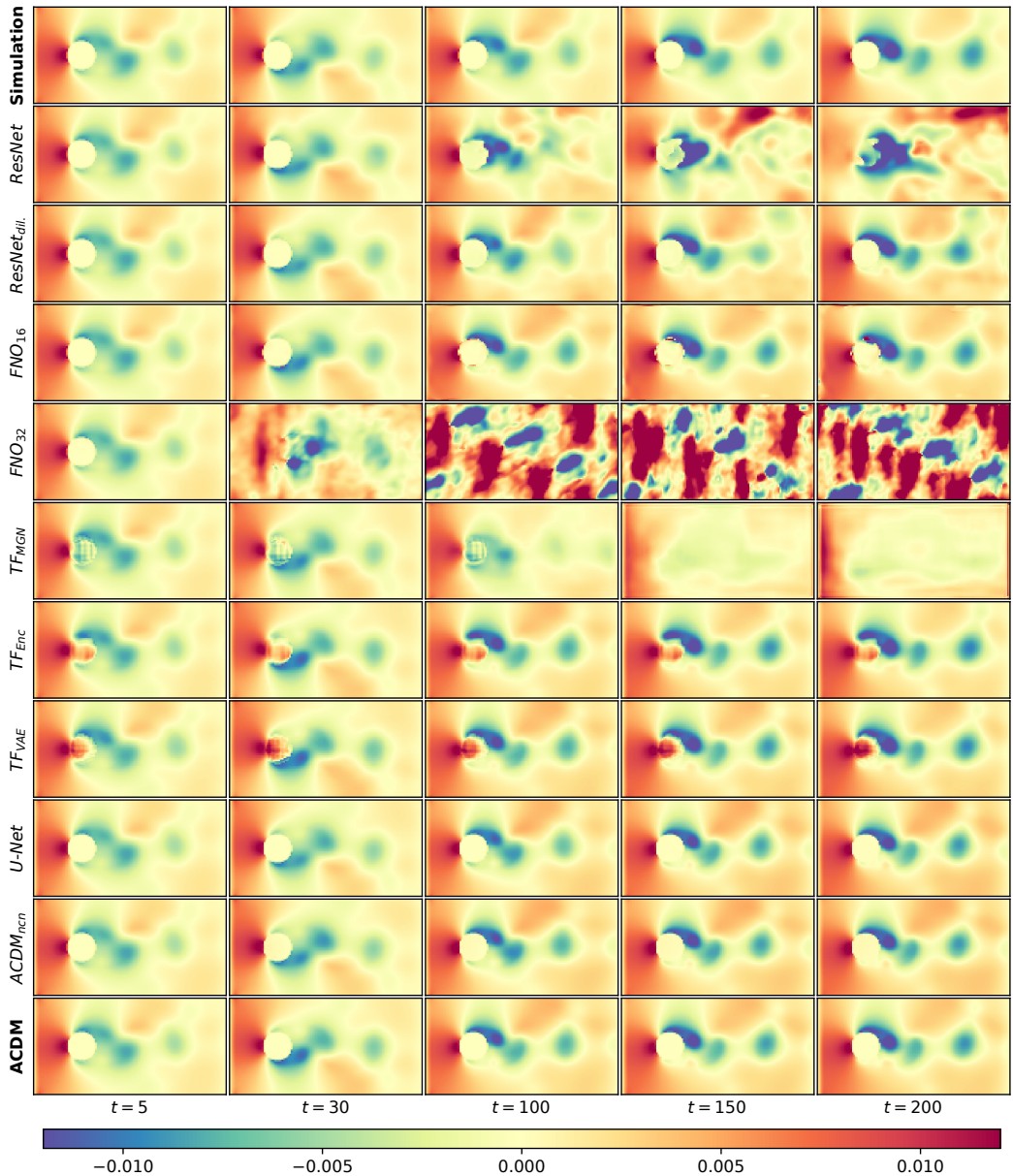

Figure 27: Pressure predictions for the $\texttt{Inc}_{\texttt{var}}$ sequence.

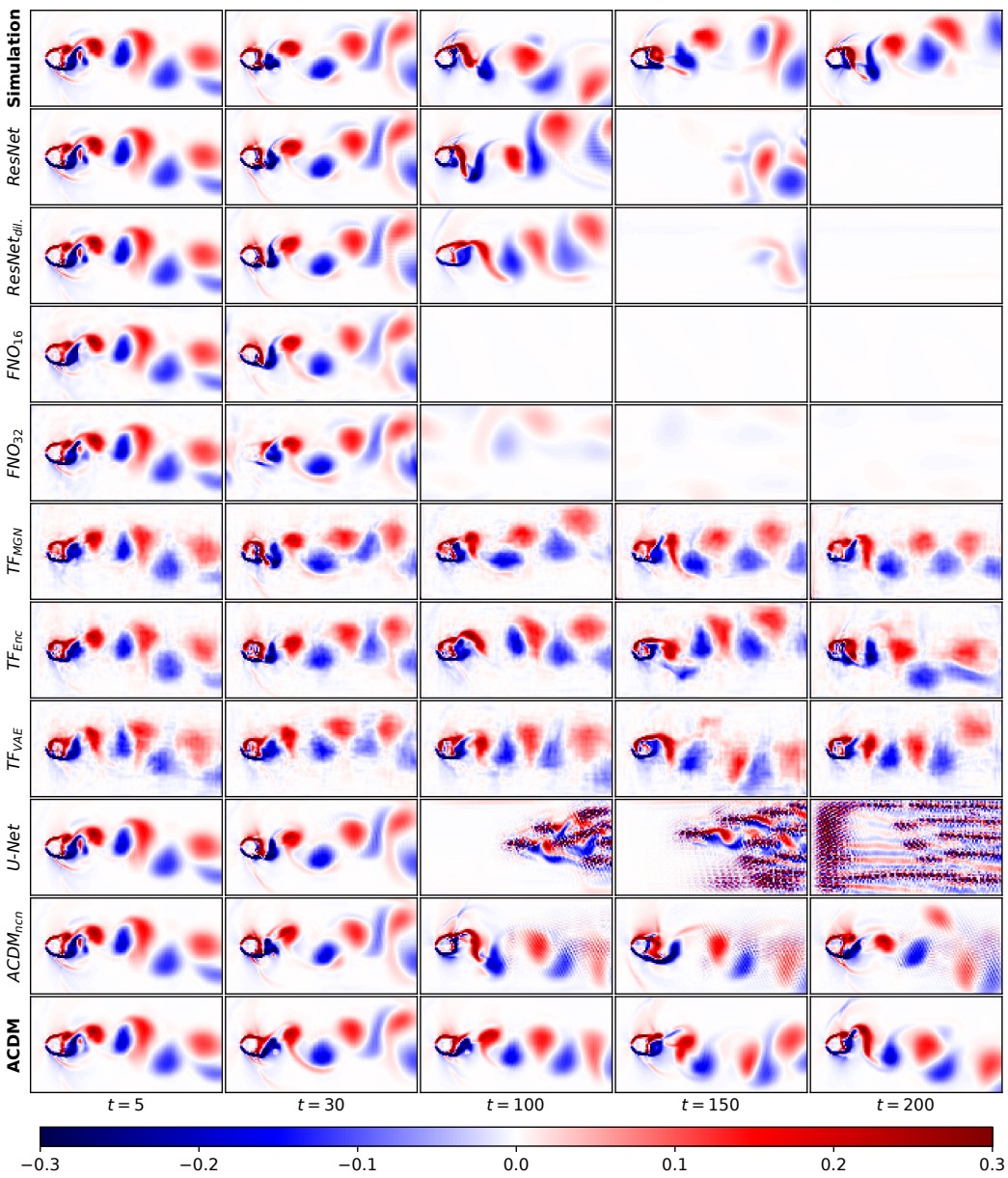

Figure 28: Vorticity predictions for an example sequence from $\texttt{Tra}_{\texttt{long}}$ with $Ma = 0.64$.

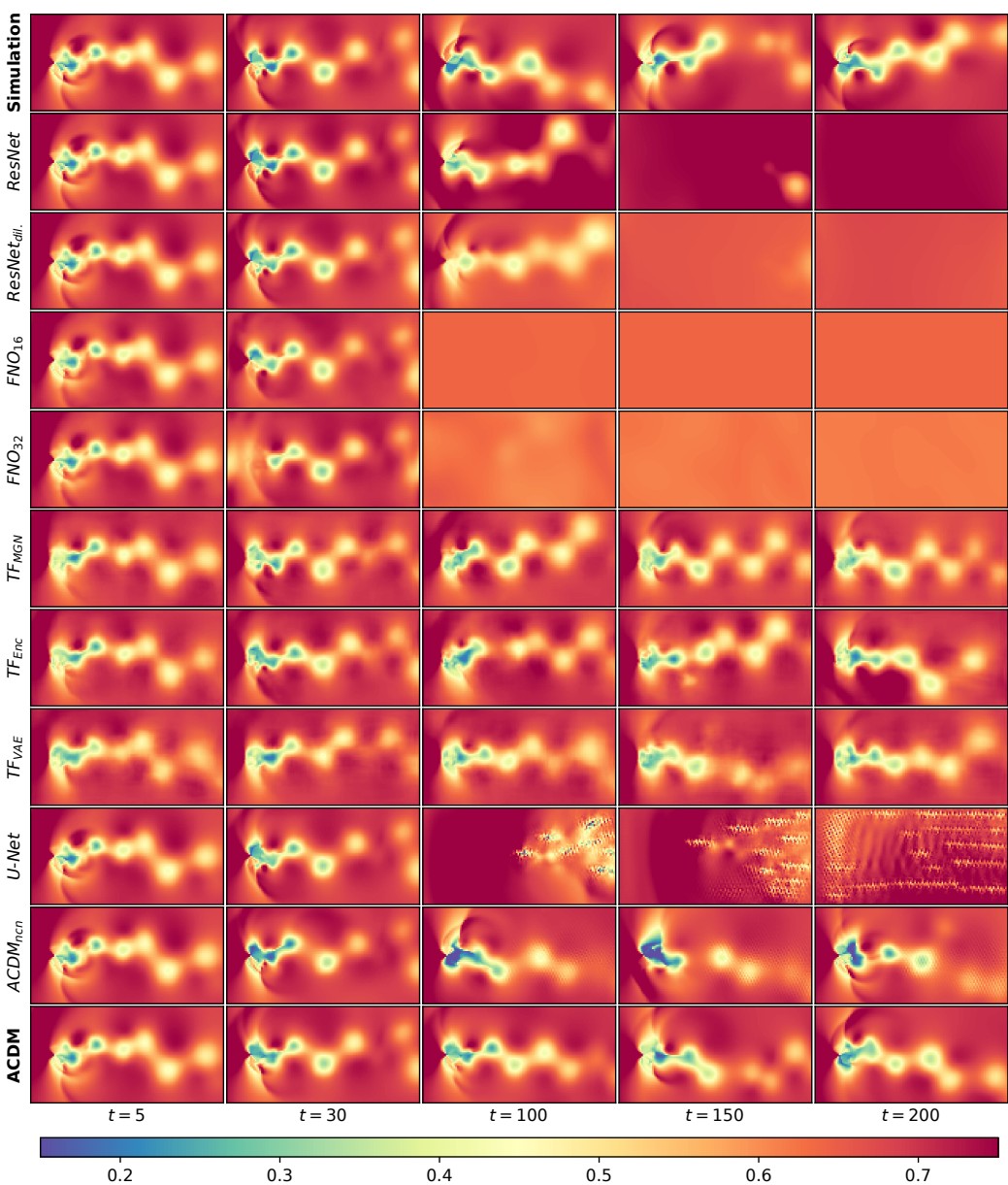

Figure 29: Pressure predictions for an example sequence from $\texttt{Tra}_\texttt{long}$ with $Ma = 0.64$.

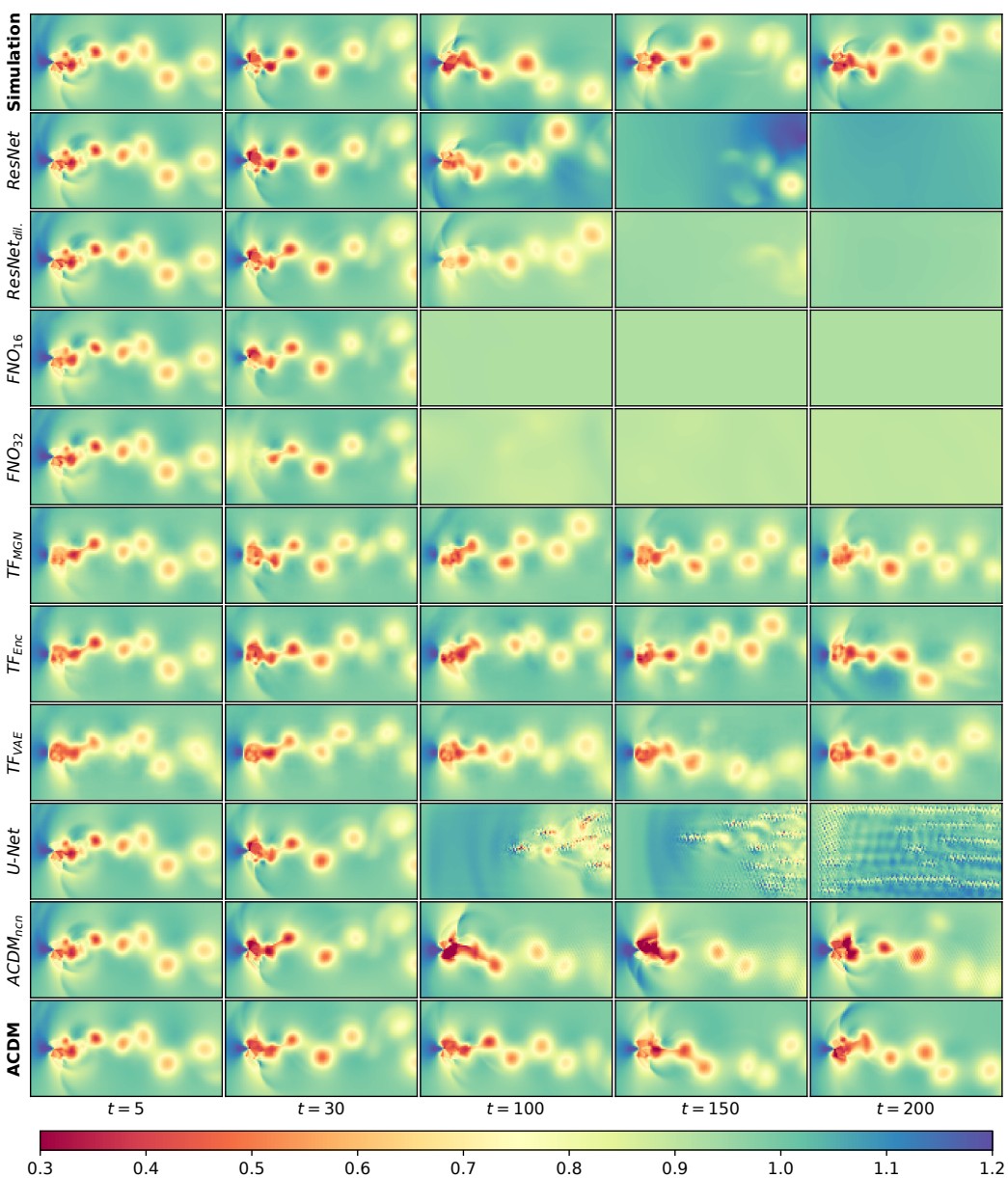

Figure 30: Density predictions for an example sequence from $\mathtt{Tra_{long}}$ with $Ma = 0.64$.

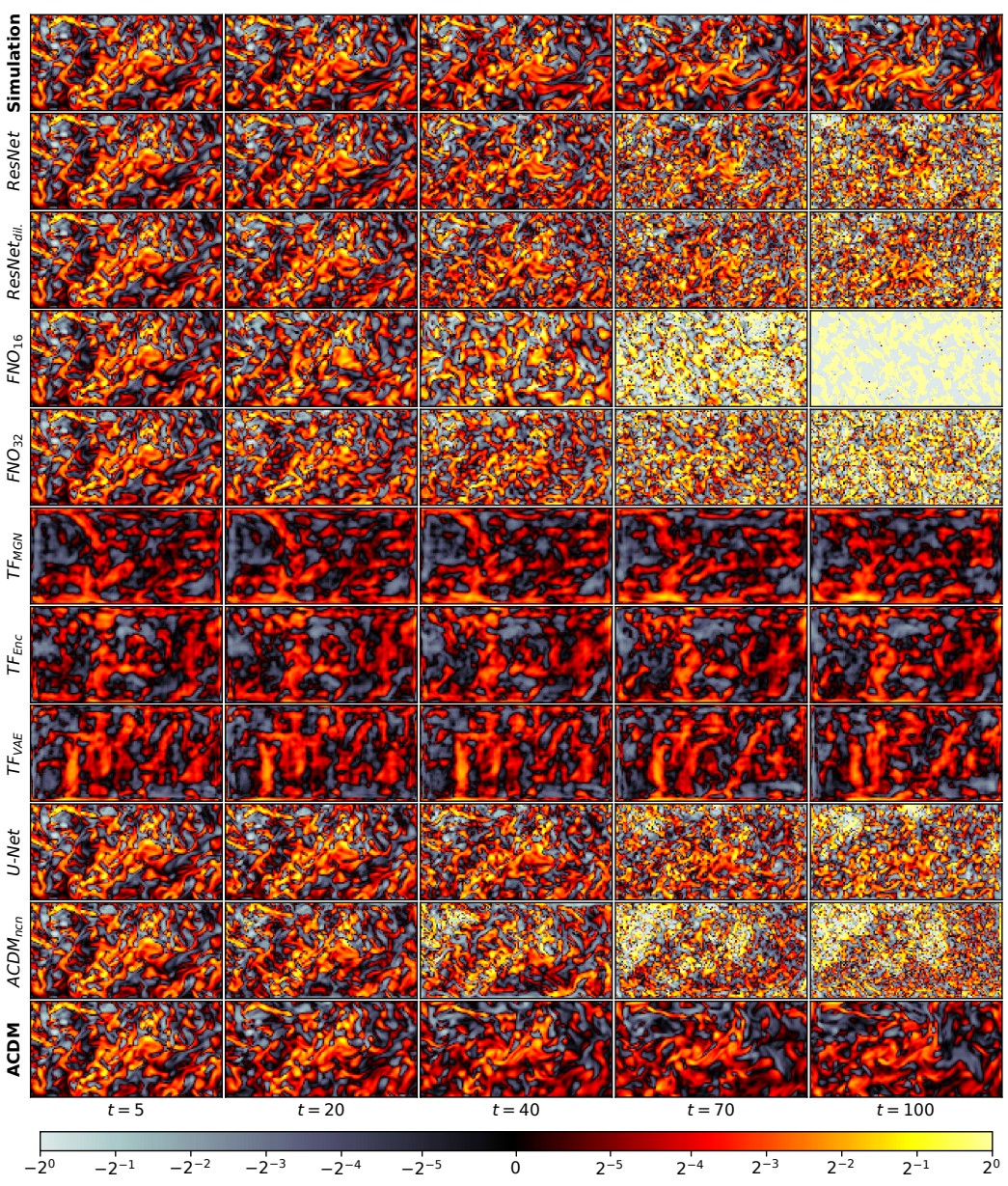

Figure 31: Vorticity predictions (only z-component) for an example sequence from $\texttt{Iso}$ with $z = 280$.

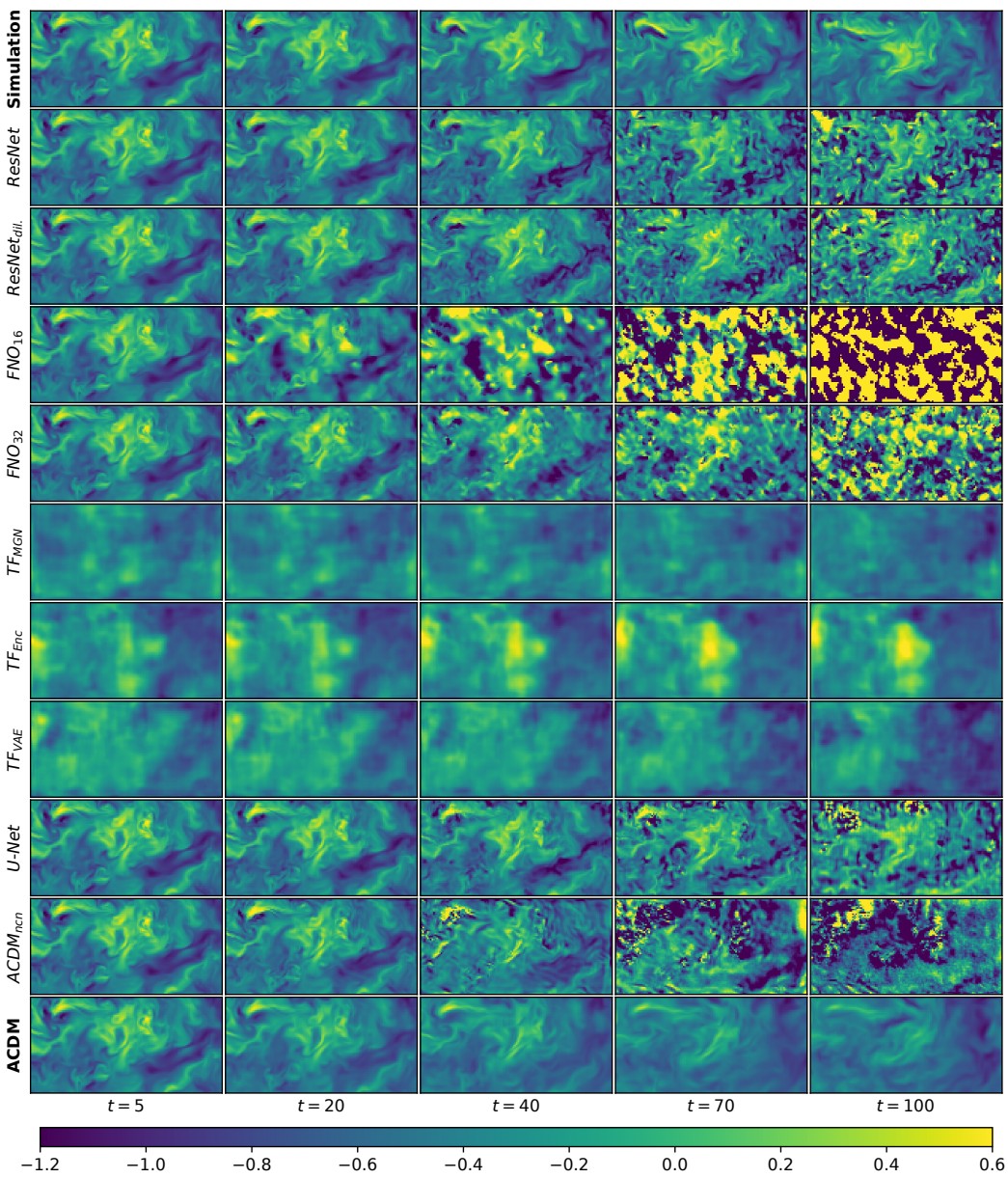

Figure 32: Z-velocity predictions for an example sequence from Iso with $z = 280$.

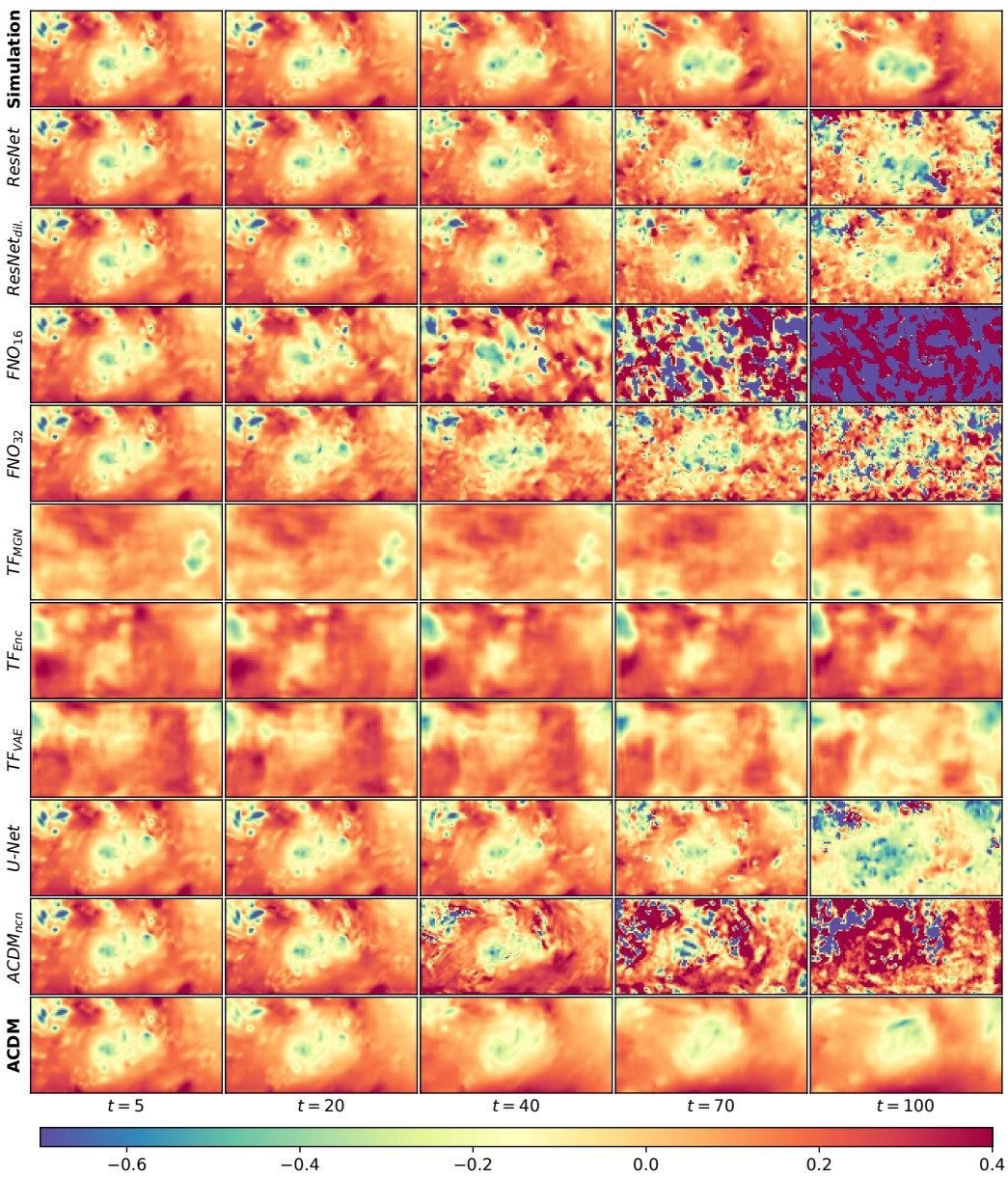

Figure 33: Pressure predictions for an example sequence from Iso with $z = 280$.

# E    ABLATION STUDY PREDICTION EXAMPLES

Over the following pages, we show prediction examples from different ablation study models provided in Appendices C.4 to C.6 and C.9. Shown are the pressure field from $\text{Tra}_{\text{long}}$ with $Ma = 0.64$, as well as a vorticity sequence from $\text{Iso}$ with $z = 280$.

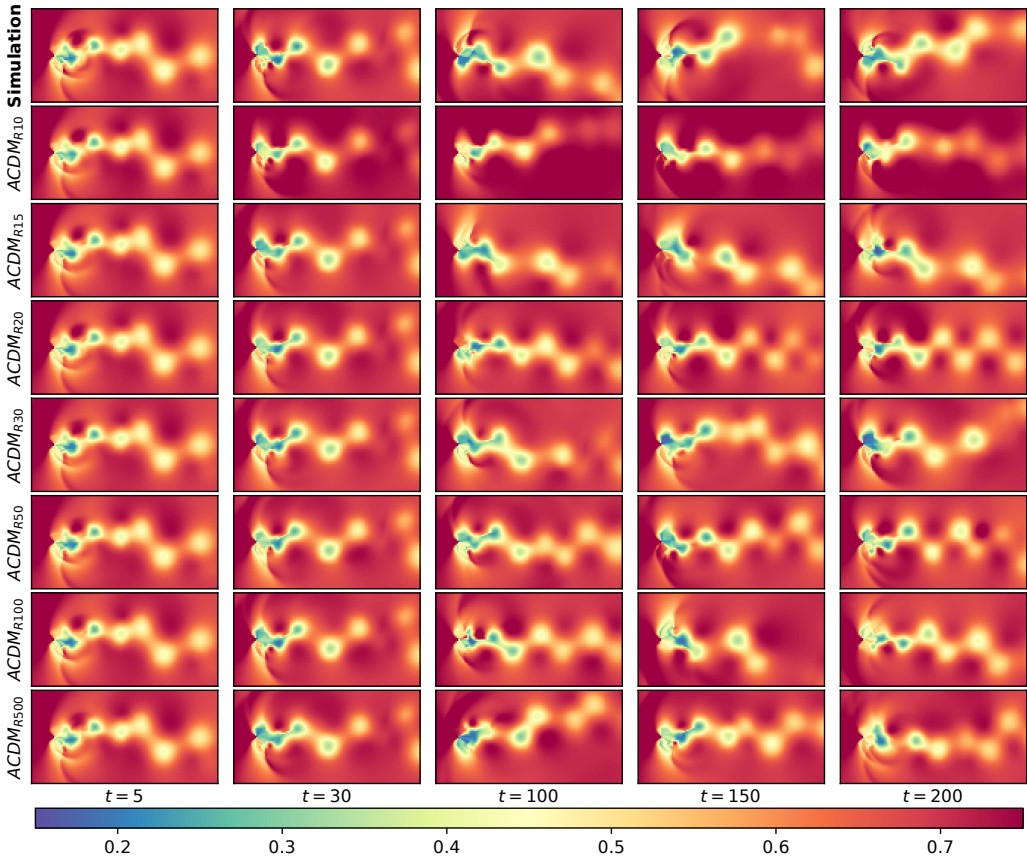

Figure 34: Diffusion Step Ablation (see App. C.4): Pressure predictions from $\text{Tra}_{\text{long}}$.

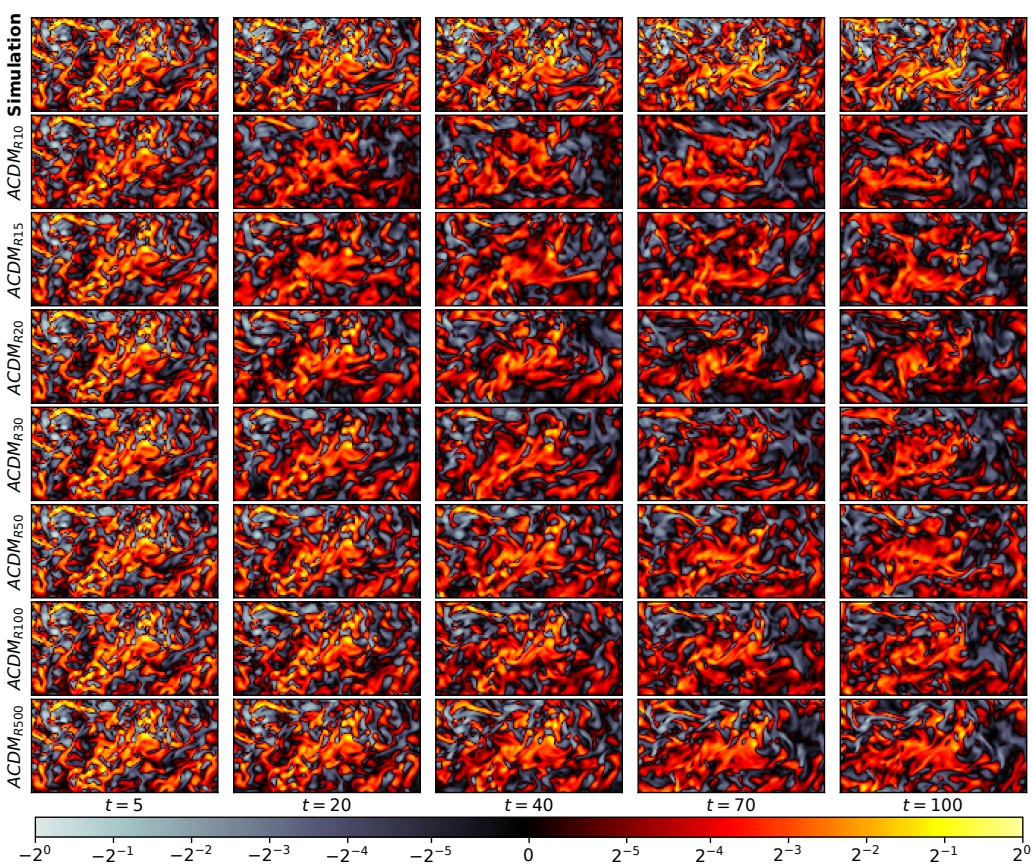

Figure 35: Diffusion Step Ablation (see App. C.4): Vorticity predictions from `Iso`.

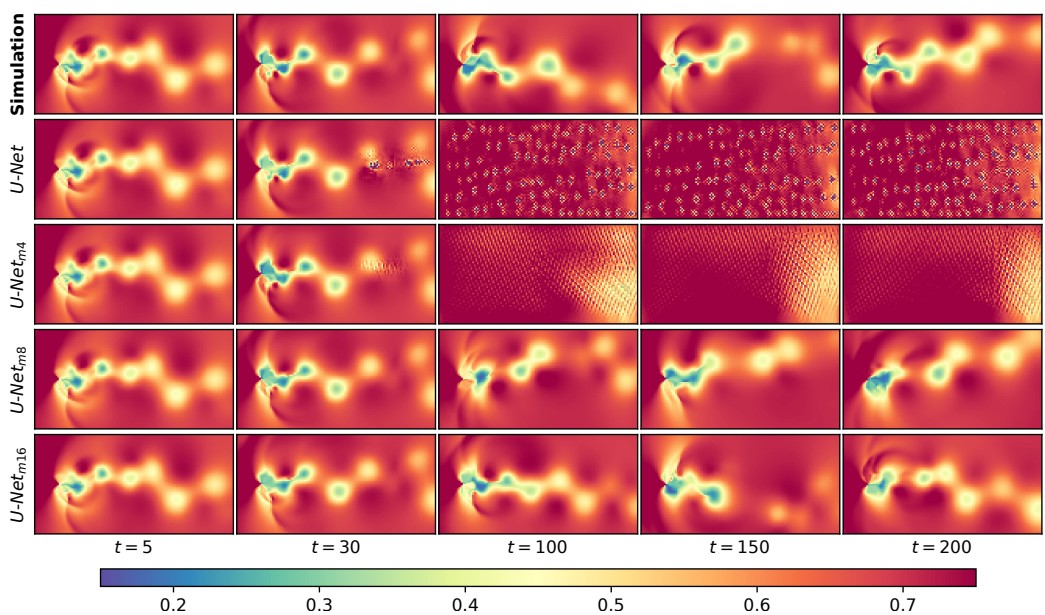

Figure 36: Training Rollout Ablation (see App. C.5): Pressure predictions from $\mathtt{Tra_{long}}$.

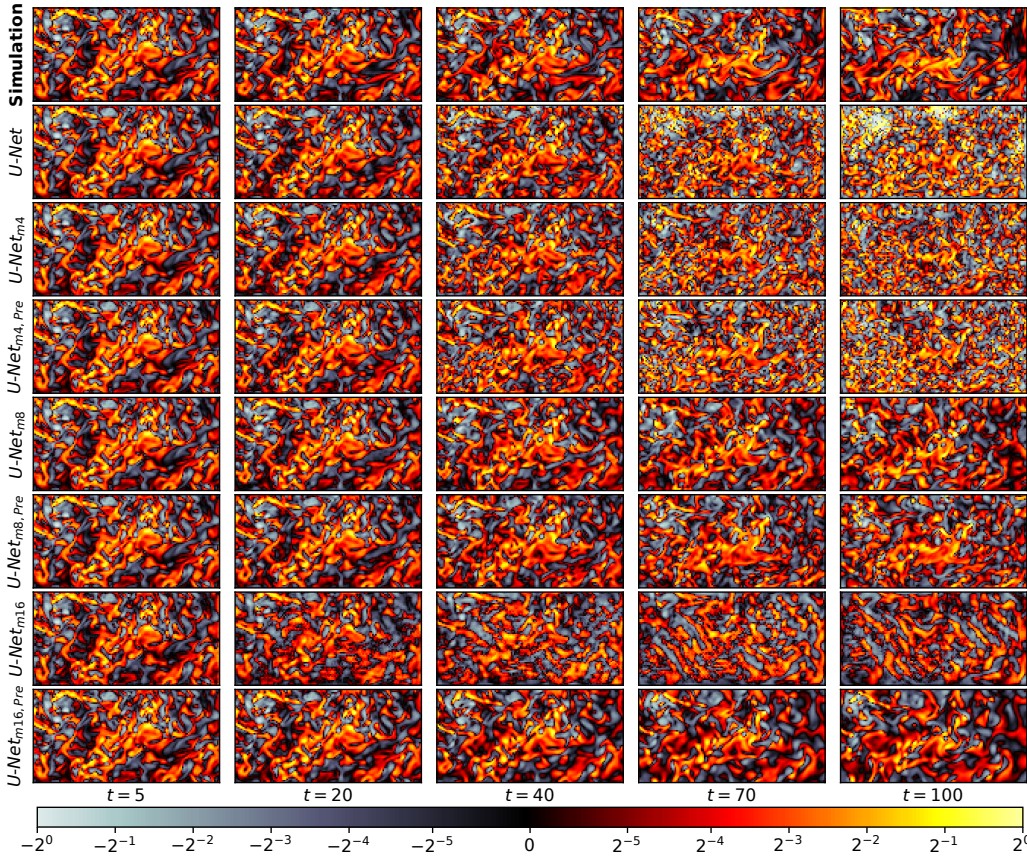

Figure 37: Training Rollout Ablation (see App. C.5): Vorticity predictions from $\mathtt{Iso}$.

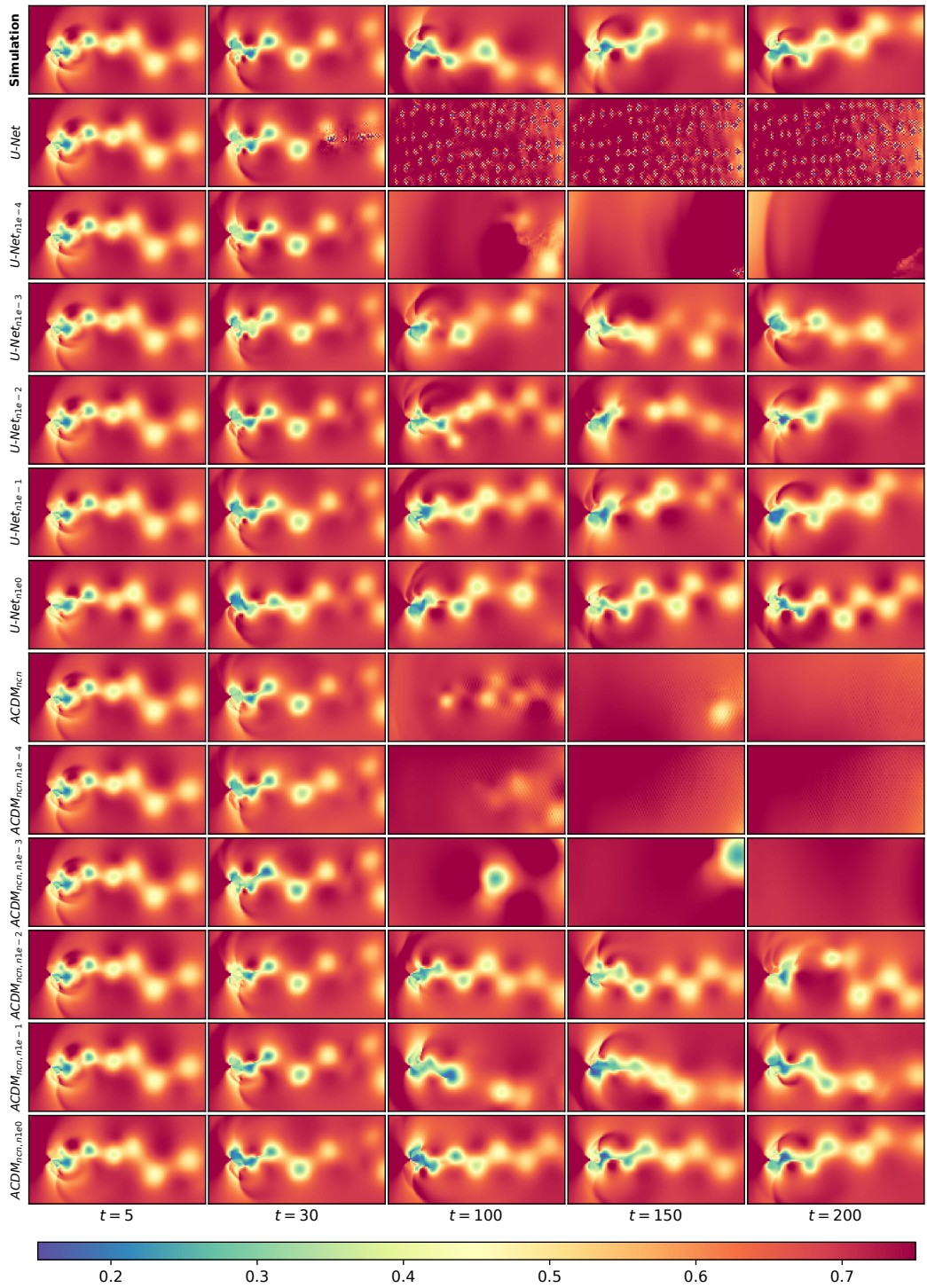

Figure 38: Training Noise Ablation (see App. C.6): Pressure predictions from $\texttt{Tra}_{\texttt{long}}$.

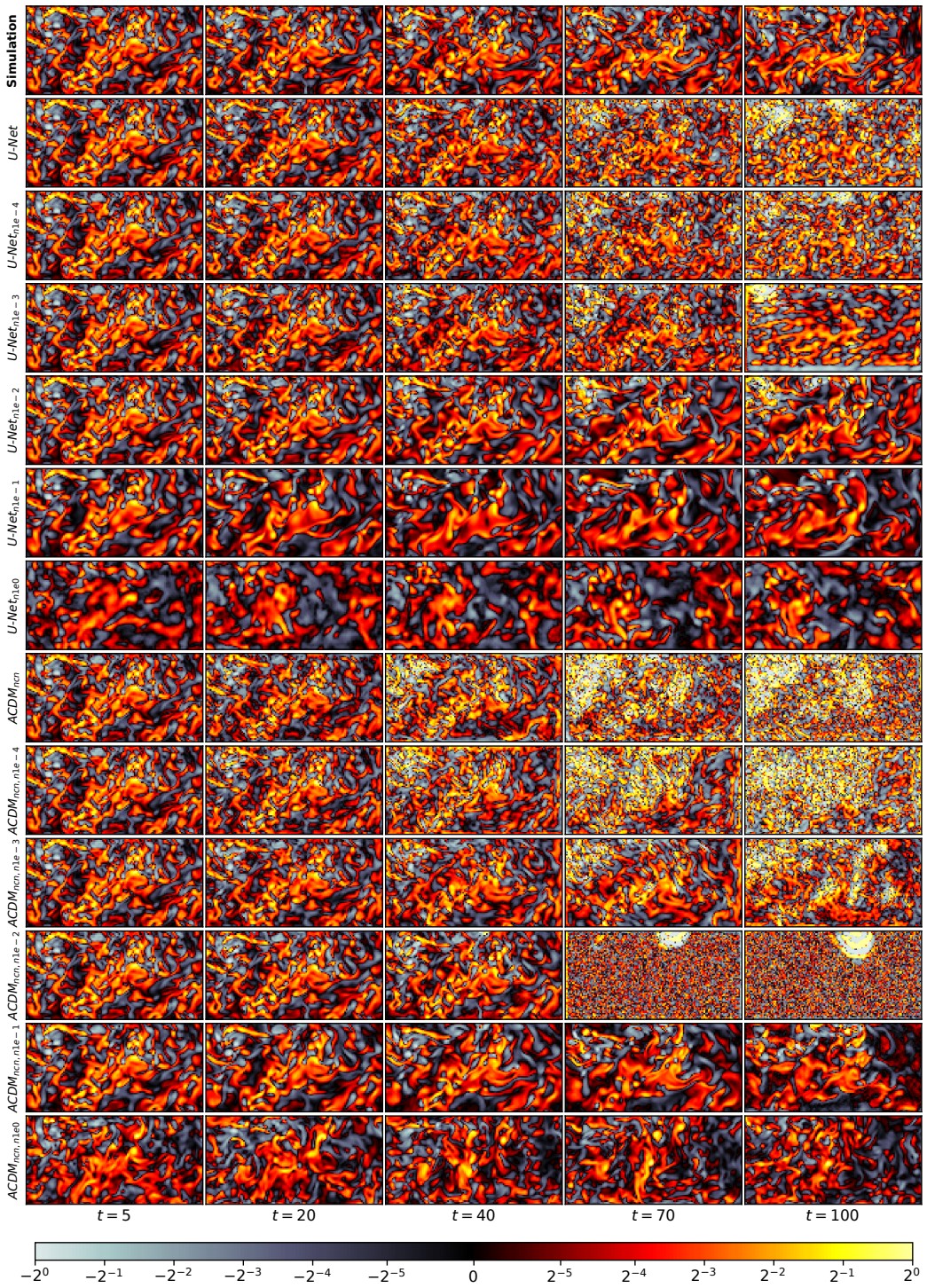

Figure 39: Training Noise Ablation (see App. C.6): Vorticity predictions from `Iso`.

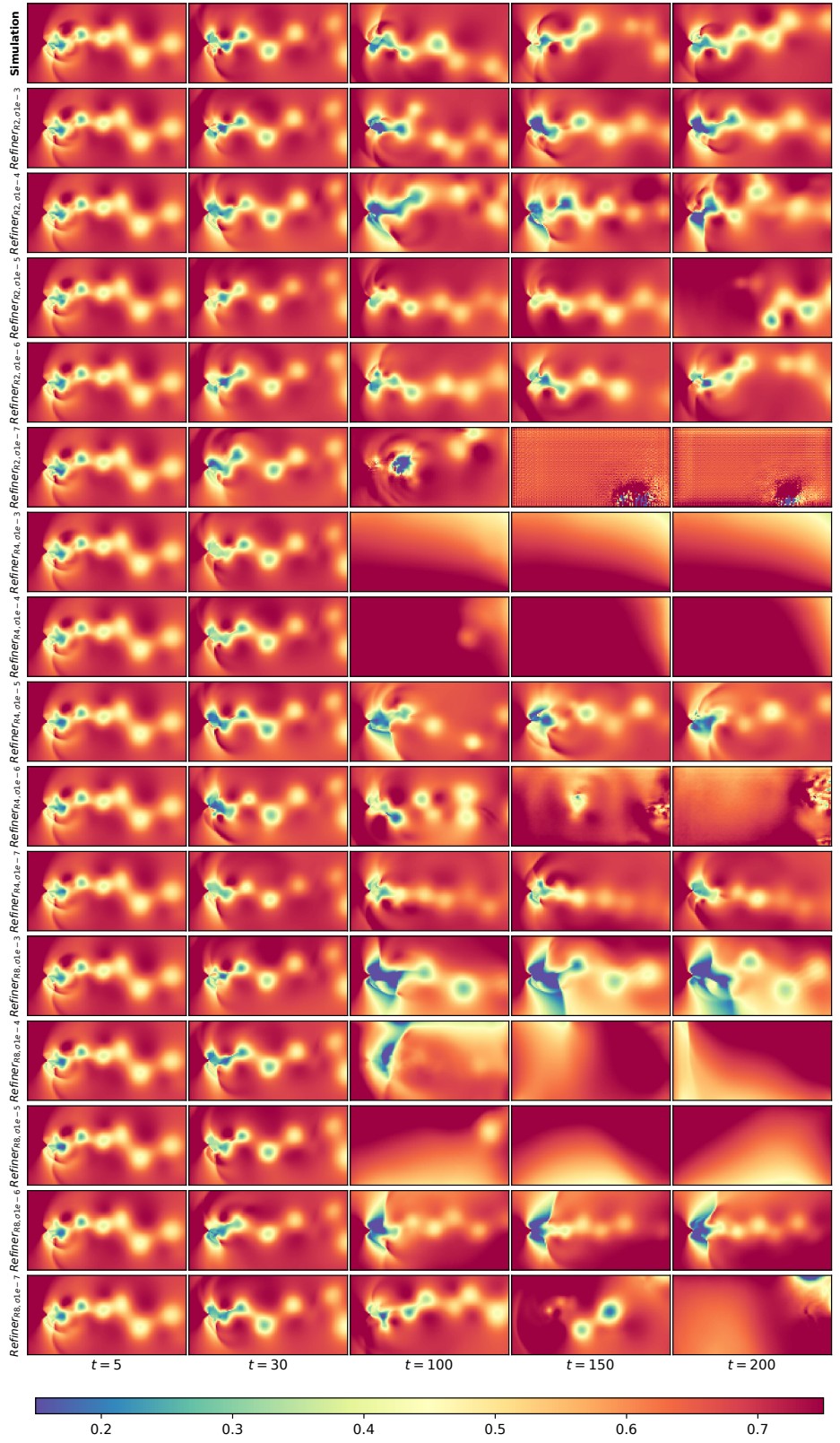

Figure 40: Comparison to PDE-Refiner (see App. C.9): Pressure predictions from $\text{Tra}_{\text{long}}$.

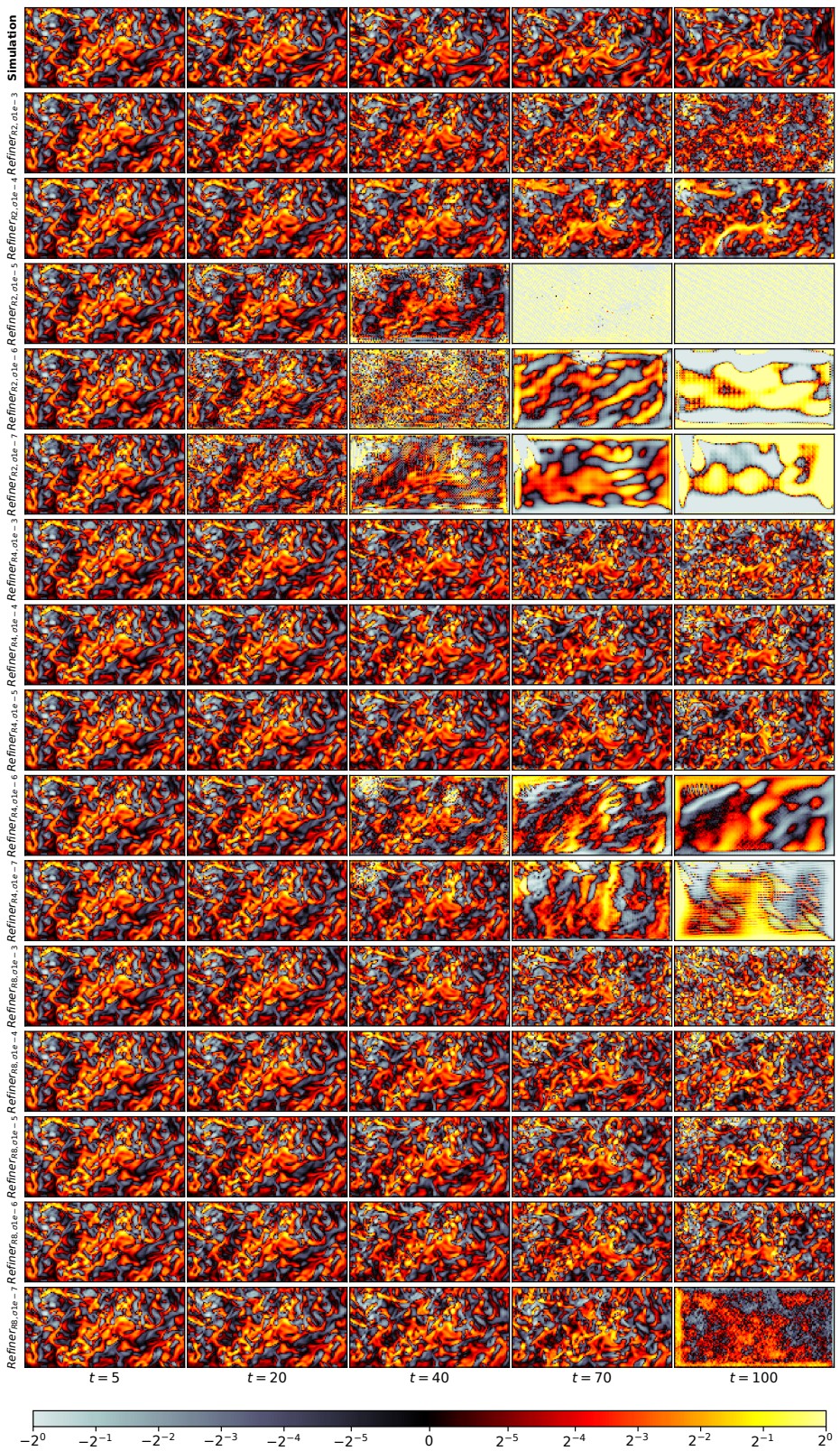

Figure 41: Comparison to PDE-Refiner (see App. C.9): Vorticity predictions from Iso.

