# OpenReview forum: "Turbulent Flow Simulation using Autoregressive Conditional Diffusion Models"
_ICLR.cc/2024/Conference — Submitted to ICLR 2024_

### Official Review · Reviewer_1NQ6 · 2023-10-25

**Soundness:** 2 fair
**Presentation:** 3 good
**Contribution:** 2 fair
**Rating:** 5
**Confidence:** 5

**Summary:**

The manuscript describes an application of the conditional diffusion model for the simulation of complex physical system, turbulent flows. The authors performed an extensive numerical studies using a range of solvers and a few different flow geometries. It is shown that overall the diffusion model outperforms supervised approaches.

**Strengths:**

The authors performed large-scale simulations and extensive numerical experiments to investigate the conditional diffusion model for the physics problems. The result seems to suggest an advantage of the diffusion model in physics simulations.

**Weaknesses:**

While it is interesting to see the capability of the diffusion model in learning physics problems, the study does not go beyond a relatively straightforward application of the conditional diffusion model, which does not align well with the scope of ICLR. The authors used simple evaluation metrics, which may miss important characteristics of physics problems.

**Questions:**

1. One of the most important characteristics of the physics problem is the conservation law. If not the mass conservation constraint, the computation becomes just a very simple matrix vector multiplications. What's the divergence-free error of the diffusion model and how it compares with the computational physics model?

2. MSE error may not be the best metric to investigate the physics problem. For example, it will be helpful to compare the power spectrum to see if the nonlinear energy transfer is correctly represented in the diffusion model. For the turbulent problems considered, there are well defined metrics that give better representation of the physics. The authors need to compare those metrics, instead of simple MSE.

3. What does it mean to have a posterior sampling? While the diffusion model can sample from the probability distribution, the problem itself is deterministic. It does not make a sense, simply because the diffusion model can generate a sample from a distribution, suddenly the authors arguing that they can sample from a posterior distribution when the problem setup is deterministic. If the authors consider the primitive variables as random variables, the problem formulation has also be properly stated and changed.

4. Again from the comment 3, the paper lacks a proper problem formulation.

---

> ### Author Response · Authors · 2023-11-21
> **Response to Reviewer 1NQ6 [PART 1/2]**
>
> We would like to thank Reviewer 1NQ6 for the comments. In the following, we respond to the weaknesses (W) and answer the raised questions (Q).
>
>
> **Novelty and Contribution (W):**
>
> While the novelty of our work with regard to diffusion methods is limited, we would nevertheless argue that our paper makes some other valuable contributions that are highly relevant for the ICLR community: Multiple concurrent methods investigate a similar problem of flow prediction, showing the relevance of the topic, however none of them considers a direct application of DDPMs in detail. We believe this is an important step that should be taken before moving on to more complex methods. To achieve this, we provide a thorough analysis for various tasks of different difficulty, by comparing a range of established baselines to our method, and we use a wide set of physical evaluation metrics. Furthermore, we provide a detailed appendix with additional evaluations and insights. We also comment on this aspect in more detail in the general response.
>
>
> **Simple metrics (W):**
>
> We would like to respectfully disagree with the statement that we only investigate simple metrics. As detailed in the following, one core aspect of our work is that a range of different and meaningful metrics are considered. This includes temporal as well as spatial power spectra, correlations, temporal gradient stabilities, as well as the deep learning-based LSiM metric, which is designed for meaningful comparisons of numerical simulation methods. As such, we believe our evaluations paint a detailed picture of the different architectures and training modalities we investigate.
>
>
> **Physical Conservation Laws (Q1):**
>
> This is interesting evaluation and we are happy to provide divergence-free errors for our method here. Since our ground truth simulations are downsampled from a higher simulation resolution for training and evaluations, the ground truth data will also not be fully divergence-free. As such, the errors are only meaningful when comparing to the divergence-free errors of the simulation data. Here, we investigate these errors only on the incompressible flow data set $\texttt{Inc}$ and the isotropic turbulence $\texttt{Iso}$, as the transonic flow $\texttt{Tra}$ is compressible, and thus does not exhibit divergence-free flow fields. To calculate the error, we compute spatial divergence values from velocity gradients computed via central differences. The absolute divergence values are averaged across spatial and temporal dimensions, samples, trained models, and data set sequences. The standard deviation is computed across samples, trained models, and data set sequences. As shown below, ACDM achieves very similar divergence-free errors on $\texttt{Inc}$ compared to the simulation baseline, and even results in lower errors on $\texttt{Iso}$.
>
> For the transonic flow $\texttt{Tra}$, we investigate the conservation of mass instead, by comparing the overall mass in the diffusion predictions with the simulation baseline once again. For this, we aggregate volume-weighted densities across spatial and temporal dimensions, samples, trained models, and data set sequences. The standard deviation is computed across samples, trained models, and data set sequences. As shown below, ACDM results in very similar mass compared to the simulation. For comparison we also report the result of the strongest baseline across cases, the dilated ResNet, here:
>
> 1. Divergence-free error on $\texttt{Inc}_{low}$:
> - Simulation: 0.00049 +- 0.00004
> - ResNet-dil: 0.00135 +- 0.00068
> - ACDM: 0.00062 +- 0.00014
>
> 2. Divergence-free error on $\texttt{Inc}_{high}$:
> - Simulation: 0.00111 +- 0.00003
> - ResNet-dil: 0.00128 +- 0.00013
> - ACDM: 0.00116 +- 0.00004
>
> 3. Divergence-free error on $\texttt{Iso}$:
> - Simulation: 0.04359 +- 0.00295
> - ResNet-dil: 0.07985 +- 0.03149
> - ACDM: 0.03069 +- 0.00455
>
> 4. Conservation of mass on $\texttt{Tra}_{ext}$:
> - Simulation: 0.94850 +- 0.00292
> - ResNet-dil: 0.94651 +- 0.00792
> - ACDM: 0.94757 +- 0.00382
>
> 5. Conservation of mass on $\texttt{Tra}_{int}$:
> - Simulation: 0.90350 +- 0.00611
> - ResNet-dil: 0.89958 +- 0.00764
> - ACDM: 0.90779 +- 0.00575

---

> ### Author Response · Authors · 2023-11-21
> **Response to Reviewer 1NQ6 [PART 2/2]**
>
> **Metrics beside MSE (Q2):**
>
> We do consider a range of physical evaluations besides MSE values. In addition to the physical metrics shown above, we the evaluate deep learning-based LSiM metric from Kohl et al. (see Fig. 4, Tab. 2). Second, we compare temporal statistics in the form of temporal change rates (see Fig. 8, and a range of results in the ablation appendices C.4 to C.9), as well as correlation to the simulation reference (see Fig. 7 on the right). Finally and most importantly, we do already consider a range of spectral evaluations for temporal as well as spatial aspects (see Fig. 6, Fig. 7 on the left, Fig. 11, Fig. 13, and a range of results in the ablation appendices C.4 to C.9).
>
>
> **Problem Setting and Posterior Sampling (Q3):**
>
> The key problems we consider are not deterministic, which means that using probabilistic methods like diffusion models are justified. The isotropic turbulence data set $\texttt{Iso}$ contains slices of 2D data from a 3D simulation. This means there is a large range of possible solutions for the evolution of a slice, depending on the flow outside this slice. Thus, the prediction of a slice without knowledge of the outside behavior is by definition underdetermined, and as such non-deterministic. The behavior of the transonic flow $\texttt{Tra}$ is also a probabilistic problem. It only exhibits statistically stable average behavior, but the evolution of individual states is not deterministic: Since the simulation is performed with a delayed detached eddy simulation (DDES), not all spatial scales are resolved, meaning there is a range of possible physically plausible evolutions starting from a single state. Furthermore, resampling the data from the fine, irregular simulation mesh to a regular, coarse Cartesian grid introduces additional errors and thus uncertainties. As such, the possibility of creating posterior samples has great potential to increase the accuracy and robustness of fluid flow predictions in these settings.
>
> The only problem we analyze that is deterministic is the incompressible wake flow data set $\texttt{Inc}$. It is an established baseline across previous work, and the simplest test case we consider, and such not crucial for the insights of our work. Nevertheless, it is interesting to investigate, to show that diffusion models can also work just as well as deterministic models on such cases.
>
> Another aspect to consider is that real-world applications (e.g., particle imaging velocimetry (PIV), which would be a natural next step for future work) are in general probabilistic. This can be due to a range of factors, the most important of which are measurement noise, measurement sparsity, and uncertainty in the estimation of physical parameters.
>
> We hope we addressed the main weaknesses and questions raised above, and as such kindly ask Reviewer 1NQ6 to reconsider their evaluation of our work.

---

### Official Review · Reviewer_ySpT · 2023-10-28

**Soundness:** 2 fair
**Presentation:** 2 fair
**Contribution:** 2 fair
**Rating:** 5
**Confidence:** 4

**Summary:**

The authors introduced a method that trains diffusion models to capture the joint distribution of turbulent flow states over a few time steps, and apply conditioning at inference time to perform autoregressive rollouts. The authors compare the proposed method against models which are trained autoregressively using multiple flow examples and observed competitive accuracy and temporal stability characteristics.

**Strengths:**

* Paper is generally well written and not difficult to follow
* Evaluation is done with meaningful benchmark methods; ablation studies are comprehensive
* Proposed model is capable of generating probabilistic predictions, whereas most competitors are deterministic in nature
* The proposed method does seem to result in good stability characteristics when rolled out for an extended period of time - an important challenge for many existing methods dealing with dynamical systems

**Weaknesses:**

* The approach is not novel - it simply applies an existing way of conditioning diffusion model to do autoregressive rollouts of turbulent flow trajectories.
* The presented benchmark is not fair in two ways
    * Autoregressive diffusion sampling is very expensive, basically taking <the total number of denoising steps> times more (~20 in this case) compute than the U-Net model. A fair comparison would involve a U-Net either with larger capacity or integrated forward at finer time steps such that the compute cost is comparable.
    * The authors do not show the results for "rolling out in training" as the primary baseline in the main text (it is instead presented as ablation studies in section C.5). However, it is well established (authors even include references supporting this) that this is the correct way of training autoregressive models. Indeed, the proposed ACDM does not have better performance compared to the models trained with such multi-step loss. Considering the significantly higher inference cost, it is hard to justify the value of the proposed method. The authors mentioned "more hyperparameters" and "higher training cost" as counter-arguments, but I do not think the former is a valid reason at all, and the latter is both weak and not supported by numbers comparison.
* In the attached videos for the isotropic turbulence "posterior_iso_samples_vort.mp4", the samples showed visible flickering, i.e. some small-scaled features present in one frame noticeably go missing in the next. This seems to suggest that the conditioning scheme adopted may not lead to sufficient coherence between conditioned and sampled parts ($d$ and $c$). This is not reflected in any of the metrics presented.

These weaknesses are fundamental enough for me to not recommend a passing score.

**Questions:**

* It would be helpful to define LSiM somewhere besides including the reference.
* Figure 4 is missing labels - which plots correspond to which test example?
* I wasn't able to find spatial frequency analysis for the isotropic turbulence example? The rollouts look a bit smoother compared to the ground truth visually.
* Appendix C.5 summary - what do you mean by "significantly reducing the complexity of the learning task, instead of fundamentally increasing the models generalization ability"? I cannot connect this statement with what is entailed by training rollouts beyond single step
* Appendix C.6 Table 5 - the "n1e-x" subscripts looked really cryptic to me initially. Maybe just create a separate column to indicate the noise level?

---

> ### Author Response · Authors · 2023-11-21
> **Response to Reviewer ySpT**
>
> We want to thank Reviewer ySpT for reviewing our submission and for the feedback. In response to the mentioned weaknesses (W) and questions (Q), we would like to present arguments to address the raised concerns.
>
>
> **Novelty (W1):**
>
> While the novelty of our work with regard to conditional diffusion methods is limited, we would nevertheless argue that our paper makes some other valuable contributions: Taking a direct approach to applying DDPMs to flow prediction is an open point in multiple concurrent methods, and an important step that should be taken first in our opinion. Here, we provide a thorough analysis of different tasks, by comparing various established baselines to our method, and use a range of physical evaluation metrics. We also comment on this aspect in more detail in the general response.
>
>
> **Comparison at Equal Inference Cost (W2.1):**
>
> Making comparisons that are equal across every aspect are very difficult for the problems and baselines we consider. Here, we chose to compare models with a similar capacity measured via the number of trainable parameters and similar training memory requirements. As such, the comparison between U-Net and ACDM is fair, as both use the same number of trainable parameters and architecture, but a different training and inference paradigm. Furthermore, the suggestion of unrolling U-Nets at finer temporal steps would potentially worsen their performance due to additional error accumulation across more rollout steps, causing earlier diverging behavior.
>
>
> **Comparison with Unrolled Training (W2.2):**
>
> In spirit of the comparison at equal terms mentioned with regard to W2.1 above, it would  conceptually not really be fair to compare U-Nets that receive a substantially larger rollout horizon at training time compared to ACDM. As mentioned in our work, an interesting direction for future research is the application of unrolling at training time to the diffusion approach. Thus, the stability gains from the diffusion approach and the training time unrolling are potentially orthogonal and could be combined, justifying a comparison to U-Nets without unrolling.
>
> We agree that the inference costs of a diffusion based setup are high, but we are confident that future work in the domain will quickly be able to improve this performance with techniques like distillation or improved sampling, in the same way as already emerging in the image domain. First steps towards these speedups were already taken in some concurrent work as discussed in the general response. Furthermore, we would like to clarify that unrolling U-Nets does lead to substantially higher training costs as shown in Appendix B.6. Another key benefit of a diffusion-based predictor model is the ability to create posterior samples which unrolled U-Nets are still unable to do.
>
>
> **Temporal Coherence (W3):**
>
> This aspect is not a fundamental problem with the conditioning for the diffusion setup. Rather it is an indication of the substantial difficulty and underdetermined nature of the $\texttt{Iso}$ data set, as it does not occur on the other test sets we investigated.
> Furthermore, the stability analysis via temporal gradients is a metric that considers such issues. For example, this is quite apparent for TF-Enc, which shows clear spikes in this evaluation in Fig. 8, indicating problems with temporal coherence issues. For ACDM, no such issues are visible in Fig. 8, meaning the impact of this visual flickering is quite minor. Substantial fluctuations would either appear as spikes in the mean or as unreasonably large or inconsistent values in the standard deviation in this evaluation. Note that deterministic models are fundamentally less susceptible to problems with temporal coherence, as they circumvent the issue by not providing the ability to create posterior samples in the first place. We also observed that this flickering behavior can be reduced further by increasing the number of diffusion steps.
>
>
> **LSiM Definition (Q1):**
>
> Thanks for the suggestion, this is included in Appendix C.3 now.
>
>
> **Fig. 4 (Q2):**
>
> Thanks for pointing this out, it is fixed.
>
>
> **Spatial Frequency for $\texttt{Iso}$ (Q3):**
>
> This is shown in Fig. 11 in Appendix C.1. As mentioned in your question, we did observe and describe that ACDM appears to be dissipative on longer rollouts for $\texttt{Iso}$.
>
>
> **Argumentation in Appendix C.5 Summary (Q4):**
>
> We agree that the formulation here was unclear, and updated it accordingly.
>
>
> **Subscripts in Appendix C.6 Table 5 (Q5):**
>
> Thanks for the suggestions, we updated this (and the other tables in the appendix) to be more clear.

---

> > ### Comment · Reviewer_ySpT · 2023-11-21
> > **Thanks for your response.**
> >
> > I would like to thank the authors for the detailed response. I remain with my assessment because I believe acceptance should not  be based on the fact that conditional diffusion (especially with low novelty/originality) can be applied, but rather there are clear advantages in applying it compared to existing methods. The additional inference (an order of magnitude higher) is really hard to justify given the relatively marginal benefits demonstrated.

---

### Official Review · Reviewer_wsiC · 2023-10-31

**Soundness:** 2 fair
**Presentation:** 3 good
**Contribution:** 2 fair
**Rating:** 5
**Confidence:** 5

**Summary:**

This paper proposes to apply conditional diffusion models to turbulence flow simulations. It demonstrates that diffusing the conditional inputs (i.e. the initial conditions of the simulation) aids performance as opposed to using the "clean" conditions for all diffusion steps.
The authors show that the resulting diffusion model attains good sample diversity and physical consistency, albeit at the expense of slower inference speed. It can also serve as an effective method for stabilizing long inference rollouts.

**Strengths:**

Overall, I enjoyed reading this paper but think that switching the focus away from the conditional diffusion model (given the limited novelty and its limitations, see Weaknesses section) to a more general focus on comparing various approaches for data-driven physics simulations could be helpful to the reader so that the authors can focus on candidly analyzing and comparing the different methods, for which the current paper already provides a lot of interesting and valuable content. This is especially so given that the authors have had to come up with their own adaptations to use some interesting baselines such as the TF_VAE. Even if they don't perform well, it is very valuable to discuss them, as is done in this paper.

List of strengths:
- Diffusing/noising the conditional inputs is shown to be an effective way of improving the performance of the diffusion model. Albeit this is a small and simple design choice, it is not obvious, and it is good to have it documented for interested practitioners.
- Several interesting observations and analyses are given in this paper, both regarding the proposed diffusion model and some of the baselines. For example, I enjoyed reading about the ablation of the number of diffusion steps and the training rollout and noise, as well as the shortcomings of the transformer-VAE.
- Data-driven models for probabilistic physics simulation is an important and somewhat underexplored field, albeit see the weaknesses below for relevant related work that goes a bit beyond this work.

**Weaknesses:**

- The proposed autoregressive conditional diffusion model (ACDM) is a direct application of common diffusion models to turbulence simulation data. That is, except for proposing to diffuse the conditional inputs there are no methodological contributions.
- Alternative methods to ACDM achieve comparable or even better benefits in terms of accuracy and rollout stability. In the paper, this is notably shown through the U-Net trained on multiple steps (i.e. m>2) or the U-Net where noise is injected into the training batches. This makes me uneasy when reading the abstract that claims *"We show that this approach offers clear advantages in terms of rollout
stability compared to other learned baselines"*.
- ACDM is extremely slow at inference time compared to the baselines (20x-100x slower almost). This is to be expected given that the baselines are single-forward pass models, but it should be made more clear by the authors when discussing the (dis-)advantages of the different baselines. E.g. in the last paragraph of the discussion and the summaries in the appendix, it is not fair to mention the disadvantages of the multi-step or training-noise U-Net baselines without noting the inference speed issue of ACDM. Especially for the training-noise baseline, I don't see any disadvantages compared to ACDM (except for potentially lower posterior sample quality. But this has not been shown here).
- A recent work [1] already goes beyond this paper by adapting diffusion models to the same problem setting, lessening the inference speed issue of this paper, and addressing multiple of the outlooks/future work points given in the last paragraph of this paper. While it can be deemed as contemporaneous work, at least, it should be discussed in this paper. Of course, a direct comparison would be optimal, especially since the multi-step U-Net which performs similarly to ACDM in this paper is a baseline that is beaten by the method from [1].
- It seems to me that some key hyperparameters unnecessarily deviate between ACMD and some baselines. This makes the significance of the results less clear. Notably, 1) ACDM uses two past timesteps as input (k=2), but many baselines only use one (k=1); 2) ACDM is trained with the Huber loss (which is a non-standard choice for diffusion models!) but all baselines use the MSE loss; 3) Number of training epochs varies wildly between models (e.g. 3100 for ACDM vs 1000 U-Net on Inc and Tra datasets). To me, these points seem very important to fix or require some explanation at least.
- It is possible to sample multiple predictions from (most of) the baselines by perturbing the inputs. This is an important baseline to have for ACDM, and much more straightforward than the transformer-VAE idea. This should be tried at least for the training-noise U-Net (just keep the same variance for noising inference inputs).
- It would be good to use benchmark datasets rather than creating new ones for the paper. E.g. see [2] which is used by [1] too, or [3]. This would make comparisons so much easier! Given the effort already spent on the current datasets of the paper (e.g. >5 days for some), will you open-source them?
- ACDM introduces multiple new hyperparameters (e.g. R, diffusion schedule, etc.), so I would advise to not claim that introducing new hyperparameters is a problem of the baselines (e.g. last paragraph of the discussion), especially when said baselines are only introducing a single new HP.
- I would advise toning down *"Unlike the original DDPM, we achieve high-quality samples with as little as R = 20 diffusion
steps. We believe this stems from our strongly conditioned setting"* a bit, given that you show that for some problems you need much more diffusion steps (e.g larger R seem to aid performance on the Iso dataset and it may not saturate at the largest R=500 that you tried, which also has the best LSiM).
- The titles of the subplots in Fig. 4 are missing, so it is hard/impossible to tell what results correspond to which dataset.
- Discussion of Fig. 6 in the main text should mention that some baselines perform very similarly, I think, on the frequency analysis (shown in Fig. 13 in the appendix)

[1] Cachay, S.R., Zhao, B., James, H. and Yu, R., 2023. "DYffusion: A Dynamics-informed Diffusion Model for Spatiotemporal Forecasting", NeurIPS

[2] Otness, K., Gjoka, A., Bruna, J., Panozzo, D., Peherstorfer, B., Schneider, T. and Zorin, D., 2021. "An extensible benchmark suite for learning to simulate physical systems", NeurIPS Track on Datasets

[3] Takamoto, M., Praditia, T., Leiteritz, R., MacKinlay, D., Alesiani, F., Pflüger, D. and Niepert, M., 2022. "PDEBench: An extensive benchmark for scientific machine learning", NeurIPS

**Questions:**

- Why not use CRPS as a metric for your probabilistic methods?
- Do you use k=2 for all U-Net models?
- In your figures showing multiple samples (e.g. Fig 5): Why is the third row separate from the first two? Why is the timestep not ordered by rows?
- The hidden spaces of 56 (but even 112) for the FNO, and L=32 for the transformer models, seem pretty low to me?
- How many diffusion steps do you train with?
- Do you always use 5 samples from the probabilistic methods?
- Why do you change your transformer adaptations TF_enc and TF_VAE in terms of encoder/decoder and residual prediction or not compared to TF_MGN?
- Table 3 in the appendix: Can you please run the dashed variants? If not, why? ACDM_R10 on Iso seems like an especially interesting run to try to me.
- What do you mean by *"However, this is achieved by significantly reducing the complexity of the learning task, instead of fundamentally increasing the models generalization ability."*
- Table 5 in the appendix: Any intuition on why perturbing the inputs with 1e-3 performs so badly? I would have expected a more or less smooth transition from 1e-2 -> 1e-3 -> 1e-4. Is there a bug maybe?
- Can you provide visualizations of the multi-step and training-noise baselines?

---

> ### Author Response · Authors · 2023-11-21
> **Response to Reviewer wsiC [PART 1/3]**
>
> We express our gratitude to Reviewer wsiC for the detailed review and feedback. Below, we address the mentioned weaknesses (W) and questions (Q).
>
>
> **Novelty (W1):**
>
> While the novelty of our paper with regard to existing diffusion methods is limited, we would argue that our paper nevertheless makes a valuable contribution: Exploring the strengths of a direct DDPM-based setup for flow prediction, is an open point in several contemporary methods, that should be taken as a first step in our opinion. For that, we provide a detailed investigation into tasks of different difficulty, compare to various established baselines, and use a range of physical evaluation metrics. We also comment on this aspect in more detail in the general response.
>
>
> **Abstract Formulation and Benefits of ACDM (W2):**
>
> We adjusted this claim in the abstract, and fully revised the argumentation regarding limitations of ACDM at the end of the paper. However, we would like to highlight that ACDM does still have the ability to create posterior samples, while the U-Net baselines with training noise or training rollout do not, as discussed in more detail with respect to W6 below.
>
>
> **Inference Speed (W3):**
>
> We agree that the discussion in terms of advantages and drawbacks could be made more clear. We updated the corresponding sections in the paper and appendix accordingly. While the inference costs of our diffusion based setup are high, we are confident that future work will be able to substantially improve performance with techniques like improved sampling or distillation, similar to the imaging domain. As mentioned in W4, first steps towards this are already taken in some concurrent work.
>
>
> **Comparison to Concurrent Work (W4):**
>
> Thanks for the suggestion, the mentioned paper is an interesting concurrent approach. We included a comparison in more detail in the general response, and discuss the paper and a comparison to ACDM in a revised version of our paper.
>
>
> **Hyperparameter Deviations (W5):**
>
> We comment on each of the mentioned hyperparameter choices in more detail in the revised version of the paper. Specifically, for the number of input steps, we found $k=1$ to work better in early exploration runs for direct predictor models, and $k=2$ to be slightly better for the diffusion setup (also see Appendix B.1 and B.2). However, the differences are quite minor and substantially less than the difference between architectures, e.g., ACDM with $k=1$ achieved about $0.1 * 10^{-3}$ worse results on the $\texttt{Tra}$ test sets in terms of the MSE ($0.1 * 10^{-1}$ for LSiM). Similarly, employing a Huber loss worked slightly better for the ACDM setup than training with MSE, but did not fundamentally alter the results (also with a result difference of about $0.1 * 10^{-3}$ for MSE, and $0.1 * 10^{-1}$ for LSiM on the $\texttt{Tra}$ test sets). Third, all models were trained until the training loss curves were visually converged. This means more complex models like ACDM or the transformer variants required more iterations, while next-step predictor like U-Net or ResNet learn faster. Furthermore, we found that the latter also deteriorate in performance when trained substantially past this convergence point (also see the details in Appendix B.6).

---

> ### Author Response · Authors · 2023-11-21
> **Response to Reviewer wsiC [PART 2/3]**
>
> **Multiple Predictions via Input Perturbations (W6):**
>
> We would like to emphasize that perturbing inputs to achieve multiple predictions actually changes the physics and fundamentally differs from posterior sampling. The former essentially measures how the system reacts to changes in the input from perturbations with forcing, while the latter provides a range of possible and likely results for a single input. The underlying assumption in your suggestion requires that the following two distribution are identical: $s^t = f(s^{t-1} + \mathcal{N}(0, \sigma))$ and $P(s^t|s^{t-1})$. Here, $f$ is a deterministic predictor, e.g. a fluid solver, and $s^{t-1}$ is the input solver state. While we do not provide a formal proof here, one example should illustrate that these distributions can be very different: Assuming $\sigma$ is sufficiently large, the input to $f$ will be so strongly perturbed, that applying $f$ could result in basically any $s^t$, even states that are completely impossible given only $s^{t-1}$. This issue is not fundamentally different for smaller $\sigma$.
>
>
> **Benchmark data sets (W7):**
>
> We opted for creating our own data sets for this work due to the following reasons: The incompressible wake flow is an established standard setup in the CFD community, and in recent works like Um et al. (2020) and Geneva and Zabaras (2022). For the more complex cases, we wanted to employ non-deterministic flow setups to highlight the strengths of diffusion models in terms of posterior sampling (see our comment to Q3 from Reviewer 1NQ6 for more details), that are not directly available from e.g. PDE-Bench. As mentioned in the paper, we will make our full data sets publicly available to download upon acceptance, in addition to the already provided source code that allows for generating the data. Nevertheless, we fully agree on the usefulness of benchmark data sets in general to achieve comparable results.
>
>
> **Hyperparameters of ACDM (W8):**
>
> We adjusted the argumentation with respect to the hyperparameters at the end of the paper. However, we would like the emphasize that our method works out-of-the-box across cases with the given, linear variance schedule (see Appendix B.1) and a high number of diffusion steps. As such, even one additional hyperparameter in the baselines is a drawback. Adjusting the number of diffusion steps is only necessary if inference performance is crucial. Also note that the ACDM performance changes very consistently across different diffusion steps as shown in Appendix C.4. For training noise or the PDE-Refiner method, parameter tuning ranges from crucial to unavoidable, as otherwise these methods do not improve upon a simple U-Net model or even deteriorate results.
>
>
> **Formulation regarding number of diffusion steps (W9):**
>
> As suggested, we adjusted the argumentation regarding the number of diffusion steps in the paper.
>
>
> **Presentation of Fig. 4 and Fig. 6 (W10):**
>
> Thanks for pointing this out, we updated Fig. 4 and the argumentation for Fig. 6 accordingly.

---

> ### Author Response · Authors · 2023-11-21
> **Response to Reviewer wsiC [PART 3/3]**
>
> **CRPS Metric (Q1):**
>
> Thanks for the suggestion, this is an interesting metric which is also use by the concurrent DYffusion method. Our initial evaluation choices were based on established methods for flow physics, and we already investigate a broad range of metrics, but we are happy to consider an evaluation with CRPS in a revised version of our work.
>
>
> **U-Net input steps (Q2):**
>
> As clarified in the appendix now, we found $k=1$ to work better for direct one-step predictor models in early exploration runs, meaning we use $k=1$ for all U-Net, ResNet, and FNO models. However, the changes between different values of $k$ are marginal compared to differences between architectures.
>
>
> **Design of Fig. 5 (Q3):**
>
> Thanks for pointing this out. The top two rows and the row below use different zoom levels, and as such we intended to visually separate them. This is clarified in the axis titles now.
>
>
> **Choices for Latent Size / Hidden Dimension (Q4):**
>
> We investigated larger latent spaces up to $L=256$ for the transformer variants in early exploration runs, but they did not fundamentally improve the behavior. We hypothesize that this is due to a trade-off between an accurate reconstruction and a well-behaved latent space that is easy to evolve temporally. While larger latent spaces tend to produce better reconstruction quality in early time steps, smaller latent spaces are more stable and behave better in terms of temporal evolution. We found $L=32$ to be good trade-off for both aspects. The number of hidden dimensions in the FNO is restricted by the overall number of trainable parameters to achieve a fair comparison with a similar parameter count compared to the other models. This also depends on the number of Fourier modes, as additional modes require more parameters as well. We also investigated a model with a larger hidden dimension of size 224 with (8, 4) modes, i.e. FNO-8, but found even worse performance than FNO-16 or FNO-32.
>
>
> **Number of Diffusion Steps during Training (Q5):**
>
> The number of diffusion steps during training is always equal to the number of diffusion steps during inference, i.e. by default $R=20$ for $\texttt{Inc}$ and $\texttt{Tra}$, and $R=100$ for $\texttt{Iso}$. This is clarified across paper and appendix now.
>
>
> **Number of Posterior Samples (Q6):**
>
> Yes, across evaluations we always use 5 samples for probabilistic methods.
>
>
> **Transformer Details (Q7):**
>
> For TF-MGN, we kept the transformer specification in line with the work from Han et al. (2021), and use transformer decoder layers and residual predictions for the latent model. However, we achieved better results by using transformer encoder layers and direct predictions for the next latent state in the latent model, which we included as the TF-Enc variant. The convolutional encoder and decoder architecture (see Appendix B.5) to embed the data into the latent space is identical across TF-MGN, TF-Enc, and TF-VAE.
>
>
> **Remaining Variants in Tab. 3 (Q8):**
>
> The remaining combinations are included in Tab. 3 and across Appendix C.4 now.
>
>
> **Formulation in Appendix C.5 Summary (Q9):**
>
> We agree that this formulation was unclear, and updated it accordingly.
>
>
> **Performance of Training Noise Variant in Tab. 5 (Q10):**
>
> While we would intuitively also expect a relatively clear trend across noise standard deviations, there is of course no guarantee for a direct, linear relationship. However, we assume this case is caused by a statistical outlier (even though this evaluation should be quite stable as it is computed across 3 random model training runs), as the training with training noise seems to exhibit a higher standard deviation across runs than the other methods.
>
>
> **Additional Visualizations of Ablation Models (Q11):**
>
> We included additional visualizations for these ablation models in Appendix E now.

---

### Official Review · Reviewer_JT77 · 2023-11-01

**Soundness:** 3 good
**Presentation:** 3 good
**Contribution:** 1 poor
**Rating:** 5
**Confidence:** 3

**Summary:**

This paper proposes autoregressive conditional diffusion models (ACDMs) to simulate turbulent flow systems in an autoregressive rollout fashion. The model shows stability on long rollout horizon simulation. The proposed ACDM is further demonstrated to generate posterior samples that align closely with genuine physical dynamics in different fluid dynamics datasets.

**Strengths:**

The paper is well-written, and the presentation is clear. The related works are well examined, encompassing the key areas of interest regarding the use of conditional diffusion models for turbulent flow simulation, and the paper is correctly placed in the current literature. The proposed approach is straightforward and the authors provide several experiments (alongside well-appreciated source code) that make evaluation robust.

**Weaknesses:**

- The novelty is not much - the idea is almost a direct application of conditional diffusion models in autoregressive settings.
- The inherent resolution at which a diffusion model generalizes is predefined during its training. This set resolution potentially restricts the model's flexibility, thereby impacting its practical utility and adaptability in diverse applications.
- As depicted in Table 2, the enhancement in accuracy across the five datasets is marginal, with ACDM outperforming other models only on two datasets in terms of LSiM error. Moreover, it is unclear what is generally the best model from this result, given that except FNO, all models seem to obtain at least one best result across datasets.
- Despite its acknowledgment of the limitations, the substantial computational cost of ACDM considerably undermines its practical utility. As state, the model can generate a solution in ~0.2 seconds; with 1000 time steps, this sums up to more than 3 minutes for a single trajectory against ~11 seconds for UNet. Therefore, it is hard to assess the Pareto-efficiency of the proposed method.
- The organization of the paper would benefit from distinguishing between the preliminary and methodology sections; Section 3 is a mix of both and as such it is hard to distinguish the real contribution.
- It would be useful to assess the performance on different datasets, such as from PDEBench [1], particularly in the 3D cases.
- Stronger baselines could be chosen. For instance, graph neural networks have shown good performance in fluid dynamics such as [2]. Moreover, FNO variants such as AFNO have been shown to be powerful in large-scale real datasets [3].
- [Minor] The explanation for Figure 4 is unclear. It appears that each subfigure lacks a title indicating the respective dataset name.

---


[1 ]Takamoto, Makoto, et al. "PDEBench: An extensive benchmark for scientific machine learning." NeurIPS (2022).
[2] Li, Zongyi, et al. "Fourier neural operator with learned deformations for pdes on general geometries." arXiv preprint arXiv:2207.05209 (2022).
[3]  Pathak, Jaideep, et al. "Fourcastnet: A global data-driven high-resolution weather model using adaptive fourier neural operators." arXiv preprint arXiv:2202.11214 (2022).

**Questions:**

1. When the time step between two rollout steps is set to a fixed interval, how can we accurately capture the dynamics between these designated rollout steps?
2. How would the model perform with real, possibly noisy data, such as the Black Sea and ScalarFlow datasets as in [1]?
3. This is an additional question that does not influence the score (since the paper should be considered as "concurrent work"). How do you think your proposed model would fare against [2]?

---

[1 ] Lienen, Marten, and Stephan Günnemann. "Learning the dynamics of physical systems from sparse observations with finite element networks." ICLR 2022.
[2] Lippe, Phillip, et al. "Pde-refiner: Achieving accurate long rollouts with neural pde solvers." NeurIPS (2023).

---

> ### Author Response · Authors · 2023-11-21
> **Response to Reviewer JT77 [PART 1/2]**
>
> We would like to thank Reviewer JT77 for the detailed review and feedback. In the following, we address mentioned weaknesses (W) and questions (Q).
>
>
> **Novelty (W1):**
>
> While the novelty of our approach with regard to DDPM is agreeably limited, we would argue that our paper nevertheless makes a valuable contribution. Exploring a direct adoption of DDPMs to flow prediction is clear gap in the concurrent work, and an important step that should be taken before moving on to more advanced methods. To achieve this, we thoroughly investigate various tasks with different difficulty, while comparing to a range of established baselines using different physically motivated metrics. We also comment on this aspect in more detail in the general response.
>
>
> **Generalization to different resolutions (W2):**
>
> We agree that generalizing to different data resolutions is an interesting direction for future work. It is a fundamental problem across the range of neural network architectures we consider in this work, and even fully convolution methods typically struggle to generalize well far outside the training resolution domain.
>
>
> **Accuracy (W3):**
>
> While it is correct that ACDM does not beat every baseline on every data set we consider, it is important to keep the bigger picture in mind and consider the following aspects: First, ACDM is within a small margin to the best method across the test sets shown in Tab. 2. This stability is unmatched compared across baselines, all of which perform substantially worse than ACDM on two or more test sets. Especially see the substantial accuracy increase of ACDM on the most complex $\texttt{Iso}$ case, compared to all other models. Second, ACDM allows for posterior sampling, while all baselines expect TF-VAE are deterministic. Third, ACDM is clearly more temporally stable than all baselines from Tab. 2, as shown for example in Fig. 8.
>
>
> **Sampling Speed (W4):**
>
> We agree that the sampling costs of ACDM in its current form can be steep compared to other methods. However, we are confident that future work will be able to substantially improve performance with techniques like better sampling or distillation, as shown by substantial recent advances in the imaging domain. First steps towards such improvements are already taken in concurrent work, as discussed in more detail in the general response.
>
>
> **Paper Organization (W5):**
>
> Thank you for the suggestion, we adjusted the subsection titles accordingly to make this more clear. We will consider a larger reorganization of this section in the next revision of our work.
>
>
> **Different Data Sets (W6):**
>
> We agree that further data sets are always an option to increase the scope of a paper, but we would like to highlight that we already evaluate a range of diverse data sets with different complexity. The incompressible wake flow is relatively simple, but a widely established standard setup in the CFD community. The transonic wake flow is fundamentally more difficult due to the formation of shock waves and the complex vortex shedding that has a range of spectral components in addition to the main shedding frequency. Finally, the isotropic turbulence case is based on direct numerical simulation and highly underdetermined due to the sliced evaluation. Thus, it is unlikely that additional data sets would fundamentally help to improve the answer regarding the central question of our paper: "How does a DDPM setup with appropriate conditioning fare on flow prediction?".
>
>
> **Additional Baselines (W7):**
>
> We would like to point out that we consider a wide range of diverse baselines, as well as averages over multiple model runs to achieve a stable and meaningful evaluation. We chose established baseline classes like FNO, ResNets, or U-Nets that are very commonly used across problems. Graph networks definitely represent an interesting class of architectures, however, for the Cartesian data sets at hand, they would not provide any benefits. But this is certainly an interesting direction for future work, as we do not see any reason why the positive aspects of diffusion models should not carry over to graph networks.
>
>
> **Fig. 4 (W8):**
>
> Thanks for pointing this out, it is fixed in the revised version.

---

> ### Author Response · Authors · 2023-11-21
> **Response to Reviewer JT77 [PART 2/2]**
>
> **Temporal super-resolution (Q1):**
>
> This is an interesting question, however we believe that it is going beyond the scope of our work. Any autoregressive rollout method that is not explicitly conditioned on physical time does not have the ability to achieve a finer temporal resolution compared to the data (including all our baselines). Of course, traditional temporal super-resolution methods like linear or higher-order interpolation between prediction steps, or training a designated interpolator network are an option. This is an interesting direction that other works like the DYffusion paper mentioned in the general response are already targeting.
>
>
> **Real-World Data Sets (Q2):**
>
> We do not see any problem with noisy, real-world data for our method. As shown in the paper, it is robust to interpolation as well as extrapolation tasks. We also experimented with running ACDM from a simple mean flow initialization on $\texttt{Inc}$ or $\texttt{Tra}$ during inference (without any re-training). We observed that ACDM was able to successfully return to a stable, oscillating prediction, without any stability problems, even from such an extreme out-of-distribution test. Thus, noisy real-world data should not fundamentally impact the approach either.
>
>
> **Concurrent Work (Q3):**
>
> We provide a detailed analysis compared to PDE-Refiner in Appendix C.9 in the revised version of our paper. To summarize, we find that ACDM has a range of benefits of PDE-Refiner, with the only drawback being the inference performance, as also mentioned in more detail in the general response.

---

### Author Response · Authors · 2023-11-21
**General Response to all Reviewers [PART 1/2]**

We are thankful to all reviewers for their time and for the provided feedback. We would like to comment on two relevant aspects for all reviewers here, but we also provide detailed answers to every reviewer individually. In addition, we already revised our paper with respect to a range of issues mentioned in the reviews. Major changes in the paper compared to the initial version are highlighted in green text color.

**Concurrent Work:**

We would like to explicitly discuss two concurrent methods (DYffusion [1] and PDE-Refiner [2]) that will appear in NeurIPS 2023, which were also mentioned by the Reviewers JT77 and wsiC. Both have a similar focus as our work, but all three approaches are conceptually different. These works point to the large interest in the research community, and we believe that our work sets the stage for both papers by providing a honest, thorough and direct comparison of how DDPM-like methods work in the context of time-dependent physics simulations.

*[1]:* The DYffusion approach is a highly interesting direction based on a jointly trained predictor+interpolator scheme. The predictor is motivated by traditional diffusion models, however Cachay et al. equate the diffusion time step with the physical time step of the simulation. This leads to a model that is conceptually halfway between our unrolled U-Net and ACDM. Furthermore, a probabilistic interpolator is used that allows for temporal predictions beyond the time step in the data set. Based on our estimate, our method has the following advantages compared to DYffusion: ACDM achieves better temporal coherence, and more variance in the posterior samples. Both advantages are caused by our fully diffusion-based approach, compared to the Bayesian interpolator model based on dropout from DYffusion. On the other hand, the DYffusion approach has a range of clear benefits as well, as the method can use very large time steps and it flexibly generalizes to arbitrary prediction time intervals. Finally, DYffusion is also significantly faster during inference than ACDM, due to equating the diffusion time step with physical time.

*[2]:* The PDE-Refiner approach relies on a refinement of direct one-step predictions, by adding noise and denoising the result with a traditional diffusion-based setup. Lippe et al. reformulate the standard diffusion training objective, such that a single model learns one-step predictions in the first step, and the refinement in all later steps. As this approach is conceptually closer to our experiments than DYffusion, we implemented it for a direct comparison against ACDM, by closely following the provided pseudocode. The only difference is, that we use our U-Net as the backbone model for a fair comparison. In a revised version of our paper, Appendix C.9 contains a detailed description of our experiments with PDE-Refiner. To summarize, while PDE-Refiner can improve upon the U-Net baseline, ACDM still has the following advantages:
1) ACDM achieves better accuracy than all PDE-Refiner models across test sets (based on a large grid-search of PDE-Refiner variants).
2) ACDM works out-of-the-box with the provided diffusion schedule and a large number of diffusion steps (which can be reduced with predictable behavior for better inference performance). PDE-Refiner requires a detailed hyperparameter search over both key parameters, the number of refinement steps as well as the minimum noise variances for each data set, to achieve better temporal stability compared to U-Net.
3) ACDM results in clearly better posterior coverage, due to PDE-Refiners approach of probabilistic iterative refinement of a deterministic prediction.
Then only major advantage of PDE-Refiner against ACDM is a faster inference. PDE-Refiner is about a factor of four to eight times slower compared to simple one-step predictors like our U-Net model (depending on the number of refinement steps), while ACDM requires about twenty to hundred times more time compared to our U-Net model (depending on the number of diffusion steps).

---

> ### Author Response · Authors · 2023-11-21
> **General Response to all Reviewers [PART 2/2]**
>
> **Novelty and Contribution:**
>
> It was mentioned across several reviews that our work exhibits little novelty. However we would like to point out that our work does make a valuable contribution to the ICLR community. We believe that a careful adaptation of *unmodified* techniques from diffusion methods in the field of image generation is actually still missing in the concurrent publications discussed above. Thus, our submission establishes an important baseline and a fundamental first step, that we believe should be taken before moving on to more advanced methods, such as the ones above [1,2]. We would argue, that our work actually highlights the importance of the overall direction and the concurrent methods, filling in gaps that these papers did not investigate in much detail. Furthermore, we provide a detailed benchmark, ranging from commonly used baselines models to very challenging flow problems, with physically meaningful evaluation metrics. We also provide a range of detailed ablation studies and further insights in our appendix, such as different variants of noisy/non-noisy conditioning. Naturally, we will release our source code and data sets to ensure the that others can use our model architectures and datasets as a benchmark.
>
> There is actually a third paper on time-dependent physics problems with diffusion models [3] at NeurIPS 2023, which, however, more strongly differs in focus. Taken together, these works show a strong interest in the direction of diffusion models in scientific machine learning, and we believe that our submission has an important message for the research community: *Even a vanilla DDPM with an appropriate conditioning fares very well on flow prediction.* To make clear that we are not inventing a new method, we renamed the main method in our paper from "ACDM" to "cDDPM". For clarity, we still refer to the method as "ACDM" in our responses to the reviews, but this change is already reflected in the revised text of the paper (the figures will be updated accordingly for the camera-ready version). We would be very happy if the reviewers could reconsider their evaluations with respect to the novelty and overall contribution of our work.
>
>
> [1] S. Cachay, B. Zhao, H. Joren, and R. Yu. Dyffusion: A dynamics-informed diffusion model for spatiotemporal forecasting. arXiv, 2023. doi:10.48550/arXiv.2306.01984
>
> [2] P. Lippe, B. Veeling, P. Perdikaris, R. Turner, and J. Brandstetter. Pde-refiner: Achieving accurate long rollouts with neural pde solvers. arXiv, 2023. doi:10.48550/arXiv.2308.05732
>
> [3] B. Holzschuh, S. Vegetti, N. Thuerey. Solving Inverse Physics Problems with Score Matching. arXiv, 2023. doi:10.48550/arXiv:2301.10250

---

### Meta-Review · Area_Chair_z1Ey · 2023-12-06

**Metareview:**

This submission deals with the simulation of turbulent flows and proposes an autoregressive diffusion model. The 4 expert reviewers were critical and raised the issues on

- novelty,
- limitations (eg fixed resolution),
- experimental setup, and fairness of the experiments,
- lukewarm performance,
- high computational complexity,
- lack of baselines.

Novelty lies in the eye of the beholder, but reviewers agreed that the paper proposes a direct application of diffusion models to fluid simulations and therefore the interest lies only in the question on how well this kind of methods work. Combined with the lukewarm performance and the fact that this leads to limitations in terms of computational complexity and also conditions of operation, the interest is therefore somewhat marginal.

The authors provided answers in their rebuttal but could not convincingly address the issues. The AC judges that the paper is not yet ready for publication.

**Justification For Why Not Higher Score:**

-

**Justification For Why Not Lower Score:**

-

---

### Decision · Program_Chairs · 2024-01-16

Reject